# M18BP1 valency and a distributed interaction footprint determine epigenetic centromere specification in humans

Kai Walstein [ID] [1,4,6], Louisa Hill [ID] [1,6], Doro Vogt [ID] [1], Lina Oberste-Lehn [ID] [1], Petra Janning [2], Ingrid R Vetter [ID] [1], Dongqing Pan [1,5] & Andrea Musacchio [ID] [1,3 ✉]

## Abstract

**The histone H3 variant CENP-A is considered an epigenetic landmark of centromeres. Its deposition reflects cell-cycle-regulated assembly of M18BP1, HJURP, and PLK1 on a divalent MIS18α/β scaffold. The localization determinants of this machinery remain poorly characterized. Here, we report that in human cells, artificial M18BP1 dimerization bypasses MIS18α/β, allowing the identification of at least four determinants of M18BP1 centromere localization. These include the SANTA domain, of which we report the first structure, as well as linear motifs in disordered neighboring regions, of which we characterize the interaction footprint on the CENP-A-associated 16-subunit constitutive centromere-associated network (CCAN). Our observations imply that M18BP1, after dimerization, is necessary and sufficient for centromere localization. Its cell-cycle-dependent dimerization on MIS18α/β promotes initial recognition of a multivalent centromeric assembly of old CENP-A and associated proteins, followed by cooption of PLK1 and HJURP and new CENP-A deposition. Our results shed new light on the determinants of centromere epigenetic inheritance in humans.**

**Keywords** Centromere; Kinetochore; Cell Cycle; CENP-A Loading; M18BP1
**Subject Categories** Cell Cycle; Chromatin, Transcription & Genomics; Structural Biology

## Introduction

Chromosomes are carriers of the genome, and their faithful distribution to the daughter cells during a mother cell's equational or reductional divisions (mitosis and meiosis, respectively) is essential for the propagation of life. These processes begin with the assembly of a cytoskeletal structure in the mother cell known as the spindle. Chromosomes interact with spindle microtubules through kinetochores, macromolecular complexes built on dedicated chromosome loci known as centromeres (Musacchio and Desai, 2017). Attainment of chromosome biorientation licenses mitotic exit, allowing the irreversible separation of chromosomes and their segregation into the daughter cells. The histone H3 variant centromere protein A (CENP-A) is an almost ubiquitous landmark of centromeres in eukaryotes (McKinley and Cheeseman, 2016; Mitra et al, 2020b; Rowley and Jansen, 2025). To build the kinetochore, CENP-A interacts with the constitutive centromere-associated network (CCAN), a supramolecular assembly of 16 proteins in human that is also largely conserved in evolution. The CCAN, in turn, recruits the components of the outer kinetochore, which form the microtubule-binding interface (McKinley and Cheeseman, 2016; Musacchio and Desai, 2017).

What ensures that centromeres maintain their size and position on chromosomes is a crucial and only partly answered question. Because CENP-A undergoes a 2-fold dilution during DNA replication, when it is redistributed to the sister chromatids, the reduction of its levels must be compensated through new CENP-A deposition. Foundational work demonstrated that this occurs in the G1 phase of the cell cycle, and thus before the dilution of CENP-A during S-phase (Jansen et al, 2007; Schuh et al, 2007). In primates, including humans, CENP-A associates with an abundant 171-bp DNA repeat known as α-satellite, which was therefore initially identified as a putative genetic determinant of centromeres (McKinley and Cheeseman, 2016; Mitra et al, 2020b; Rowley and Jansen, 2025). As new CENP-A is deposited near the old CENP-A, it might be expected to be directly attracted to vacant α-satellite repeats. However, the great variability of repeat sequences in different organism, and the discovery in several organisms of chromosomes with centromeres established on non-repetitive but diverse DNA sequences, forced to abandon the idea of a strict relationship between DNA sequence and centromere identity, in favor of a model in which a self-sustaining protein-based template maintains centromere identity through subsequent cell divisions (McKinley and Cheeseman, 2016; Mitra et al, 2020b). Reconstitution of ectopic centromeres after forcing repositioning of CENP-A or other CCAN subunits proved an important prediction of this

[1]Department of Mechanistic Cell Biology, Max Planck Institute of Molecular Physiology, Otto-Hahn-Straße 11, Dortmund 44227, Germany. [2]Mass Spectrometry Facility, Max Planck Institute of Molecular Physiology, Otto-Hahn-Straße 11, Dortmund 44227, Germany. [3]Centre for Medical Biotechnology, Faculty of Biology, University Duisburg-Essen, Essen, Germany. [4]Present address: Centre for Clinical Trials Münster, Münster 48149, Germany. [5]Present address: Research Division, Chugai Pharmaceutical Co., Ltd., Yokohama 244-8602, Japan. [6]These authors contributed equally: Kai Walstein, Louisa Hill. ✉E-mail: andrea.musacchio@mpi-dortmund.mpg.de

model (Barnhart et al, 2011; Gascoigne et al, 2011; Hori et al, 2013; Mendiburo et al, 2011).

Further support to an epigenetic model of centromere specification based on a self-sustaining protein-based template came from the discovery of dedicated CENP-A deposition machinery (Mitra et al, 2020b; Stellfox et al, 2013). In humans, this machinery includes the Mis18 complex (a 4:2 hexamer of the MIS18α and MIS18β subunits, henceforth MIS18α/β), M18BP1 (also known as KNL2), the specialized CENP-A:H4 chaperone HJURP, and Polo-like kinase 1 (PLK1) (Dunleavy et al, 2009; Foltz et al, 2009; Fujita et al, 2007; Hayashi et al, 2004; Maddox et al, 2007; McKinley and Cheeseman, 2014). Interaction of these proteins is disallowed by cyclin-dependent kinase (CDK) phosphorylation from late G1 phase to M-phase (Conti et al, 2024; McKinley and Cheeseman, 2014; Pan et al, 2017; Parashara et al, 2024; Silva et al, 2012; Spiller et al, 2017; Stankovic et al, 2017) (Fig. 1A). Rapid decline of CDK activity at the metaphase-to-anaphase transition permits interactions among these proteins that promote their recruitment and activation at centromeres and the discharge of new CENP-A near the existing CENP-A pool (Jansen et al, 2007). Thus, CENP-A deposition occurs during the G1 phase of the cell cycle, before DNA replication, contrary to the replenishment of H3:H4, which occurs concomitantly with its dilution during DNA replication (Alabert et al, 2015; Xu et al, 2010).

The epigenetic specification model puts emphasis on the protein-based interactions between centromeres and CENP-A loading machinery (herewith CALM) that promote CENP-A deposition at the appropriate time and place. Nonetheless, the precise features that allow the CALM to become recruited specifically to the centromere remain largely elusive. In previous work, specific subunits of the CCAN, most notably CENP-C and CENP-I, were implicated in centromere recruitment of the CALM (Dambacher et al, 2012; French and Straight, 2019; Hoffmann et al, 2020; McKinley and Cheeseman, 2014; Mitra et al, 2020a; Moree et al, 2011; Shono et al, 2015). However, both subunits are also important for CCAN stability (Basilico et al, 2014; Guo et al, 2017; Klare et al, 2015; McKinley et al, 2015; Pesenti et al, 2022; Tachiwana et al, 2015; Walstein et al, 2021), and whether they are directly involved as CALM receptors or rather in the stabilization or recruitment of other CCAN subunits acting as CALM receptors remains unclear. On the CALM side, MIS18α/β has been proposed to cooperate with M18BP1 for kinetochore recruitment (Fujita et al, 2007; Pan et al, 2017; Stellfox et al, 2016). Furthermore, also HJURP was proposed to contain autonomous centromere-binding regions (Flores Servin et al, 2023; French et al, 2017; Tachiwana et al, 2015). Recent work, however, demonstrated that centromere recruitment of HJURP in early G1, at least in humans, is entirely contingent on its ability to interact with MIS18α/β under the control of PLK1 kinase activity (Conti et al, 2024; Parashara et al, 2024). Thus, collectively, the view of how the CALM is recruited to centromeres remains fragmented. Here, we shed light on this crucial question by mobilizing a combination of complex biochemical reconstitutions, cell biology in human cells, experimental structural work, and structural modeling.

## Results

### Sequence requirements for kinetochore recruitment of M18BP1

Residues 1–140 of human (Hs) M18BP1 (M18BP1$^{1-140}$) are necessary for kinetochore recruitment of M18BP1 (Pan et al, 2017; Stellfox et al, 2016; Thamkachy et al, 2024). Two CDK phosphorylation sites within this fragment, T40 and S110, prevent binding to MIS18α/β. Their dephosphorylation upon mitotic exit allows binding of M18BP1 to MIS18α/β and robust centromere localization of the resulting octameric complex (McKinley and Cheeseman, 2014; Ohzeki et al, 2016; Pan et al, 2017; Silva et al, 2012; Spiller et al, 2017; Stankovic et al, 2017; Stellfox et al, 2016; Thamkachy et al, 2024). These observations are consistent with the idea that the centromere-binding determinants of M18BP1 are insufficient for robust centromere localization and need to be combined with those on MIS18α/β for stable recruitment (Stellfox et al, 2016). Indeed, M18BP1$^{141-1132}$, which lacks the MIS18α/β binding region in M18BP1$^{1-140}$, is not recruited to centromeres (Pan et al, 2017).

Nonetheless, we reasoned that an alternative interpretation of these observations is that dimerization of M18BP1 on MIS18α/β effectively makes it divalent, potentially increasing its kinetochore binding affinity (Erlendsson and Teilum, 2020; Mammen et al, 1998). This alternative interpretation would explain our puzzling previous observation that fusing M18BP1$^{141-1132}$ to an N-terminal GST, a strong dimer, allowed robust kinetochore localization of the resulting construct despite its inability to bind MIS18α/β (Pan et al, 2017). In this view, phosphorylation of Thr40 and Ser110 may effectively control the valency of M18BP1 and its effective binding affinity for centromeres in G1.

To test this idea, we initially verified through comparison of mass photometry and mass spectrometry measurements that fusion of M18BP1$^{312-490}$ to GST, but not MBP, increased molecular mass (Fig. EV1A–E). Next, we asked if ectopic, cell-cycle invariant dimerization of M18BP1 through GST would allow us to identify localization determinants of M18BP1 and their cell cycle regulation. Towards this goal, we initially generated stable HeLa cell lines for inducible expression of various M18BP1 deletion mutants tagged with EGFP (and other tags as indicated). Constructs encompassing residues 1–490 of M18BP1 (M18BP1$^{1-490}$) localized robustly to kinetochores in G1 phase, in agreement with a previous study (McKinley and Cheeseman, 2014), and regardless of artificial GST dimerization (Appendix Fig. S1A,B,D). However, the GST-fused M18BP1$^{1-490}$ construct localized robustly also in mitosis, when the interaction with the MIS18 complex is suppressed by CDK phosphorylation (Appendix Fig. S1C,E; Fig. EV1F–H). This GST-fused construct was expressed at much lower levels relative to its counterpart lacking GST (Appendix Fig. S1A; Fig. EV1F–H). Likely due to overexpression, the latter showed strong diffuse chromatin localization in addition to an unfocused kinetochore signal, in line with a previous study (McKinley and Cheeseman, 2014) (Appendix Fig. S1C,E). Comparison of kinetochore levels relative to the distribution on chromatin demonstrated the effect of GST-based dimerization as a facilitator of localization (Fig. EV1F–H). Thus, binding determinants for the loading machinery at the kinetochore may exist not only in G1, but also in mitosis, suggesting that they are not cell-cycle regulated, contrary to the CENP-A loading machinery.

When gauged against our previous observation that the deleterious effects on kinetochore localization resulting from deleting residues 1–140 of M18BP1 are rescued by GST fusion (Pan et al, 2017), these results strongly suggest that residues 141–490 of M18BP1 contain crucial kinetochore binding determinants. To further characterize these determinants, we built three

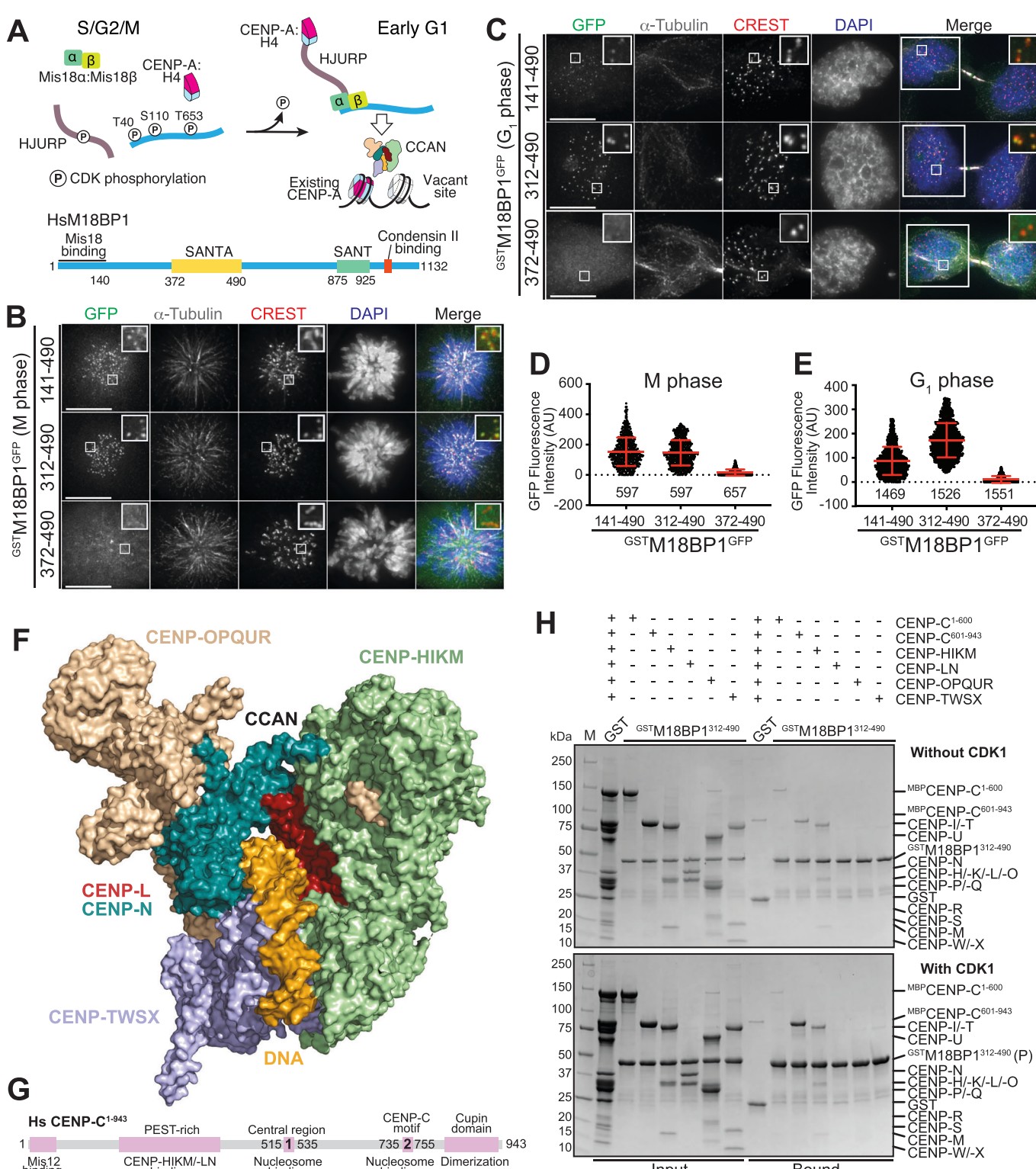

**A** S/G2/M — Early G1

HsM18BP1

**B** GSTM18BP1GFP (M phase)

**C** GSTM18BP1GFP (G1 phase)

**D** M phase

**E** G1 phase

**F** CCAN

**G** Hs CENP-C1-943

**H**

additional stable cell lines expressing segments 141–490, 312–490, and 372–490 fused to GST and EGFP (Appendix Fig. S1F) and assessed their localization in G1- and M-phases. GSTM18BP1141–490-EGFP and GSTM18BP1312–490-EGFP, but not GSTM18BP1372–490-EGFP, localized robustly to mitotic kinetochores (Fig. 1B,D). Similarly,

GSTM18BP1141–490-EGFP and GSTM18BP1312–490-EGFP, but not GSTM18BP1372–490-EGFP, localized to kinetochores in G1, with GSTM18BP1312–490-EGFP showing even more robust localization (Fig. 1C,E). Thus, residues 312–490 of M18BP1, which encompass the SANTA (SANT-associated) domain (372–490) and a segment

**Figure 1.   Identification of a minimal centromere targeting module of M18BP1.**

(**A**) Schematic model of the CDK phosphorylation-regulated assembly of the Mis18 complex and its centromeric recruitment in G1. HJURP is phosphorylated at multiple sites during mitosis (Flores Servin et al, 2023; Muller et al, 2014; Stankovic et al, 2017; Wang et al, 2014). (**B, C**) Representative images of fixed HeLa cells showing GFP fluorescence of indicated stably expressed M18BP1 variants and immuno-stained α-tubulin in mitosis and G1. Centromeres were visualized by CREST sera, and DNA was stained by DAPI. White scale bars indicate 10 μm. (**D, E**) Quantification of the centromeric GFP-fluorescence intensities of the indicated expressed M18BP1 variants in mitosis and G1. Centromeres were detected in the CREST channel using a script for semiautomated quantification. Red bars indicate the mean and standard deviation of all quantified centromere fluorescence intensities (*n*, as indicated below each plot) from three independent experiments. Here and in other equivalent panels reporting quantifications of fluorescence intensity, the number of counted kinetochores is indicated. (**F**) Structural model of the constitutive centromere-associated network (CCAN) bound to DNA (PDB 7R5S). Due to its large disordered regions, CENP-C was largely invisible in the structure. (**G**) Organization of human CENP-C. The arrangement of the binding sites recapitulates the outer-to-inner kinetochore axis. (**H**) GSH-resin pull-down assay with GST (negative control) and $^{GST}$M18BP1$^{312-490}$ as baits. The added CCAN subcomplexes are indicated above each lane. The experiment was performed in the absence (upper gel) and in the presence (lower gel) of CCC kinase complex. Source data are available online for this figure.

N-terminal to the SANTA (pre-SANTA) predicted to lack structural order (312–371), are a minimal kinetochore-targeting module of M18BP1 when fused to GST.

## M18BP1$^{312-490}$ binds the CCAN

We harnessed in vitro biochemistry to identify the binding determinants of M18BP1$^{312-490}$ within the kinetochore. In solid phase binding assays, M18BP1$^{312-490}$ bound to CENP-A and H3 nucleosome core particles (H3$^{NCPs}$ and CENP-A$^{NCPs}$) indistinguishably (Appendix Fig. S2A). Furthermore, incorporation of two centromere-specific histone marks, Histone H4 Lys20 monomethylation or α-amino-trimethylation of the CENP-A N-terminus (H4-K20me1 or CENP-A-Nme3, introduced enzymatically with SETD8 and NRMT1, respectively, in the presence of S-adenosyl methionine) (Hori et al, 2014; Sathyan et al, 2017), did not affect the apparent affinity of M18BP1$^{312-490}$ for CENP-A nucleosomes in electrophoretic mobility shift assays (EMSAs) (Appendix Fig. S2B–D). In EMSA assays, M18BP1$^{312-490}$ also bound DNA (Appendix Fig. S2E). These interactions, whose likely primary determinant is unspecific DNA binding, are unlikely to explain the exquisitely specific kinetochore localization of M18BP1$^{312-490}$.

We therefore turned onto the CCAN, which associates with CENP-A at the centromere-kinetochore interface. CCAN consists of several stable subcomplexes (Fig. 1F). Its interaction with the CENP-A nucleosome is mediated by CENP-C (Fig. 1G), a 943-residue disordered protein with multiple interaction motifs disseminated along its length. We asked whether recombinant CCAN subunits or subcomplexes interacted with $^{GST}$M18BP1$^{312-490}$ immobilized on solid phase (Walstein et al, 2021; Weir et al, 2016). These assays identified interactions with the C-terminal region of CENP-C (CENP-C$^{601-943}$, also indicated as CENP-C$^C$) and with the CENP-HIKM complex. No binding or only weak binding was observed with CENP-C$^{1-600}$, CENP-LN, CENP-OPQUR, or CENP-TWSX (Fig. 1H, *upper panel*). Phosphorylation of the CCAN subunits with CDK1/Cyclin B/CKS1 kinase (CCC complex) and ATP (Huis In 't Veld et al, 2022) caused a strong enhancement of the interaction with CENP-C$^C$ (Fig. 1H, lower panel), suggesting regulation of this interaction by mitotic phosphorylation (see below). Further dissection demonstrated that the entire M18BP1$^{312-490}$ segment was required for robust binding to CENP-C$^{601-943}$, as neither of two sub-fragments, M18BP1$^{312-371}$ and M18BP1$^{372-490}$ fused with GST and immobilized on solid phase bound it robustly (Fig. 2A). Conversely, $^{GST}$M18BP1$^{312-371}$ was sufficient to bind the CENP-HIKM complex, whereas no binding to $^{GST}$M18BP1$^{372-490}$ (SANTA domain) was observed (Fig. 2B).

These results implicate the disordered segment M18BP1$^{312-371}$ (herewith also referred to as pre-SANTA region) in CENP-C$^{601-943}$ and CENP-HIKM binding. They also implicate the SANTA domain in CENP-C$^{601-943}$ binding, and indicate a potential role of mitotic CDK phosphorylation in modulating the interaction with CENP-C (summarized in Fig. 2C).

## Crystal structure of the SANTA domain

We determined a crystal structure of a fusion construct of the SANTA domain (see "Methods"), the first for this domain class (Zhang et al, 2006), to a Bragg spacing of 2.4 Å (1 Å = 0.1 nm. See Appendix Table S1 for a summary of relevant crystallographic data). The folded region begins around residue 380 and ends at residue 490. It consists of a small globular domain formed by a pair of three- and four-stranded β-sheets packing against each other through a shared hydrophobic core. This core β-domain is followed by three consecutive α-helices, the third considerably longer than the first two, packing against one edge of the globule (Fig. 2D). A search of the protein databank for structurally related domains using DALI (Holm, 2022) did not identify folds closely related to the SANTA domain, while the Foldseek server (van Kempen et al, 2024) identified β-domains closely related to the SANTA's in the predicted structures of several bacteriophages.

The SANTA domain displays several highly conserved residues, distributed both in its hydrophobic core or on its surface (Fig. 2E. These residues are respectively indicated with black and green asterisks under the alignment in Fig. 2F). Among the most prominent surface features are the parallel "tracks" of the exposed side chains of Trp413 and His414 on the β4 strand (indicated with a white asterisk in Fig. 2E), as well as a prominent hydrophobic cavity (indicated with a red asterisk) delimited by the side chain of Lys468, at the beginning on the α3 helix. Together with the side chains of other highly conserved residues, including Glu420, Arg421, and Ser431, we predict these neighboring prominent conserved surface features to be part of a continuous, convex binding interface for an unknown target, likely an extended, flexible motif (see Discussion).

## Kinetochore-binding determinants of M18BP1$^{312-490}$

To probe the significance of the interactions of M18BP1$^{312-490}$ with CCAN for kinetochore recruitment of M18BP1, we introduced mutations in the pre-SANTA region and SANTA domain. Specifically, we created a double alanine mutant of Trp413 and

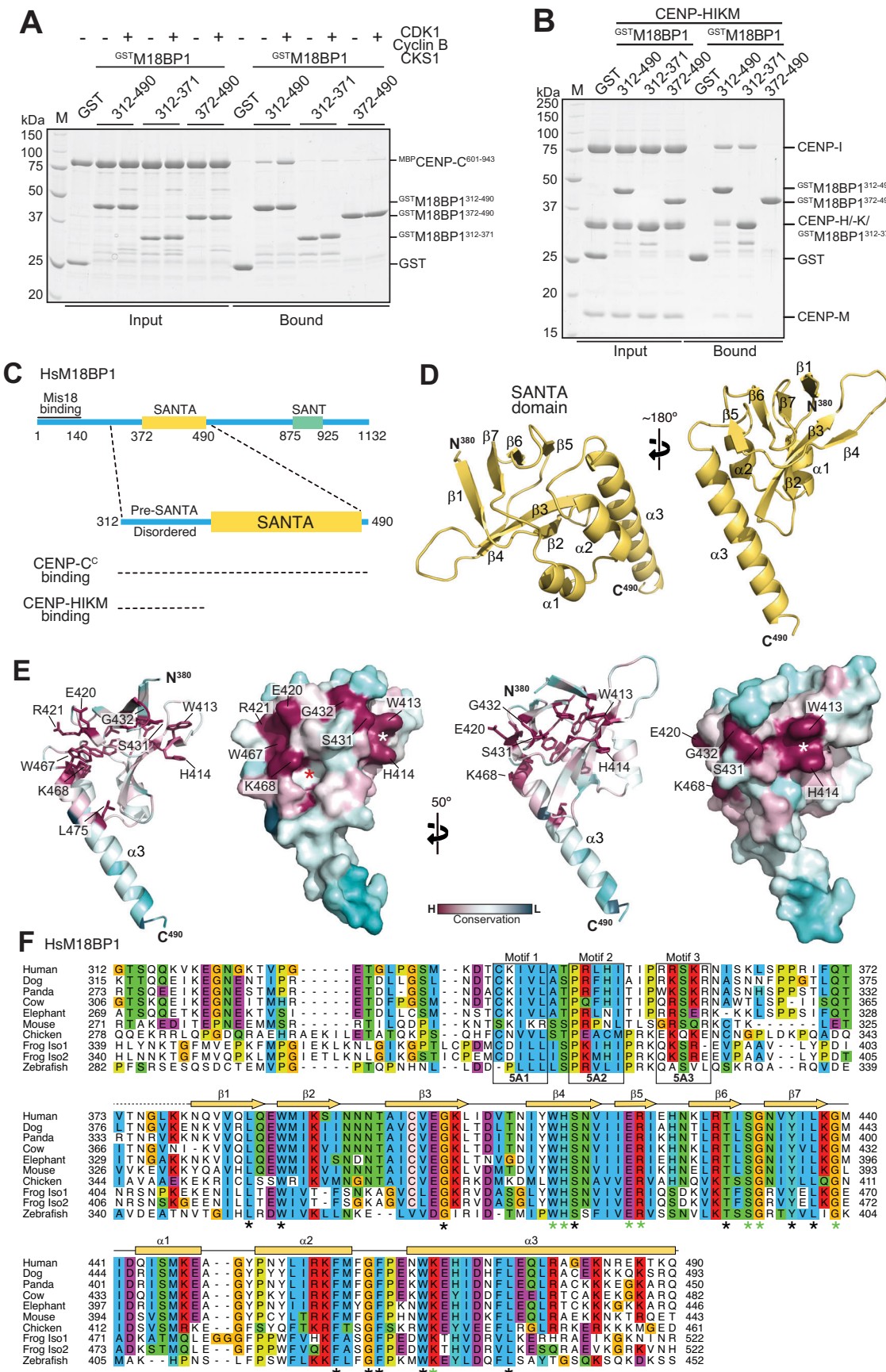

◀ **Figure 2. The SANTA domain and its N-terminal extension provide distinct binding sites for the CCAN.**

(A) GSH-resin pull-down assay with GST (negative control) and ᴳˢᵀM18BP1 fragments as baits testing the binding to ᴹᴮᴾCENP-C⁶⁰¹⁻⁹⁴³. The M18BP1 fragments used as baits and the addition of CCC kinase complex are indicated above each lane. (B) GSH-resin pull-down assay with GST (negative control) and ᴳˢᵀM18BP1 fragments as baits, testing the binding to CENP-HIKM complex. The M18BP1 fragments used as baits are indicated above each lane. (C) Graphical summary of the obtained results in (A, B). (D) Crystal structure of the human M18BP1 SANTA domain depicting the folded region from residue 380 to 490. The SANTA domain was crystallized as the fusion protein construct EGFP-**AS**-SANTA-**LEGT**-anti-GFP Nanobody-GGHHHHHH (see "Methods"). Only the SANTA moiety is shown, while the rest was omitted from the representation. (E) Structure of the SANTA domain. Conservation is indicated by a color code ranging from high (magenta) to low (turquoise) conservation from sequences in the alignment in (F). A prominent hydrophobic cavity and the exposed side chains of Trp413 and His414 are indicated with a red and a white asterisk, respectively. (F) Multiple sequence alignment showing conserved residues within M18BP1³¹²⁻⁴⁹⁰ across mammalian and non-mammalian vertebrate species. Conserved residues in the hydrophobic core and on the surface are indicated with black and green asterisks, respectively. Source data are available online for this figure.

His414 (the WH/AA mutant) to try impair the function of the SANTA domain. We also identified three short conserved motifs in the pre-SANTA region (Motifs 1–3, with the following boundaries and sequences: Motif 1, 340-CKIVL-344; Motif 2, 347-PRLHI-351; Motif 3, 355-RRSKR-359) and generated three five-alanine mutants, named respectively 5A1, 5A2, and 5A3 (Fig. 2F) to inactivate them. We introduced these mutations in the ᴳˢᵀM18BP1³¹²⁻⁴⁹⁰⁻ᴱᴳᶠᴾ construct and generated stable cell lines (Fig. 3A; Appendix Fig. S1G). The 5A1, 5A3, and WH/AA mutations all impaired kinetochore recruitment of ᴳˢᵀM18BP1³¹²⁻⁴⁹⁰⁻ᴱᴳᶠᴾ in early G1 cells (Fig. 3B,D). Albeit expressed at lower levels, the 5A2 mutant retained the ability to decorate kinetochores. In mitosis, neither the 5A1 nor the WH/AA mutant decorated kinetochores, while both 5A2 and 5A3 did (Fig. 3C,D). Thus, Motif 3 is necessary for interphase recruitment of ᴳˢᵀM18BP1³¹²⁻⁴⁹⁰⁻ᴱᴳᶠᴾ, but dispensable for its mitotic localization, possibly because increased binding affinity caused by CDK1 activity (presented below) reduces dependency on Motif 3 during mitosis. As the SANTA domain is essential for mitotic as well as interphase localization of ᴳˢᵀM18BP1³¹²⁻⁴⁹⁰⁻ᴱᴳᶠᴾ, we surmise that its centromere target persists during the cell cycle.

We used in vitro binding assays to identify potential binding partners of Motifs 1–3 and of the SANTA domain. Initially, we immobilized wild-type and mutant ᴳˢᵀM18BP1³¹²⁻⁴⁹⁰ and tested their binding to the CENP-HIKM complex. Both the 5A2 and 5A3 mutations impaired binding of ᴳˢᵀM18BP1³¹²⁻⁴⁹⁰ to CENP-HIKM, whereas 5A1 and the SANTA mutants bound normally (Fig. 3E). To identify binding determinants of ᴳˢᵀM18BP1³¹²⁻⁴⁹⁰ and CENP-HIKM, we performed crosslinking-mass spectrometry (XL-MS) experiments with CENP-HIKM and wild-type ᴳˢᵀM18BP1³¹²⁻⁴⁹⁰, using the 5A3 mutant as a negative control (Fig. EV2A,B). ᴳˢᵀM18BP1³¹²⁻⁴⁹⁰ crosslinked prominently with CENP-K, and less prominently with CENP-H and CENP-I, while the 5A3 mutant showed only a few crosslinks. Albeit structurally stable (Fig. EV2C), a CENP-HIKM mutant complex carrying substitutions at residues identified in the XL-MS experiment did not interact with ᴳˢᵀM18BP1³¹²⁻⁴⁹⁰ in in vitro binding assays (Fig. EV2D,E). To assess if the presence of the Mis18 complex interfered with the interaction with CENP-HIKM, we immobilized ᴹᴮᴾM18BP1¹⁻⁴⁹⁰ and added CENP-HIKM in the presence or absence of the Mis18 complex. CENP-HIKM binding was essentially identical, indicating that it is not influenced by the interaction of ᴹᴮᴾM18BP1¹⁻⁴⁹⁰ with the Mis18 complex (Fig. EV2F).

AlphaFold (AF) (Abramson et al, 2024; Jumper et al, 2021) predicts that Motif 2 adopts an extended conformation and pokes its hydrophobic side chains into a cradle between the CENP-H and CENP-K helices (Fig. EV2G; Appendix Fig. S3A). The positively charged residues of Motif 3 are predicted to interact with an acidic patch comprising Glu182, Glu185, Asp186, Glu214, and Glu217 of CENP-K (Fig. EV2G), which were all targeted in our CENP-HIKM mutant construct. Thus, the crosslinking data provide a good explanation of the effects of mutations in Motifs 2 and 3.

Next, we immobilized wild-type and mutant ᴳˢᵀM18BP1³¹²⁻⁴⁹⁰ and tested binding to ᴹᴮᴾCENP-C⁶⁰¹⁻⁹⁴³ (Fig. 3F). The 5A1 mutant appeared to disrupt CENP-C binding regardless of its CDK phosphorylation status. The 5A2, 5A3, and WH/AA mutants, on the other hand, bound immobilized ᴹᴮᴾCENP-C⁶⁰¹⁻⁹⁴³, but less strongly than the wild-type protein, with 5A2 apparently eliminating stimulation of binding by CDK phosphorylation (an observation on which we return below). To analyze how CDK phosphorylation increases CENP-Cᶜ binding, we pre-phosphorylated ᴳˢᵀM18BP1³¹²⁻⁴⁹⁰ or ᴹᴮᴾCENP-C⁶⁰¹⁻⁹⁴³, and only then tested binding to the unphosphorylated partner (Fig. 3G). These results demonstrated unequivocally that the phosphorylation modulating the interaction is on ᴳˢᵀM18BP1³¹²⁻⁴⁹⁰, as the phosphorylated ᴹᴮᴾCENP-C⁶⁰¹⁻⁹⁴³ bound ᴳˢᵀM18BP1³¹²⁻⁴⁹⁰ to levels comparable to those in the absence of phosphorylation. Conversely, binding to ᴹᴮᴾCENP-C⁶⁰¹⁻⁹⁴³ was strongly enhanced by CDK phosphorylation of ᴳˢᵀM18BP1³¹²⁻⁴⁹⁰ (Figs. 3G and EV1I). Using CENP-C⁶⁰¹⁻⁹⁴³ fused to GFP as an immobilized prey, we also verified that fusing M18BP1³¹²⁻⁴⁹⁰ to GST positively affected binding relative to the MBP fusion (Fig. EV1J).

The 5A3 mutant, but not the WH/AA mutant, affected nucleosome and DNA binding (Appendix Fig. S2A,D–H), likely because positively charged residues in Motif 3, mutated in 5A3, mediate non-specific DNA binding (like the WH/AA mutant, also the 5A1 and 5A2 mutants bound nucleosomes; KW and AM, unpublished observations). In size-exclusion chromatography experiments with phosphorylated proteins, a direct interaction of ᴳˢᵀM18BP1³¹²⁻⁴⁹⁰ with CENP-A nucleosomes was not visible, likely because of fast complex dissociation. Conversely, ᴹᴮᴾCENP-C⁶⁰¹⁻⁹⁴³ bound robustly to CENP-A nucleosomes and to ᴳˢᵀM18BP1³¹²⁻⁴⁹⁰ (Appendix Fig. S4A,B). In line with the solid phase experiments, ᴳˢᵀM18BP1³¹²⁻⁴⁹⁰⁻⁵ᴬ¹ did not interact with ᴹᴮᴾCENP-C⁶⁰¹⁻⁹⁴³, whereas ᴳˢᵀM18BP1³¹²⁻⁴⁹⁰⁻⁵ᴬ³ and ᴳˢᵀM18BP1³¹²⁻⁴⁹⁰⁻ᵂᴴ/ᴬᴬ retained significant binding affinity (Appendix Fig. S4C).

## The role of CENP-C

Thus, collectively, our binding analyses in vivo and in vitro indicate that Motif 1 and the SANTA domain are critical binding determinants to mitotic kinetochores. Motif 1 mediates an interaction with CENP-Cᶜ, while the main target of the SANTA domain appears not to be present in our reconstitutions. Motif 3,

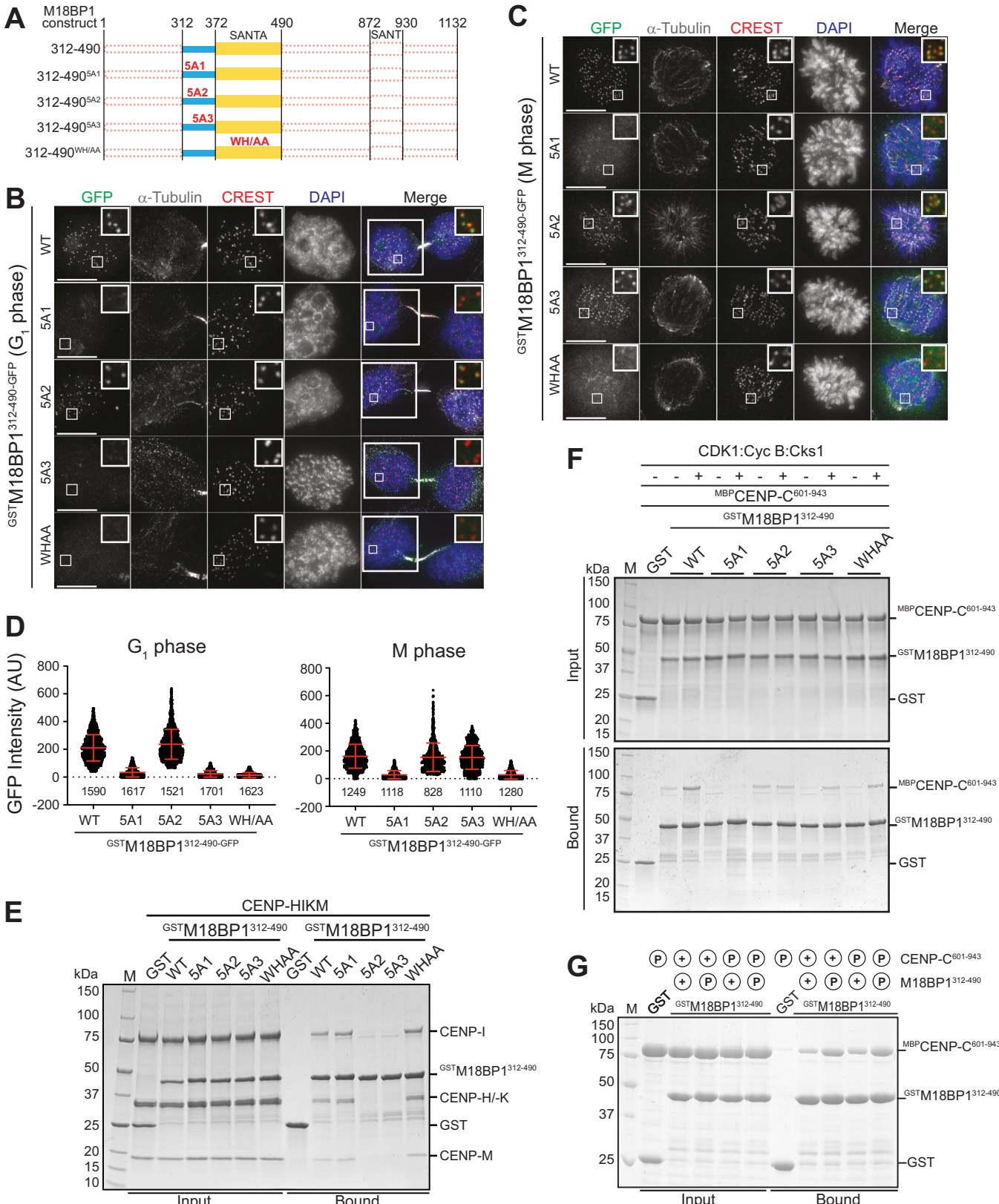

◀ **Figure 3. Unraveling the crucial interactions for mitotic and G1 localization of M18BP1.**

(A) Schematic showing the expressed $^{GST}$M18BP1$^{312-490-GFP}$ variants. (B, C) Representative images of fixed HeLa cells showing GFP fluorescence of indicated stably expressed M18BP1 variants and immuno-stained α-tubulin in G1 and mitosis, respectively. Centromeres were visualized by CREST sera, and DNA was stained by DAPI. White scale bars indicate 10 μm. (D) Quantification of the centromeric GFP-fluorescence intensities of the indicated expressed M18BP1 variants in G1 and mitosis, respectively. Centromeres were detected in the CREST channel using a script for semiautomated quantification. Red bars indicate the mean and standard deviation of all quantified centromere fluorescence intensities (*n*, as indicated below each plot) from three independent experiments. (E) GSH-resin pull-down assay with GST (negative control) and $^{GST}$M18BP1$^{312-490}$ variants as baits, testing the binding to CENP-HIKM complex. The M18BP1 variants used as baits are indicated above each lane. (F) GSH-resin pull-down assay with GST (negative control) and $^{GST}$M18BP1$^{312-490}$ variants as baits testing the binding to $^{MBP}$CENP-C$^{601-943}$. The M18BP1 variants used as baits and the addition of CDK1:Cyclin B:CKS1 kinase complex are indicated above each lane. (G) GSH-resin pull-down assay with GST (negative control) and $^{GST}$M18BP1$^{312-490}$ as baits testing the binding to $^{MBP}$CENP-C$^{601-943}$. In this experiment, $^{GST}$M18BP1$^{312-490}$ and $^{MBP}$CENP-C$^{601-943}$ were individually pre-phosphorylated by CDK1:Cyclin B:CKS1 kinase complex as indicated by a red encircled P above each lane. Source data are available online for this figure.

while apparently dispensable in mitosis, contributes to kinetochore localization in interphase, likely by binding to CENP-HIKM or DNA. To corroborate the role of CENP-C in M18BP1 localization, we rapidly depleted CENP-C endogenously tagged with an auxin-inducible degron (Fachinetti et al, 2015) by the addition of indole acetic acid (IAA). CENP-C depletion prevented kinetochore decoration of electroporated $^{GST}$M18BP1$^{312-490}$ both in interphase and in mitosis, confirming that CENP-C is essential for recruitment of $^{GST}$M18BP1$^{312-490}$ (Fig. 4A,B). Importantly, we have previously shown that the localization of other CCAN subunits after rapid removal of CENP-C is not affected at early time points (Pesenti et al, 2022; Schweighofer et al, 2025), suggesting that CENP-C is directly involved in M18BP1 localization, and not indirectly through the stabilization of other CCAN subunits. The C-terminal nucleosome-binding motif of CENP-C (the CENP-C motif, Fig. 1G) did not appear to be required for the interaction with M18BP1, as mutations in it did not affect the interaction with M18BP1 (Appendix Fig. S5A). Rather, cross-linking-mass spectrometry experiments were consistent with a putative interaction of $^{GST}$M18BP1$^{312-490}$ with the Cupin domain (Appendix Fig. S5B,C). Indeed, size-exclusion chromatography experiments demonstrated a robust interaction of the CENP-C Cupin domain (residue 775–943, lacking the nucleosome-binding motif) with $^{GST}$M18BP1$^{312-490}$ (Appendix Fig. S5D). AF3 predicted that Motif 1 of M18BP1 interacts with a pocket in the Cupin domain of CENP-C, exposing the side chains of several conserved hydrophobic and positively charged residues (Fig. 4C; Appendix Fig. S3B). Combining mutations V858E, K880A, L890R, and F938A in this pocket (abbreviated as CENP-C$^{Cmut}$, for Cupin mutant) entirely abrogated binding to $^{GST}$M18BP1$^{312-490}$ (Fig. 4D).

Thr346, at the boundary between motifs 1 and 2, is situated in a Thr-Pro motif expected to be a substrate for proline-directed CDK activity. The AF prediction shows the phosphate to point towards Ly880 of the Cupin domain (Fig. 4C). Using ProQ staining, which visualizes phosphorylation, we detected reduced phosphorylation of the 5A2 mutant (Appendix Fig. S1H), likely explaining why its binding to CENP-C (Fig. 3F) appeared insensitive to phosphorylation. Mutation of T346 to Val or Glu decreased phosphate incorporation by CDK1 to an even higher extent (Appendix Fig. S1H). Mutation of M18BP1$^{Thr346}$ to Glu did not create a phospho-mimetic mutant, as the levels of $^{MBP}$CENP-C$^{601-943}$ bound to $^{GST}$M18BP1$^{312-490-T346E}$ were no higher, and in fact slightly lower, than those observed with wild-type $^{GST}$M18BP11$^{312-490}$. Nonetheless, the mutation prevented enhanced binding of $^{MBP}$CENP-C$^{601-943}$ in the presence of CDK activity (Fig. 4E). Essentially identical results were observed when Thr346 was mutated to Val, or after including

an additional alanine mutation in another putative CDK site, Ser365, which removed phosphorylation of $^{GST}$M18BP1$^{312-490}$ altogether (Fig. 4E; Appendix Fig. S1H).

## CENP-A deposition assays identify an additional kinetochore-targeting region

$^{GFP}$M18BP1$^{1-490}$ contains the MIS18-binding region (Figs. 1A and 5A) and localizes to kinetochores in G1 (Appendix Fig. S1B,D). We therefore expected this construct to retain CENP-A loading capabilities in the absence of endogenous M18BP1. Confirming the efficiency of endogenous M18BP1 depletion, deposition of new CENP-A was entirely impaired in a cell line not expressing a $^{GFP}$M18BP1 rescue construct (Fig. EV3A,B). In agreement with our prediction, $^{GFP}$M18BP1$^{1-490}$ promoted robust CENP-A deposition in the early G1 phase in cells depleted of endogenous M18BP1 (Fig. 5B–D; expression levels of rescue constructs are in Appendix Fig. S1I). Conversely, all mutations affecting G1 kinetochore localization of $^{GFP}$M18BP1$^{1-490}$, including the 5A1, 5A3, and WH/AA mutants, also prevented its ability to support CENP-A loading, confirming that centromere localization is essential for M18BP1's function in CENP-A loading (Fig. 5A–D). These mutants failed to localize also when expressed in the presence of endogenous M18BP1, and accordingly also displayed a dominant-negative effect on the ability of endogenous M18BP1 to load CENP-A (Fig. EV3C, quantified in Fig. 5C,D). In contrast, even if interfering with the interaction with CENP-HIKM in vitro, the 5A2 mutant showed robust CENP-A loading in G1 (Fig. EV3D–F), in agreement with our observation that it lacks penetrance even when introduced in the minimal localization module $^{GST}$M18BP1$^{312-490-GFP}$ (Fig. 3B–D).

Next, we repeated these experiments with the same mutants introduced in full-length M18BP1 rather than M18BP1$^{1-490}$ (Fig. 5E; expression levels of rescue constructs are in Appendix Fig. S1J). The detrimental effects of the 5A1 and 5A3 mutants on localization and CENP-A loading observed in the context of M18BP1$^{1-490}$ were largely or completely rescued when the same mutations were expressed in the context of full-length M18BP1 (Figs. 5F–H and EV4A). These observations suggest that regions downstream of residue 490 in the mutant full-length constructs contribute to kinetochore recruitment, thus rescuing their kinetochore localization in the presence of mutations that instead impair M18BP1$^{1-490}$ localization. Only the WH/AA mutant in the SANTA domain, as well as a new deletion mutant where the entire SANTA domain had been deleted, were unable to localize to kinetochores and to load CENP-A in the absence of endogenous M18BP1 (Figs. 5F–H and EV4A–D). The SANTA

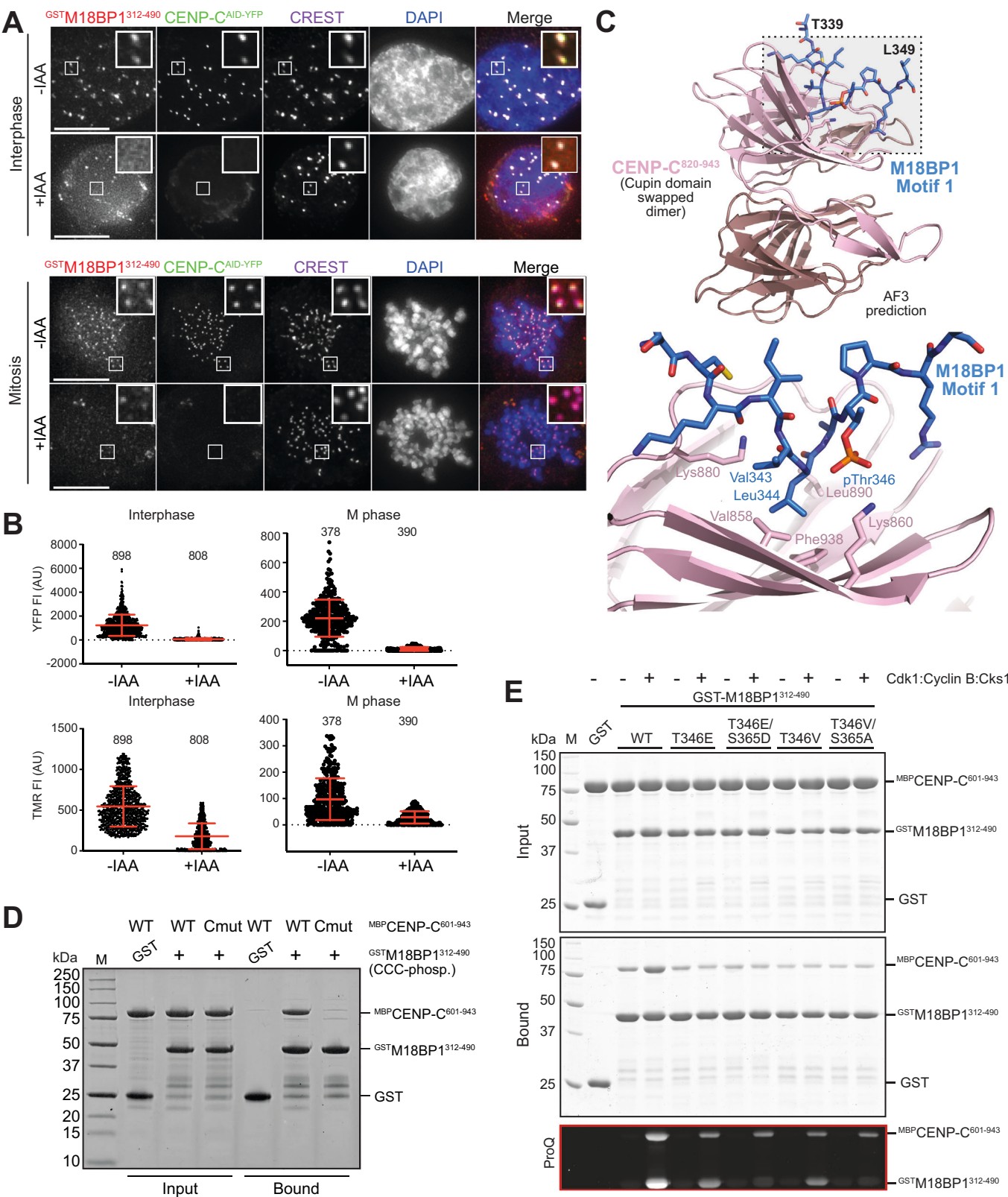

◄  **Figure 4.   Rapid CENP-C depletion prevents efficient M18BP1 recruitment.**

(A) Representative images of fixed DLD-1 cells showing YFP fluorescence of endogenous CENP-C$^{AID-YFP}$, TMR fluorescence of electroporated recombinant $^{GST}$M18BP1$^{312-490}$, and immuno-stained α-tubulin in interphase (top) and mitosis (bottom). Centromeres were visualized by CREST sera, and DNA was stained by DAPI. Cells were treated with IAA as indicated to rapidly degrade endogenous CENP-C. White scale bars indicate 10 μm. (B) Quantification of the centromeric YFP-fluorescence intensities of endogenous CENP-C and the centromeric TMR-fluorescence intensities of the electroporated M18BP1 protein in interphase and mitosis. Centromeres were detected in the CREST channel using a script for semiautomated quantification. Red bars indicate the mean and standard deviation of all quantified centromere fluorescence intensities ($n$, as indicated below each plot) from three independent experiments. (C) AlphaFold prediction of the interaction of M18BP1 Motif 1 with the CENP-C Cupin domain. (D) GSH-resin pull-down assay with GST (negative control) and $^{GST}$M18BP1$^{312-490}$ as baits testing the binding to $^{MBP}$CENP-C$^{601-943}$ WT and Cmut (V858E, K880A, L890R, and F938A) variants. The CENP-C variants used and, when applicable, the presence of CDK1:Cyclin B:CKS1 kinase complex are indicated above each lane. (E) GSH-resin pull-down assay with GST (negative control) and $^{GST}$M18BP1$^{312-490}$ variants as baits testing the binding to $^{MBP}$CENP-C$^{601-943}$. The M18BP1 variants used as baits, and the addition of CDK1:Cyclin B:CKS1 kinase complex are indicated above each lane. ProQ staining demonstrating the phosphorylation intensities of the different $^{GST}$M18BP1$^{312-490}$ variants and $^{MBP}$CENP-C$^{601-943}$ in the bound fraction. Source data are available online for this figure.

deletion mutant, and to a minor extent the WH/AA mutant, had a dominant-negative effect on CENP-A loading in the presence of endogenous M18BP1, an effect that was more prominent in cells with high expression levels of these M18BP1 mutants (Fig. EV4E–H). These observations demonstrate that the SANTA domain is a crucial determinant of M18BP1 localization and CENP-A loading.

Next, we aimed to identify regions of M18BP1 downstream of residue 490 that contribute to kinetochore localization in the presence of the 5A3 mutation. For this, we fused various C-terminal fragments to a core segment encompassing residues 1–490 and carrying the 5A3 mutation (Fig. 5I). We then asked which of these constructs localized correctly to kinetochores in G1 phase, thus rescuing the localization defect of M18BP1$^{1-490-5A3}$ and if they loaded new CENP-A, in the presence or absence of endogenous M18BP1. These experiments unequivocally identified residues 491-872 as being necessary and sufficient for rescuing the kinetochore recruitment and CENP-A loading deficiency when fused to M18BP1$^{1-490-5A3}$ (while deletion of the 491–872 segment did not result in a dominant-negative effect on CENP-A loading. Figures 5J–L and EV5A; Appendix Fig. S1K shows expression levels of these constructs).

## Cell-cycle control of M18BP1 localization

The M18BP1 region comprised between 491 and 872 includes a previously identified CDK site, Thr653, that contributes to the suppression of mitotic localization of M18BP1 (Stankovic et al, 2017). As our analysis indicates that this residue is contained in a region contributing to kinetochore localization of M18BP1, we investigated its mechanism of action in mitosis and interphase. First, we generated $^{GST}$M18BP1$^{EGFP}$ constructs encompassing the 312–490 or 312–872 fragments and carrying both the 5A1 and 5A3 mutations to completely inactivate the kinetochore localization determinants of the pre-SANTA region (Fig. 6A; Appendix Fig. S1L). Neither mutant localized to kinetochores in mitosis, but further mutating Thr653 to Val in the context of $^{GST}$M18BP1$^{312-872-5A1/5A3-EGFP}$ rescued robust kinetochore localization (Fig. 6B,D). As the region preceding the SANTA domain is impaired by the 5A1 and 5A3 mutations, this result is a strong indication that phosphorylation of Thr653 prevents kinetochore localization by suppressing a localization determinant within the 491–872 fragment. This determinant ought to be functional in interphase, when CDK activity is at its lowest levels. Accordingly, the $^{GST}$M18BP1$^{312-872-5A1/5A3-EGFP}$ construct localized robustly to kinetochores in G1 phase, and the Thr653 mutant did not enhance its localization (Fig. 6C,E).

As a further demonstration that activation of the 491–872 segment requires dephosphorylation of Thr653 and that it can bypass a localization requirement for the pre-SANTA region of M18BP1, we generated new constructs lacking the pre-SANTA region altogether (Figs. 6F and EV5B). Unlike the control construct containing the pre-SANTA region, $^{GST}$M18BP1$^{372-872-EGFP}$ failed to localize to mitotic kinetochores, but localized robustly to kinetochores in G1 phase, i.e., when Thr653 is expected to become dephosphorylated (Fig. 6G–J). On the other hand, the $^{GST}$M18BP1$^{372-872-T653V-EGFP}$ construct localized robustly to both mitotic and interphase kinetochores (Fig. 6G–J). Thus, phosphorylation of T653 appears to control the function of a kinetochore-localization determinant in the 491–872 region. The $^{GST}$M18BP1$^{491-872-GFP}$ construct, which additionally removed the SANTA domain, was unable to decorate kinetochores, regardless of the cell cycle phase and of whether the deletion was combined with the T653V mutation (Fig. EV5C–F,G). Thus, the kinetochore-localization determinant in the 491–872 region is insufficient for kinetochore recruitment in the absence of the SANTA domain.

We hypothesized that Thr653 may be part of a motif directly involved in the interaction with a centromere receptor, and that its phosphorylation may directly obstruct binding, a hypothesis reinforced by strong evolutionary sequence conservation around this site. We therefore deleted residues 630–660, comprising Thr653 and its surrounding region, and evaluated the localization of the resulting construct ($^{GST}$M18BP1$^{372-872-Δ630-660-EGFP}$) in mitosis and G1. Contrary to our hypothesis, $^{GST}$M18BP1$^{372-872-Δ630-660-EGFP}$ localized to kinetochores in both interphase and mitosis, i.e., it had a localization pattern identical to that of the construct carrying the T653V mutation (Figs. 6G–J and EV5H). These observations suggest that phosphorylation of Thr653 acts on M18BP1 localization from a distance, rather than in the immediate vicinity of the phosphorylation site.

## A crucial kinetochore-binding determinant in M18BP1$^{491-872}$

N-terminally just adjacent to the 630-660 region of M18BP1 is a conserved sequence motif (residues 623–630) bearing sequence similarity to Motif 1, the CENP-C binding motif of M18BP1 examined above. A related sequence motif is also present in HJURP (Appendix Fig. S5E). AF3 predicts this new motif, which we refer to as Motif 4, to bind the same pocket of the Cupin domain bound by Motif 1 (Appendix Fig. S3C). In vitro, however, we only observed modest binding to the CENP-C Cupin domain of recombinant

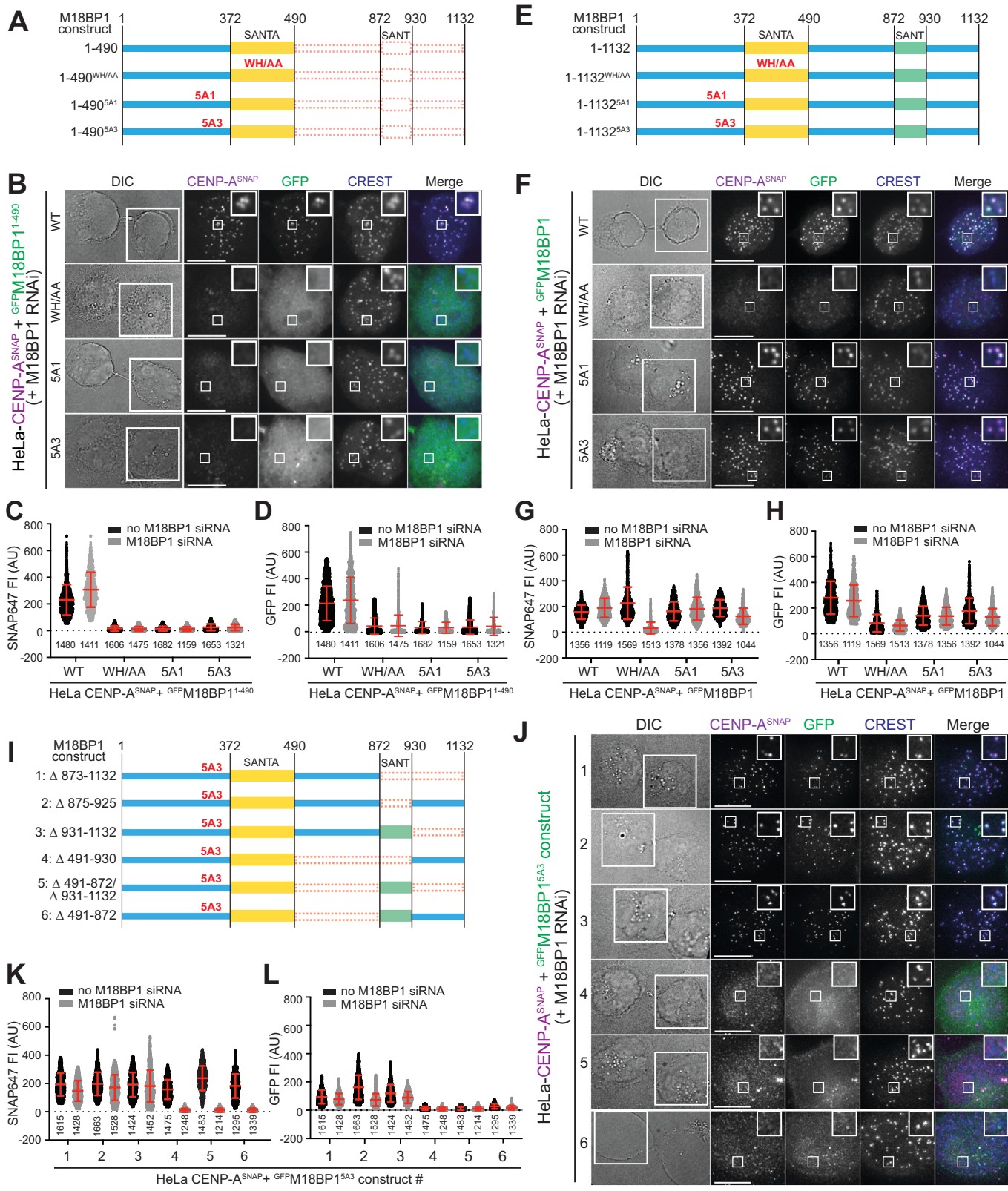

**Figure 5. A C-terminal region in M18BP1 provides further binding affinity to the centromere.**

(A) Schematic showing the expressed $^{GFP}$M18BP1$^{1-490}$ variants in (B). (B) Representative images of fixed HeLa cells showing CENP-A$^{SNAP}$ fluorescence labeled with SNAP-Cell 647-SiR and GFP fluorescence of indicated stably expressed M18BP1 variants in early G1. Centromeres were visualized by CREST sera. The G1 couple is shown in the differential interference contrast (DIC) channel. White scale bars indicate 10 μm. (C, D) Quantification of the centromeric CENP-A-SNAP and centromeric GFP fluorescence intensities, respectively, of HeLa cell lines stably expressing the indicated M18BP1 variants. Centromeres were detected in the CREST channel using a script for semiautomated quantification. Red bars indicate the mean and standard deviation of all quantified centromere fluorescence intensities (n, as indicated below each plot) from three independent experiments. (E) Schematic showing the expressed $^{GFP}$M18BP1$^{1-1132}$ variants in (F). (F) Representative images of fixed HeLa cells showing CENP-A$^{SNAP}$ fluorescence labeled with SNAP-Cell 647-SiR and GFP fluorescence of indicated stably expressed M18BP1 variants in early G1. Centromeres were visualized by CREST sera. The G1 couple is shown in the differential interference contrast (DIC) channel. White scale bars indicate 10 μm. (G, H) Quantification of the centromeric CENP-A-SNAP and centromeric GFP fluorescence intensities, respectively, of HeLa cell lines stably expressing the indicated M18BP1 variants. Centromeres were detected in the CREST channel using a script for semiautomated quantification. Red bars indicate mean and standard deviation of all quantified centromere fluorescence intensities (n, as indicated below each plot) from three independent experiments. (I) Schematic showing the expressed $^{GFP}$M18BP1$^{5A3}$ truncation variants in (J). (J) Representative images of fixed HeLa cells showing CENP-A$^{SNAP}$ fluorescence labeled with SNAP-Cell 647-SiR and GFP fluorescence of indicated stably expressed M18BP1 variants in early G1. Centromeres were visualized by CREST sera. The G1 couple is shown in the differential interference contrast (DIC) channel. White scale bars indicate 10 μm. (K, L) Quantification of the centromeric CENP-A-SNAP and centromeric GFP fluorescence intensities, respectively, of HeLa cell lines stably expressing the indicated M18BP1 variants. Centromeres were detected in the CREST channel using a script for semiautomated quantification. Red bars indicate the mean and standard deviation of all quantified centromere fluorescence intensities from three independent experiments. (L) Quantification of the centromeric GFP-fluorescence intensities of HeLa cell lines stably expressing the indicated M18BP1 variants. Centromeres were detected in the CREST channel using a script for semiautomated quantification. Red bars indicate the mean and standard deviation of all quantified centromere fluorescence intensities (n, as indicated below each plot) from three independent experiments. Source data are available online for this figure.

M18BP1 constructs encompassing Motif 4 (M18BP1$^{581-630}$ or M18BP1$^{591-640}$; LH and AM, unpublished observations). Nonetheless, we decided to test the effects of removing this sequence motif, which lies in a region predicted to be disordered (see AF prediction Q6P0N0). Like the wild-type $^{GST}$M18BP1$^{372-872}$ counterpart, $^{GST}$M18BP1$^{372-872}$ carrying mutations in Motif 4 (residues 623–630) to alanine ($^{GST}$M18BP1$^{372-872-623-630A-EGFP}$, Fig. EV5H) was not recruited to mitotic kinetochores. In interphase, there was residual recruitment of the mutant construct to chromatin foci, only a subset of which neighbored kinetochores but without overlapping with them (Fig. 7A–D). These observations indicate that Motif 4 encodes an additional determinant of kinetochore localization of M18BP1 in G1 phase.

### Determinants on M18BP1 are sufficient for kinetochore localization upon dimerization

To assess if the kinetochore-binding determinants of M18BP1 are sufficient for kinetochore recruitment after dimerization, we expressed full-length $^{GST-GFP}$M18BP1 or $^{GST-GFP}$M18BP1$^{T653V}$ (Fig. EV5I) and ascertained that these constructs, as expected, were recruited to kinetochores in mitotic cells (Fig. 7E,F). As phosphorylation of T40 and S110 prevents the interaction of M18BP1 with the MIS18 complex in mitosis, $^{mCherry}$MIS18α co-expressed in the same cells was not identified at kinetochores (Fig. 7E,F), demonstrating that the kinetochore localization determinants of M18BP1 are sufficient for kinetochore recruitment. Conversely, expression of $^{GFP}$M18BP1$^{T40V/S110A/T653V}$, devoid of GST but allowing an ectopic interaction with the MIS18 complex in mitosis, led to substantial accumulation of co-expressed $^{mCherry}$MIS18α to mitotic kinetochores (Fig. 7E,F), in agreement with the idea that M18BP1 recruits the MIS18 complex when allowed to dimerize on it.

## Discussion

Unraveling the localization determinants of the CALM is key to decipher the molecular basis of epigenetic centromere inheritance.

The work presented here on human M18BP1 rationalizes a large body of seemingly puzzling previous observations on the requirements for CALM localization. Previous work on M18BP1 had brought to light important species-specific features, for instance, in nematodes (de Groot et al, 2021; Prosee et al, 2021). More widespread, a CENP-C-like motif present in M18BP1 orthologs in birds, reptiles, frogs, fish, and plants was shown to promote kinetochore localization of M18BP1, but the motif is not present in mammals (French and Straight, 2019; French et al, 2017; Hori et al, 2017; Kral, 2015; Sandmann et al, 2017). The CENP-C-like motif of M18BP1, where present, appears to interact directly with CENP-A, like genuine CENP-C motifs (preprint: Brown et al, 2025; Jiang et al, 2023; Kato et al, 2013). Its deletion, however, only reduces M18BP1 kinetochore levels (French and Straight, 2019), suggesting the existence of additional localization determinants.

The identity of these additional determinants, and how mammalian M18BP1 replaces the CENP-C-like motif, has remained unclear. Previous work had reported that deletions of the SANTA domain do not prevent M18BP1 localization in humans, chicken, or Arabidopsis (Hori et al, 2017; Lermontova et al, 2013; Stellfox et al, 2016). More recent work, however, identified a requirement for the SANTA domain for kinetochore localization of M18BP1 in humans and Xenopus (French and Straight, 2019; French et al, 2017). Our results provide new strong support for the essentiality of the SANTA domain in centromere recruitment. However, while essential for kinetochore recruitment, the SANTA domain is insufficient for it. We comment on this more extensively below. The SANT domain of mammalian M18BP1 has also been implicated in centromere localization (Dambacher et al, 2012). Here, we found no evidence supporting a major involvement of the SANT in M18BP1 localization, although we cannot exclude minor, non-essential roles.

Another aspect that had remained unclear is whether the localization determinants in M18BP1 are sufficient for kinetochore recruitment. M18BP1 localization to centromeres is contingent on the interaction with the MIS18α/β hexamer, which binds two M18BP1 molecules (Pan et al, 2017; Spiller et al, 2017; Subramanian et al, 2016; Thamkachy et al, 2024). By fusing

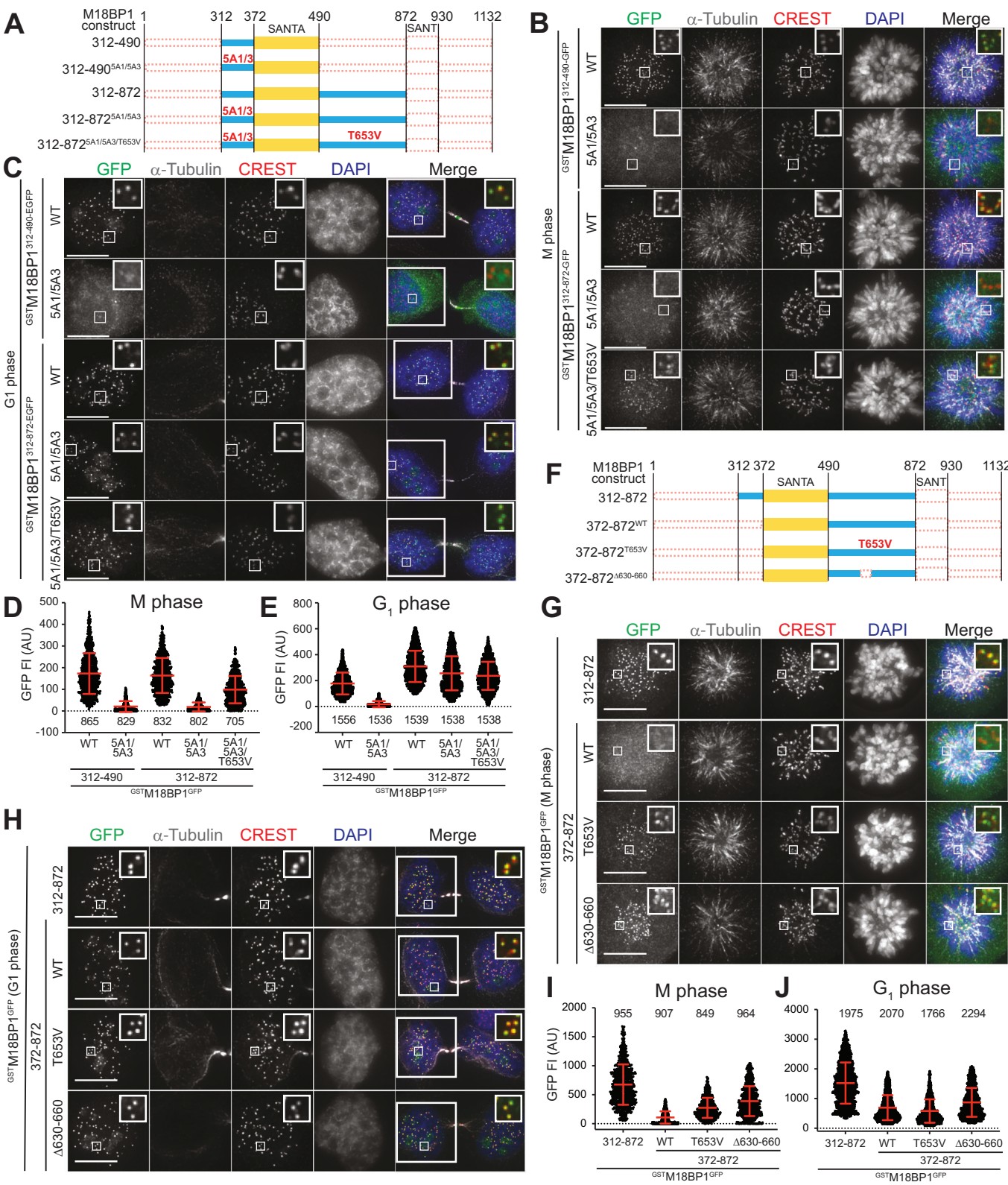

M18BP1 to a strong constitutive dimer, we mimicked dimerization occurring when dephosphorylation of M18BP1 in G1 phase allows its interaction with MIS18α/β. A requirement for dimerization for robust kinetochore recruitment strongly suggests that the kinetochore target of M18BP1 is multivalent (as discussed below). Ectopic dimerization is sufficient for robust centromere recruitment of M18BP1 throughout the cell cycle, including mitosis, when CDK phosphorylation prevents M18BP1 from binding MIS18α/β.

**Figure 6.  Thr653 is a regulatory key site of M18BP1 during mitosis.**

(A) Schematic showing the expressed $^{GST}$M18BP1$^{GFP}$ variants in (B, C). (B, C) Representative images of fixed HeLa cells showing GFP fluorescence of indicated stably expressed M18BP1 variants and immuno-stained α-tubulin in mitosis and G1, respectively. Centromeres were visualized by CREST sera, and DNA was stained by DAPI. White scale bars indicate 10 μm. (D, E) Quantification of the centromeric GFP-fluorescence intensities of the indicated expressed M18BP1 variants in mitosis and G1, respectively. Centromeres were detected in the CREST channel using a script for semiautomated quantification. Red bars indicate mean and standard deviation of all quantified centromere fluorescence intensities (n, as indicated below each plot) from three independent experiments. (F) Schematic showing the expressed $^{GST}$M18BP1$^{GFP}$ variants in (G, H). (G, H) Representative images of fixed HeLa cells showing GFP fluorescence of indicated stably expressed M18BP1 variants and immuno-stained α-tubulin in mitosis and G1, respectively. Centromeres were visualized by CREST sera, and DNA was stained by DAPI. White scale bars indicate 10 μm. (I, J) Quantification of the centromeric GFP-fluorescence intensities of the indicated expressed M18BP1 variants in mitosis and G1, respectively. Centromeres were detected in the CREST channel using a script for semiautomated quantification. Red bars indicate the mean and standard deviation of all quantified centromere fluorescence intensities (n, as indicated below each plot) from three independent experiments. Source data are available online for this figure.

We observed localization of M18BP1 in the absence of MIS18α/β, and also localization of M18BP1 constructs that do not bind MIS18α/β at all. Conversely, we did not observe localization of MIS18α/β under conditions that prevent its interaction with M18BP1. Thus, any contribution of the MIS18α/β to CALM localization (in addition to dimerizing M18BP1) may be dispensable, at least under the conditions of our assay, as well as insufficient for MIS18α/β's own centromere localization. In humans, the CENP-A-specific chaperone HJURP is recruited to MIS18α/β and interacts with CENP-C (Pan et al, 2019; Tachiwana et al, 2015; Thamkachy et al, 2024). Even HJURP recruitment, however, is exquisitely dependent on M18BP1, as it requires prior docking of PLK1 kinase onto two self-primed phosphorylation sites on M18BP1, Thr78 and Ser93 (Conti et al, 2024; Parashara et al, 2024). Thus, like MIS18α/β, also HJURP does not contain determinants sufficient for its kinetochore localization in the absence of M18BP1. In essence, our work identified M18BP1, after dimerization, as the main reader of epigenetic centromere specification in humans.

Using the ectopic dimerization and localization assay in cells depleted of endogenous M18BP1, we identified the essential role of the M18BP1 SANTA domain. The apparent discrepancy with previous work referred to above likely stems from the fact that the previous experiments had been carried out in the presence of endogenous M18BP1, whose dimerization with exogenous mutant M18BP1 on the MIS18α/β might have led to substantial residual recruitment of mutant constructs. This interpretation is supported by our experiments involving the overexpression of mutant M18BP1 lacking a functional SANTA domain. The SANTA domain was the only non-dispensable determinant of centromere localization of M18BP1 we have identified.

We determined the first experimental structure of the SANTA domain. This domain may be unique to M18BP1, as we could not identify close structural homologs in other proteins. A previous report identified CENP-C as a potential target of a construct consisting of pre-SANTA and the SANTA domain, and concluded CENP-C may be a direct target of the SANTA domain (French and Straight, 2019). We detected a very modest contribution of the SANTA domain to CENP-C binding, which was not affected by the mutation of W413 and H414, two residues required for cellular localization of M18BP1. Given the relatively small size of the SANTA domain and its convex binding interface, the target of the SANTA domain may be a motif embedded in an extended flexible segment of the polypeptide chain of a target protein. As this segment does not appear to be present in our biochemical

reconstitutions of the CCAN or of centromeric nucleosomes, its identity remains unknown. What could it be? Histone tails modified with centromere-specific modifications are potential candidates. However, enzymatic incorporation of two defining centromere-specific histone marks, Histone H4 Lys20 monomethylation or α-amino-trimethylation of the CENP-A N-terminus (H4-K20me1 or CENP-A-Nme3) (Bailey et al, 2016; Hori et al, 2014; Sathyan et al, 2017) did not increase the CENP-A-nucleosome-binding affinity for M18BP1 in vitro, nor was the binding sensitive to mutations of W413 and His414. KAT7, a histone acetyltransferase (HAT), binds M18BP1 and contributes to centromere stability (Ohzeki et al, 2016), and our future work will address the potential role of this enzyme in M18BP1 recruitment. While pre-nucleosomal CENP-A:H4 has also been shown to have specific post-translational modifications (Bailey et al, 2016; Sathyan et al, 2017; Shang et al, 2016), we doubt that they can function as guides for M18BP1, as the latter can be forced to localize in mitosis, where we expect no pre-nucleosomal CENP-A:H4 at centromeres (Jansen et al, 2007).

As the SANTA domain is required for kinetochore localization of M18BP1 both in interphase and in mitosis (upon forced dimerization), its elusive target exists at the centromere throughout the cell cycle. This target is a defining epigenetic marker of the centromere that remains elusive and whose identification is a clear goal of future studies. Even if essential, however, the SANTA domain is not sufficient for centromere localization and is complemented by adjacent pre- and post-SANTA disordered regions. Within these regions, we identify at least three distinct, biologically active centromere-targeting motifs, and a fourth motif is also active in vitro (Fig. 7G). Among the three biologically active motifs, two (Motifs 1 and 4) are putative CENP-C-binding motifs, while the other (Motif 3) binds at the interface of the CCAN subunits CENP-H and CENP-K but also interacts with nucleosomes in vitro, albeit non-specifically. The N-terminal region of CENP-C binds weakly to M18BP1 (Fig. 1H) and may also contribute to this interaction scheme, but here we did not characterize its role further.

Phosphorylation of residues near these motifs influences their ability to interact with the centromere during different phases of the cell cycle. In the pre-SANTA region, Motif 1–3 mediate interactions with CCAN, including phosphorylation-dependent interactions of Motif 1 with the CENP-C Cupin domain, facilitated by phosphorylation of the pre-SANTA region at residue Thr346, and of Motif 3 with CENP-HIKM. During interphase, both the pre-SANTA and the post-SANTA become dephosphorylated, allowing

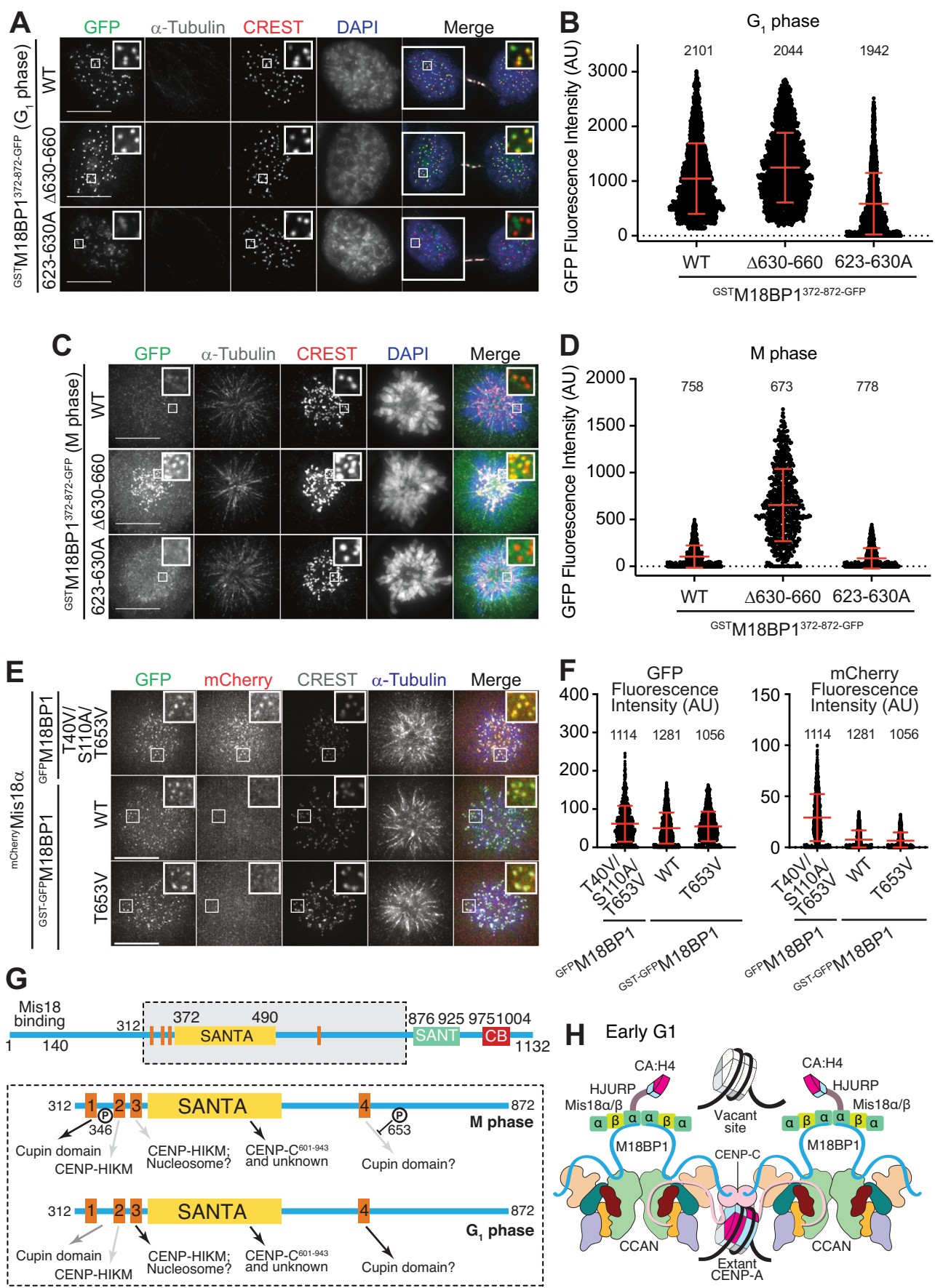

**Figure 7.  Crucial kinetochore-binding determinants in M18BP1 are sufficient for kinetochore localization upon dimerization.**

(A) Representative images of fixed HeLa cells showing GFP fluorescence of indicated stably expressed M18BP1 variants and immuno-stained α-tubulin in mitosis and G1. Centromeres were visualized by CREST sera, and DNA was stained by DAPI. White scale bars indicate 10 μm. (B) Quantification of the centromeric GFP-fluorescence intensities (*n*, as indicated below each plot) of the indicated expressed M18BP1 variants shown in (A). Centromeres were detected in the CREST channel using a script for semiautomated quantification. Red bars indicate the mean and standard deviation of all quantified centromere fluorescence intensities from three independent experiments. (C) Representative images of fixed HeLa cells showing GFP fluorescence of indicated stably expressed M18BP1 variants and immuno-stained α-tubulin in mitosis. Centromeres were visualized by CREST sera, and DNA was stained by DAPI. White scale bars indicate 10 μm. (D) Quantification of the centromeric GFP-fluorescence intensities (*n*, as indicated below each plot) of the indicated expressed M18BP1 variants shown in (C). Centromeres were detected in the CREST channel using a script for semiautomated quantification. Red bars indicate the mean and standard deviation of all quantified centromere fluorescence intensities from three independent experiments. (E) Representative images of fixed HeLa cells showing GFP fluorescence of indicated stably expressed M18BP1 variants, mCherry fluorescence of stably co-expressed Mis18α, and immuno-stained α-tubulin in mitosis. Centromeres were visualized by CREST sera. White scale bars indicate 10 μm. (F) Quantification of the centromeric GFP and mCherry fluorescence intensities of the indicated expressed M18BP1 variants and co-expressed Mis18α shown in (E). Centromeres were detected in the CREST channel using a script for semiautomated quantification. Red bars indicate the mean and standard deviation of all quantified centromere fluorescence intensities (*n*, as indicated below each plot) from three independent experiments. (G) Model of M18BP1 recruitment to the centromere in M- and G1 phase. The SANTA domain and four motifs mediate interactions with a variety of binding partners. The color of arrows reflects the strength of binding (from black/strong to light gray/weak). (H) Role of dimerization in the recruitment of M18BP1 to CCAN and possible stoichiometries of CENP-A deposition by HJURP bound to the Mis18 complex. Source data are available online for this figure.

the pre-SANTA and post-SANTA to switch roles as main drivers of kinetochore localization, as indicated by our observation that Motif 1 is dispensable for localization in interphase. The post-SANTA region in mammals may be functionally equivalent to the CENP-C-like binding motif in other organisms. Phosphorylation of Thr653 functionally limits this region during mitosis. In contrast, M18BP1 orthologs containing the CENP-C-like motif localize throughout the cell cycle in Xenopus, Caenorhabditis elegans, and chicken (Hori et al, 2017; Maddox et al, 2007; Moree et al, 2011; Perpelescu et al, 2015). While previous observations suggested that CENP-C may be dispensable for interphase localization of M18BP1 (French and Straight, 2019; French et al, 2017), our work in human cells clearly indicates that CENP-C is also required in interphase.

The cell-cycle dependence of kinetochore localization of M18BP1 is of special importance, not only in view of the role of this protein in CENP-A deposition, but also in view of the recently discovered function of M18BP1 in chromosome condensation through the loading of Condensin II, which is activated by CDK phosphorylation and engages a C-terminal region of M18BP1 apparently not involved in kinetochore recruitment (Borsellini et al, 2025; Wenda et al, 2021). As cyclin-dependent activity inhibits assembly of the M18BP1:Mis18 complex and CENP-A deposition, the observation that M18BP1 phosphorylation by CDK1 in the pre-SANTA region promotes binding to CENP-C is surprising. The functional relevance of this mitotic phosphorylation of the pre-SANTA localization module is unclear. It may promote a weak interaction with centromeres that facilitates deposition of Condensin II. Alternatively, or in addition, it may facilitate CENP-A deposition in the subsequent G1 phase. These hypotheses will need to be verified in future studies.

As potential limitations of this study, we would like to mention that in a subset of our experiments, we score "functionality" of mutant M18BP1 in terms of the ability to deposit CENP-A, assessed by fluorescence colocalization of newly deposited CENP-A with CREST. Even if this is a measure of functionality, we have no evidence that CENP-A is deposited at the correct place by these mutants. So, what we identify as being sufficient for CENP-A deposition may not imply that CENP-A is correctly deposited. It may be deposited, but incorrectly, possibly with long-term detrimental effects on centromere stability that we have not

monitored. We also note that multivalency is a common strategy in nature for increasing binding affinity to a multivalent target. Many of our results are based on an artificial dimerization strategy designed to mimic the dimerization of M18BP1 on the MIS18α/β complex. We cannot rule out that this strategy, by artificially increasing the binding affinity of M18BP1 for an intrinsically multivalent target like the centromere, with multiple copies of CENP-A and associated CCAN, introduced artifacts rather than truly mimicking a consequence of MIS18α/β binding. We mitigated this concern by re-introducing and validating many of our mutations or deletions into full-length, monomeric M18BP1.

To conclude, our observations delineate a defined hierarchy in kinetochore recruitment of the CALM, where M18BP1, after dimerization on MIS18α/β, is the only fully autonomous centromere targeting subunit of the human CENP-A loading machinery. Likely, the "epigenetic image" of M18BP1 is also a dimer built by CCAN and CENP-A nucleosomes in a specific reciprocal organization (Fig. 7H). While recent structural work illuminated the organization of single CCAN complexes and a neighboring nucleosome (Pesenti et al, 2022; Tian et al, 2022; Yatskevich et al, 2022), our new observations on the mechanism of M18BP1 recruitment strongly suggest the existence of a higher-order organization that acts as a multivalent target of the CALM. This higher-order organization, also suggested by super-resolution fluorescence microscopy (Andronov et al, 2019; preprint: Gupta et al, 2025), may consist of neighboring CCAN complexes, possibly in different states of assemblage, flanking one or more CENP-A nucleosomes. Ultimately, the crucial function of the loading machinery is to identify "temporary space" for stocking CENP-A between its deposition in G1 phase and its redistribution to the sister chromatids in S-phase. In this context, the local spatial organization of CCAN and of CENP-A nucleosomes at the target site is expected to be a crucial determinant of the outcome of the deposition reaction. Importantly, the MIS18α/β-M18BP1 octamer binds to a single HJURP molecule, which in turn binds a single CENP-A:H4 dimer (Barnhart et al, 2011; Pan et al, 2017; Thamkachy et al, 2024) (Fig. 7H). These stoichiometries may hold key significance for stable centromere inheritance, and their dissection is a key goal for future work.

# Methods

### Reagents and tools table

| Reagent/resource | Reference or source | Identifier or catalog number |
| --- | --- | --- |
| **Experimental models** | | |
| HeLa cells | IEO Milan | N/A |
| Flp-In T-REx HeLa | Tighe et al, 2008 | N/A |
| HeLa CENP-A-SNAP | Pan et al, 2017 | N/A |
| HeLa CENP-A-SNAP + EGFP-M18BP1$^{1-1132}$-P2A-T2A-mCherry-Mis18a | Pan et al, 2017 | N/A |
| HeLa CENP-A-SNAP + EGFP-M18BP1$^{1-1132\ 340-344A}$-P2A-T2A-mCherry-Mis18a | This study | N/A |
| HeLa CENP-A-SNAP + EGFP-M18BP1$^{1-1132\ 347-351A}$-P2A-T2A-mCherry-Mis18a | This study | N/A |
| HeLa CENP-A-SNAP + EGFP-M18BP1$^{1-1132\ 355-359A}$-P2A-T2A-mCherry-Mis18a | This study | N/A |
| HeLa CENP-A-SNAP + EGFP-M18BP1$^{1-1132\ 355-359A\ D931-1132}$-P2A-T2A-mCherry-Mis18a | This study | N/A |
| HeLa CENP-A-SNAP + EGFP-M18BP1$^{1-1132\ 355-359A\ D875-935}$-P2A-T2A-mCherry-Mis18a | This study | N/A |
| HeLa CENP-A-SNAP + EGFP-M18BP1$^{1-1132\ 355-359A\ D873-1132}$-P2A-T2A-mCherry-Mis18a | This study | N/A |
| HeLa CENP-A-SNAP + EGFP-M18BP1$^{1-1132\ 355-359A\ D491-930}$-P2A-T2A-mCherry-Mis18a | This study | N/A |
| HeLa CENP-A-SNAP + EGFP-M18BP1$^{1-1132\ 355-359A\ D491-872/D931-1132}$-P2A-T2A-mCherry-Mis18a | This study | N/A |
| HeLa CENP-A-SNAP + EGFP-M18BP1$^{1-1132\ 355-359A\ D491-872}$-P2A-T2A-mCherry-Mis18a | This study | N/A |
| HeLa CENP-A-SNAP + EGFP-M18BP1$^{1-1132\ 355-359A\ D630-660}$-P2A-T2A-mCherry-Mis18a | This study | N/A |
| HeLa CENP-A-SNAP + EGFP-M18BP1$^{1-1132\ 355-359A\ D710-740}$-P2A-T2A-mCherry-Mis18a | This study | N/A |
| HeLa CENP-A-SNAP + EGFP-M18BP1$^{1-1132\ 355-359A\ D630-660/\ D710-740}$-P2A-T2A-mCherry-Mis18a | This study | N/A |
| HeLa CENP-A-SNAP + EGFP-M18BP1$^{1-1132\ W413A/H414A}$-P2A-T2A-mCherry-Mis18a | This study | N/A |
| HeLa CENP-A-SNAP + EGFP-M18BP1$^{1-1132\ D372-490}$-P2A-T2A-mCherry-Mis18a | This study | N/A |
| HeLa CENP-A-SNAP + LAP-M18BP1$^{1-490}$ | This study | N/A |
| HeLa CENP-A-SNAP + LAP-M18BP1$^{1-490\ 340-344A}$ | This study | N/A |
| HeLa CENP-A-SNAP + LAP-M18BP1$^{1-490\ 347-351A}$ | This study | N/A |
| HeLa CENP-A-SNAP + LAP-M18BP1$^{1-490\ 355-359A}$ | This study | N/A |
| HeLa CENP-A-SNAP + LAP-M18BP1$^{1-490\ W413A/H414A}$ | This study | N/A |
| HeLa CENP-A-SNAP + GST-M18BP1$^{1-490}$-EGFP | This study | N/A |
| HeLa-GST-M18BP1$^{312-490}$-EGFP | This study | N/A |
| HeLa-GST-M18BP1$^{141-490}$-EGFP | This study | N/A |
| HeLa-GST-M18BP1$^{372-490}$-EGFP | This study | N/A |
| HeLa-GST-M18BP1$^{312-490\ 340-344A}$-EGFP | This study | N/A |
| HeLa-GST-M18BP1$^{312-490\ 347-351A}$-EGFP | This study | N/A |
| HeLa-GST-M18BP1$^{312-490\ 355-359A}$-EGFP | This study | N/A |
| HeLa-GST-M18BP1$^{312-490\ 340-344A,\ 355-359A}$-EGFP | This study | N/A |
| HeLa-GST-M18BP1$^{312-490\ W413A/H414A}$-EGFP | This study | N/A |
| HeLa-GST-M18BP1$^{312-872}$-EGFP | This study | N/A |
| HeLa-GST-M18BP1$^{312-872\ 340-344A,\ 355-359A}$-EGFP | This study | N/A |
| HeLa-GST-M18BP1$^{312-872\ 340-344A,\ 355-359A,\ T653V}$-EGFP | This study | N/A |
| HeLa-GST-M18BP1$^{372-872}$-EGFP | This study | N/A |
| HeLa-GST-M18BP1$^{372-872\ T653V}$-EGFP | This study | N/A |
| HeLa-GST-M18BP1$^{372-872\ D630-660}$-EGFP | This study | N/A |
| HeLa-GST-M18BP1$^{372-872\ 623-630A}$-EGFP | This study | N/A |
| HeLa-GST-M18BP1$^{491-872}$-EGFP | This study | N/A |
| HeLa-GST-M18BP1$^{491-872\ T653V}$-EGFP | This study | N/A |
| E.coli:BL21CodonPlus(DE3)-RIL strain | Agilent Technologies | Cat#230240 |
| E.coli:OmniMax | | |
| **Recombinant DNA** | | |
| pCDNA5/FRT/TO | Invitrogen | Cat#V601020 |
| pCDNA5/FRT/TO-EGFP-IRES | Krenn et al, 2012 | N/A |
| pOG44 | Gift from S. Taylor (Krenn et al, 2012) | N/A |
| pDuet-1 | Novagen | Cat#71146 |
| pEVOL-pBpF | Addgene #31190 | N/A |
| pcDNA5-EGFP-M18BP1(1-1132)-P2AT2A-mCherry-Mis18a | Pan et al, 2017 | |
| pcDNA5-EGFP-M18BP1(1-1132 340-344 A)-P2AT2A-mCherry-Mis18a | This study | |
| pcDNA5-EGFP-M18BP1(1-1132 347-351 A)-P2AT2A-mCherry-Mis18a | This study | |
| pcDNA5-EGFP-M18BP1(1-1132 355-359 A)-P2AT2A-mCherry-Mis18a | This study | |

| Reagent/resource | Reference or source | Identifier or catalog number |
|---|---|---|
| pcDNA5-EGFP-M18BP1(1-1132 W413A/H414A)-P2AT2A-mCherry-Mis18a | This study | |
| pcDNA5-EGFP-M18BP1(1-1132_ D372-490)-P2AT2A-mCherry-Mis18a | This study | |
| pcDNA5-EGFP-M18BP1(1-1132)-P2AT2A-mCherry-Mis18a | This study | |
| pcDNA5-LAP-M18BP1(1-490 340-344 A)-GFP-IRES2 | This study | |
| pcDNA5-LAP-M18BP1(1-490 347-351 A)-GFP-IRES2 | This study | |
| pcDNA5-LAP-M18BP1(1-490 355–359 A)-GFP-IRES2 | This study | |
| pcDNA5-LAP-M18BP1(1-490 W413A/H414A)-GFP-IRES2 | This study | |
| pcDNA5-LAP-M18BP1(1-490)-GFP-IRES2 | This study | |
| pcDNA5-GST-M18BP1(312-490)-GFP-IRES2 | This study | |
| pcDNA5-GST-M18BP1(312-490 340-344 A)-GFP-IRES2 | This study | |
| pcDNA5-GST-M18BP1(312-490 347-351 A)-GFP-IRES2 | This study | |
| pcDNA5-GST-M18BP1(312-490 355–359 A)-GFP-IRES2 | This study | |
| pcDNA5-GST-M18BP1(312-490 W413A/H414A)-GFP-IRES2 | This study | |
| pcDNA5-GST-M18BP1(312-490 340-344 A, 355–359 A)-GFP-IRES2 | This study | |
| pcDNA5-GST-M18BP1(1-490)-GFP-IRES2 | This study | |
| pcDNA5-GST-M18BP1(141-490)-GFP-IRES2 | This study | |
| pcDNA5-GST-M18BP1(372–490)-GFP-IRES2 | This study | |
| pcDNA5-GST-M18BP1(312-872)-GFP-IRES2 | This study | |
| pcDNA5-GST-M18BP1(312-872 340-344 A, 355–359 A)-GFP-IRES2 | This study | |
| pcDNA5-GST-M18BP1(312-872 340-344 A, 355–359 A, T653V)-GFP-IRES2 | This study | |
| pcDNA5-GST-M18BP1(372-872)-GFP-IRES2 | This study | |
| pcDNA5-GST-M18BP1(372-872 T653V)-GFP-IRES2 | This study | |
| pcDNA5-GST-M18BP1(372-872 D630-660)-GFP-IRES2 | This study | |
| pcDNA5-GST-M18BP1(372-872 623-630 A)-GFP-IRES2 | This study | |
| pcDNA5-GST-M18BP1(491-872)-GFP-IRES2 | This study | |
| pcDNA5-GST-M18BP1(491-872 T653V)-GFP-IRES2 | This study | |
| pETDuet-MBPT-SpyTag-CENP-C(775–943)-His | This study | |
| pETDuet-MBPT-SpyTag-CENP-C(601-943)-His | Walstein et al, 2021 | |
| pETDuet-MBPT-SpyTag-CENP-C(601-943 R742A/W751A)-His | Walstein et al, 2021 | |
| pETDuet-MBPT-SpyTag-CENP-C(601-943 T734V)-His | This study | |
| pETDuet-MBPT-SpyTag-CENP-C(601-943 S773A)-His | This study | |
| pETDuet-MBPT-SpyTag-CENP-C(601-943 T734V, S773A)-His | This study | |
| pETDuet-MBPT-SpyTag-CENP-C(601-943 V858E, K880A, L890R, F938A)-His | This study | |
| pGEXPT-M18BP1(312-490) | Conti et al, 2024 | |
| pGEXPT-M18BP1(312-490)-LPETGG-His | This study | |
| pGEXPT-M18BP1(312-490 340-344 A)-LPETGG-His | This study | |
| pGEXPT-M18BP1(312-490 347-351 A)-LPETGG-His | This study | |
| pGEXPT-M18BP1(312-490 355–359 A)-LPETGG-His | This study | |
| pGEXPT-M18BP1(312-490 W413A/H414A)-LPETGG-His | This study | |
| pGEXPT-M18BP1(312-490 T346V)-LPETGG-His | This study | |
| pGEXPT-M18BP1(312-490 T346V/S365A)-LPETGG-His | This study | |
| pGEXPT-M18BP1(312-490 T346E)-LPETGG-His | This study | |
| pGEXPT-M18BP1(312-490 T346E/S365D)-LPETGG-His | This study | |
| pET15b 6His NRMT | Gift from D.R. Foltz/I.G.Macara (Sathyan et al, 2017; Tooley et al, 2010) | |
| pGEXPT-M18BP1(312-490 412TAG)-LPETGG-His | This study | |
| pGEXPT-M18BP1(312-371)-LPETGG-His | This study | |
| pGEXPT-M18BP1(372–490)-LPETGG-His | This study | |
| **Antibodies** | | |
| Anti-GFP (1:3000 for western blotting) | Custom made, EIO antibody facility, Milan (Italy) | N/A |
| CREST sera (1:200 for immunofluorescence) | Antibodies Inc. | 15-234 |
| Anti-CENP-A (1:1000 for western blotting) | Cell Signalling | Cat#2186S |
| Anti-H4K20me1 (1:1000 for western blotting) | Abcam | Cat#ab9051 |

| Reagent/resource | Reference or source | Identifier or catalog number |
|---|---|---|
| Anti-M18BP1 (1:500 for immunofluorescence and western blotting) | Conti et al, 2024 | N/A |
| Anti-human IgG DyLight 405 (1:200 for immunofluorescence) | Jackson ImmunoResearch | 706-475-148 |
| Anti-human Rhodamine Red (1:200 for immunofluorescence) | Jackson ImmunoResearch | 706-295-148 |
| Anti-human Alexa Fluor 647 (1:200 for immunofluorescence) | Jackson Immuno Research | 109-603-003 |
| Anti-tubulin, mouse monoclonal (1:8000 for western blotting; 1:4000 for immunofluorescence) | Sigma | T9026 |
| Anti-vinculin, mouse monoclonal (1:10,000 for western blotting) | Sigma | V9131 |
| Anti-rabbit IgG Alexa 488 secondary (1:200 for immunofluorescence) | Invitrogen | A21206 |
| Anti-rabbit IgG Rhodamine Red secondary (1:200 for immunofluorescence) | Jackson Immuno Research | 711-295-152 |
| Anti-mouse IgG Rhodamine Red secondary (1:200 for immunofluorescence) | Jackson Immuno Research | 115-295-003 |
| Anti-rat IgG HRP (1:10,000 for western blotting) | Amersham | NXA935 |
| Anti-rabbit IgG HRP (1:10,000 for western blotting) | Cytiva (GE) | NA934 |
| Anti-mouse IgG HRP (1:10,000 for western blotting) | Amersham | NXA931-1ML |
| **Oligonucleotides and other sequence-based reagents** | | |
| siM18BP1 5'-GAAGUCUGGUGUUAGGAAAdTdT-3' | Pan et al, 2017 | N/A |
| **Chemicals, enzymes, and other reagents** | | |
| L(+)-Arabinose | Thermo Fisher | Cat#365181000 |
| Tween-20 (Polysorbat 20) | Thermo Fisher | Cat#9005-64-5 |
| STLC | Sigma | Cat#164739 |
| Bpa | Bachem | Product#4017646 |
| Nonidet-P40 Substitute | Fluka Chemie AG | Cat#9016-45-9 |
| ATP | | |
| Thymidine | Sigma | Cat#T9250 |
| Fetal Bovine Serum (FBS) | neoFroxx GmbH | |
| Q5 High-Fidelity 2X Master Mix | NEB | Cat#M0492L |
| T4 Polynucleotide Kinase | New England BioLabs | |
| T4 DNA Ligase | New England BioLabs | |
| Pfu Turbo Polymerase | Agilent Technologies | |
| DSBU ("BuUrBu") | Alinda Chemical Limited | N/A |
| SNAP-Cell Block | NEB | Cat#S9106S |
| SNAP-Cell 647-SiR | NEB | Cat#S9102S |
| Bovine serum albumin (BSA) | VWR | Cat#422351S |

| Reagent/resource | Reference or source | Identifier or catalog number |
|---|---|---|
| L-glutamine | PAN Biotech | Cat#P04-80100 |
| RNAiMAX | Invitrogen | Cat#13778100 |
| DAPI | Sigma | Cat#D9542 |
| Trypsin/EDTA | PAN Biotech | Cat#P10-027100 |
| Doxycycline | Sigma | Cat#D9891 |
| Triton X-100 | Sigma | Cat#T8787 |
| Paraformaldehyde | VWR | Cat#100503-914 |
| NEON electroporation kit (including buffer) | Invitrogen | Cat#MPK10096 |
| Penicillin-Streptomycin | PAN Biotech | Cat# P06-07100 |
| Blasticidin | Invitrogen | Cat#A1113902 |
| Hygromycin B | Invitrogen | Cat#10687010 |
| DMEM | PAN Biotech | Cat#P0403609 |
| OptiMEM | Life Technologies/ Gibco | Cat#31985-047 |
| X-tremeGENE | Roche | Cat#4476093001 |
| Poly-L-Lysine | Sigma | P4832-50ml |
| HEPES | Sigma | Cat#H3375 |
| MgCl$_2$ | Sigma | Cat#M8266 |
| CaCl$_2$ | Merck | Cat#2382 |
| Mowiol | Calbiochem | Cat#475904 |
| IPTG | Sigma | Cat#I6758 |
| NaCl | Sigma | Cat#S9888 |
| Glycerol | Sigma | Cat#G5516 |
| Imidazole | Sigma | Cat#I5513 |
| TCEP | Sigma | Cat#75259 |
| SAM | NEB | Cat#B9003S |
| Protease inhibitor cocktail | Serva | Cat#39107 |
| Coomassie Brilliant Blue R-250 dye | Thermo Fisher | Cat#20278 |
| TEV protease | Produced in-house | N/A |
| HRV3C protease | Produced in-house | N/A |
| CENP-OPQUR complex | Pesenti et al, 2018 | N/A |
| CENP-HIKM complex | Weir et al, 2016 | N/A |
| CENP-HI$^{Y488A}$K$^{S113R, G181R, E182K, E185K, D186R, E214K, E217K}$M complex | This study | N/A |
| CENP-LN complex | Pentakota et al, 2017 | N/A |
| CENP-TWSX complex | Walstein et al, 2021 | N/A |
| HisT-MBP-CENP-C$^{1-600}$-SpyCatcher | Walstein et al, 2021 | N/A |
| MBPT-SpyTag-CENP-C$^{C601-943}$-His | Walstein et al, 2021 | N/A |
| MBPT-SpyTag-CENP-C$^{601-943\ R742A/W751A}$-His | Walstein et al, 2021 | N/A |

| Reagent/resource | Reference or source | Identifier or catalog number |
|---|---|---|
| MBPT-SpyTag-CENP-C$^{601\text{-}943\ T734V}$-His | This study | N/A |
| MBPT-SpyTag-CENP-C$^{601\text{-}943\ S773A}$-His | This study | N/A |
| MBPT-SpyTag-CENP-C$^{601\text{-}943\ T734V/S773A}$-His | This study | N/A |
| MBPT-SpyTag-CENP-C$^{775\text{-}943}$-His | This study | N/A |
| MBPT-SpyTag-CENP-C$^{601\text{-}943\_V858E/K880A/L890R/F938A}$-His | This study | N/A |
| GST | This study | N/A |
| GSTPT-M18BP1$^{312\text{-}490}$-LPETGG-His | This study | N/A |
| GSTPT-M18BP1$^{312\text{-}490\ 340\text{-}344A}$-LPETGG-His | This study | N/A |
| GSTPT-M18BP1$^{312\text{-}490\ 347\text{-}351A}$-LPETGG-His | This study | N/A |
| GSTPT-M18BP1$^{312\text{-}490\ 355\text{-}359A}$-LPETGG-His | This study | N/A |
| GSTPT-M18BP1$^{312\text{-}490\ W413A/H414A}$-LPETGG-His | This study | N/A |
| GSTPT-M18BP1$^{312\text{-}490\ T346V}$-LPETGG-His | This study | N/A |
| GSTPT-M18BP1$^{312\text{-}490\ T346E}$-LPETGG-His | This study | N/A |
| GSTPT-M18BP1$^{312\text{-}490\ T346V/S365A}$-LPETGG-His | This study | N/A |
| GSTPT-M18BP1$^{312\text{-}490\ T346E/S365D}$-LPETGG-His | This study | N/A |
| GSTPT-M18BP1$^{312\text{-}490\ Y412Bpa}$-LPETGG-His | This study | N/A |
| GSTPT-M18BP1$^{312\text{-}371}$-LPETGG-His | This study | N/A |
| GSTPT-M18BP1$^{372\text{-}490}$-LPETGG-His | This study | N/A |
| CENP-A$^{NCP}$ | Weir et al, 2016 | N/A |
| H3$^{NCP}$ | Weir et al, 2016 | N/A |
| His-NRMT | This study | N/A |
| SETD8 | Active motif | Cat#31427, 31827 |
| CDK1:Cyclin-B:CKS1 | Huis In 't Veld et al, 2022 | N/A |
| **Software** | | |
| GraphPad Prism 10.0.3.(217) | GraphPad Software Inc. | https://www.graphpad.com |
| ImageJ Version 2.0.0-rc-69/1.52p | NIH | https://imagej.nih.gov/ij/ |
| softWoRx Version | GE Healthcare | |
| **Other** | | |

## Human cell lines

Parental Flp-In T-REx HeLa cells were a gift from S. Taylor (University of Manchester, Manchester, England, UK). Flp-In T-REx DLD-1–CENP-C–AID-EYFP cells were a gift from D. Fachinetti (Institut Curie, Paris, France) and D. C. Cleveland (University of California, San Diego, USA). These cells have both alleles of CENP-C tagged at the C terminus with an AID-EYFP fusion (Fachinetti et al, 2015). Furthermore, a gene encoding the plant E3 ubiquitin ligase osTIR1 was stably integrated into the genome of the cells. To induce rapid depletion of the endogenous AID-tagged CENP-C, 500 μM of the synthetic auxin Indole-3-acetic acid (IAA, Sigma-Aldrich) was added to the cells. In all cell culture experiments, cells were cultured in Dulbecco's modified Eagle's medium (DMEM; PAN-Biotech) supplemented with 10% tetracycline-free fetal bovine serum (neoFroxx GmbH), 100 U/ml Penicillin/100 μg/ml Streptomycin (PAN-Biotech), and 2 mM l-glutamine (PAN-Biotech) at 37 °C in a 5% $CO_2$ atmosphere. Expression of GFP-M18BP1 fusion proteins was induced by the addition of doxycycline (50 ng/ml; Sigma-Aldrich) for at least 24 h. In total, 10 μM STLC was used to arrest cells in prometaphase. For a complete list of the cell lines used in this study, see the Reagents and Tools Table.

## Bacterial strains

*E. coli* BL21CodonPlus(DE3)-RIL (Agilent Technologies, Santa Clara, CA, USA) strains were cultured on LB agar or liquid media at 37 °C supplemented with ampicillin (100 μg/mL) to maintain the pETDuet or pGEX plasmids and with chloramphenicol (34 μg/mL) to maintain the extra copies of tRNA Genes in the CodonPlus strain.

## Plasmids and cloning

pGEX6PT-M18BP1$^{312–490}$ (Conti et al, 2024) was modified to pGEX6PT-M18BP1$^{312–490}$-LPETGG-6His by inserting the PCR-amplified sequence of M18BP1$^{312–490}$-LPETGG-6His into the pGEX6PT backbone using the *Bam*HI and *Xho*I sites. The T346V/E, S365A/D, W413A, H414A, and point mutations were introduced by PCR-based site-directed mutagenesis. The 5-Alanine substitution mutants (residues 341–345 mutated to Ala, named 5A1; residues 347–351 mutated to Ala, named 5A2; and residues 355–359 mutated to Ala, named 5A3, were all generated by inverse PCR. To clone pGEX6PT-M18BP1$^{312–371}$-LPETGG-6His and pGEX6PT-M18BP1$^{372–490}$-LPETGG-6His, the respective PCR-amplified CDS of M18BP1 was inserted into the pGEX6PT-LPETGG-6His backbone using *Bam*HI and *Xho*I sites. pET-Duet-MBPT-SpyTag-CENP-C$^{601–943}$-8His (Walstein et al, 2021) was used as a template for the introduction of the CENP-C point mutations T734V, S773A, V858E, K880A, L890R, and F938A, which were introduced by site-directed mutagenesis. pcDNA5-EGFP-M18BP1$^{1–1132}$-P2AT2A-mCherry-Mis18α (Pan et al, 2017) was used as a template to introduce the W413A, H414A, T653V, 5A1, 5A2, and 5A3 mutations as described above. The truncation mutants M18BP1$^{1–872–5A3}$ and M18BP1$^{1–930–5A3}$ were PCR-amplified using pcDNA5-EGFP-M18BP1$^{1–1132–5A3}$-2AT2A-mCherry-Mis18α as template and inserted using the BamHI and XhoI sites. The deletion mutants M18BP15$^{Δ875–925–5A3}$, M18BP15$^{Δ491–930–5A3}$, M18BP15$^{Δ491–872–5A3}$, and M18BP15$^{Δ491–872–5A3}$ were generated by PCR-based site-directed mutagenesis using pcDNA5-EGFP-M18BP1$^{1–1132–5A3}$-P2AT2A-mCherry-Mis18α as template. pcDNA5-GST-M18BP1$^{312–490}$-GFP was generated by inserting the CDS of GST-PresC-TEV and GFP into pcDNA5 using Gibson Assembly. The CDS of M18BP1$^{312–490}$ was inserted using BamHI and XhoI sites. To generate pcDNA5-GST-M18BP1$^{312–872}$-GFP, the CDS of M18BP1$^{312–872}$ was inserted using BamHI and XhoI sites.

pcDNA5-GST-M18BP1$^{312–872}$-GFP was used as a template to introduce the deletion mutant M18BP1$^{312–872Δ630–660}$ and the alanine substitution mutant M18BP1$^{312–872\ 623–660A}$ using site-directed mutagenesis. To generate pcDNA5-LAP-M18BP1$^{1–490}$-IRES, PCR-amplified EGFP-TEV-S-tag CDS was inserted into linearized pcDNA5-M18BP1$^{1–490}$-IRES plasmid using Gibson assembly.

## Protein expression and purification

The *E. coli* expression plasmid pETDuet-EGFP-M18BP1$^{372–490}$-GNano-6His used for crystallization of the SANTA domain was used to transform BL21-CodonPlus(DE3)-RIL strain (Agilent Technologies, #230240). The transformed cells were cultured in TB medium to an OD$_{600}$ of 0.8, and the protein expression was induced by adding IPTG to a final concentration of 0.5 mM and further incubation at 20 °C for 16 h. Cells expressing EGFP-SANTA-GFPnanobody-6His were disrupted by Microfluidizer® (Microfluidics) in a lysis buffer containing 50 mM HEPES pH 7.5, 300 mM NaCl, 1 mM DTE, and 10 mM PMSF. The cleared lysate after centrifugation was applied to a Ni-affinity column, and the bound protein was eluted using buffer containing 50 mM HEPES pH 7.5, 300 mM NaCl, 400 mM imidazole, and 1 mM DTE. The eluate was applied to a Hiload 26/600 Superdex 200 pg SEC column (GE Healthcare) equilibrated with buffer containing 50 mM HEPES pH 7.5, 300 mM NaCl, and 1 mM DTE. The elution fractions containing monomeric EGFP-SANTA-GFPnanobody-6His were pooled, concentrated, and stored at −80 °C.

The expression plasmids pGEXPT-M18BP1$^{312–490}$-LPETGG-His (carrying the WT sequence or the 5A1, 5A2, 5A3, WH/AA, T346V, T346E, T346V/S365A, T346D/S365D mutations) were used to transform BL21-CodonPlus(DE3)-RIL strain. The transformed cells were cultured in TB medium to an OD$_{600}$ of 0.8, and the protein expression was induced by adding IPTG to a final concentration of 0.2 mM and further incubation at 20 °C for 16 h. Cells expressing GST-M18BP1$^{312–490}$-LPETGG-His variants were lysed by sonication in a lysis buffer containing 50 mM HEPES pH 7.5, 500 mM NaCl, 10% glycerol, 1 mM TCEP, and 1 mM PMSF. The cleared lysate after centrifugation was incubated with cOmplete His-Tag purification resin (Roche) for 16 h at 4 °C, and the bound protein was eluted using lysis buffer supplemented with 400 mM imidazole. The concentrated eluate was applied to a HiLoad 16/600 Superdex 200 pg SEC column (Cytiva) equilibrated with buffer containing 20 mM HEPES pH 7.5, 300 mM NaCl, 2.5% glycerol, and 1 mM TCEP. The elution fractions containing the GST-M18BP1$^{312-490}$-LPETGG-His variants were pooled, concentrated, and stored at −80 °C.

MBPT-SpyTag-CENP-C-601-943-His and HisTMBP-CENP-C1-600-SpyCatcher were expressed and purified as previously described (Walstein et al, 2021). MBPT-SpyTag-CENP-C-775–943-His and MBPT-SpyTag-CENP-C-601-943$^{T734V/S773A}$-His and MBPT-SpyTag-CENP-C601-943_V858E,K880A,L890R,F938A-His point mutants were expressed and purified as MBPT-SpyTag-CENP-C-601-943-His.

The CCAN subcomplexes CENP-HIKM, CENP-LN, CENP-OPQUR, and CENP-TWSX were expressed and purified as previously described (Pesenti et al, 2018; Pesenti et al, 2022; Walstein et al, 2021; Weir et al, 2016). The CENP-HI$^{Y488A}$K$^{S113R, G181R, E182K, E185K, D186R, E214K, E217K}$M mutant complex used in this study was expressed and purified exactly as the WT CENP-HIKM complex. The BPA-incorporated $^{GST}$M18BP1$^{312–490}$-LPETGG-His variant was

expressed in *E. coli* BL21(DE3) strain (Agilent Technologies, #200131). The pGEX plasmid encoding the M18BP1 gene with TAG codon at position 412 was used to transform *E. coli* cells together with pEVOL-pBpF. The cells were cultured in TB media supplemented with ampicillin, chloramphenicol, and 0.2% arabinose at 37 °C. Protein expression was induced by adding IPTG to a final concentration of 0.2 mM and the unnatural amino acid BPA at a final concentration of 1 mM when OD$_{600}$ of the culture reached 0.6. The culture was further incubated at 20 °C for 16 h. Purification of the GST-M18BP1$^{312–490\ 412Bpa}$-LPETGG-His variant was performed in the same way as the other GST-M18BP1$^{312–490}$-LPETGG-His variants described above.

## Sortase-mediated labeling of proteins with TMR-conjugated peptides

GST-M18BP1$^{312–490}$-LPETGG was labeled with GGGGK peptides with a C-terminally conjugated Tetramethylrhodamine (TMR, Genscript) fluorophore using purified Sortase 7 M mutant. Labeling was performed for ~16 h at 4 °C by incubation of 20 µM MBP-M18BP1$^{312–490}$-LPETGG with 200 µM GGGGK-TMR peptides and 1 µM Sortase 7 M in the reaction buffer containing 20 mM HEPES (pH 7.5), 300 mM NaCl, 2.5% glycerol, and 1 mM TCEP. 7 M Sortase and the excess of GGGGK-TMR peptides were removed by size-exclusion chromatography using a Superdex Increase 200 10/300 column.

## Analytical size-exclusion chromatography

Analytical size-exclusion chromatography of samples containing GST-M18BP1$^{312–490}$ variants, CENP-HIKM, MBP-CENP-C$^{601–943}$, and CENP-A$^{NCP}$ was performed on a Superdex 200 Increase 5/150 column (Cytiva) in SEC buffer containing 20 mM HEPES (pH 7.5), 300 mM NaCl, 2.5% glycerol, and 1 mM TCEP on an ÄKTAmicro system. The proteins and CENP-A$^{NCP}$ were incubated alone and in combination in a total volume of 40 µL SEC buffer. All proteins were diluted to 10 µM, whereas CENP-A$^{NCP}$ was diluted to 5 µM. The samples were eluted under isocratic conditions at 4 °C in SEC buffer at a flow rate of 0.15 ml/min. Elution of protein/NCP complexes was monitored at 280 nm and 254 nm wavelengths. 100 µL fractions were collected and analyzed by SDS-PAGE and Coomassie Brilliant Blue staining.

## In vitro protein phosphorylation

When indicated, samples were phosphorylated using in-house-generated CDK1:Cyclin-B:CKS1 (CCC). CCC was purified as described previously (Huis In 't Veld et al, 2022). Phosphorylation reactions were set up in SEC buffer [20 mM HEPES (pH 7.5), 300 mM NaCl, 2.5% glycerol, and 1 mM TCEP] containing 50 nM CCC, 10 µM CENP-C or M18BP1 substrate, 2 mM adenosine triphosphate, and 10 mM MgCl$_2$. Reaction mixtures were incubated at 30 °C for 30 min, alternatively at 10 °C for 16 h.

## Pull-down assays

Glutathione-resin and amylose bead pull-down assays were performed to check the binding of nucleosomes and kinetochore components to GST- or MBP-tagged M18BP1 truncation variants, respectively. The proteins were diluted with binding buffer [20 mM

HEPES (pH 7.5), 300 mM NaCl, 2.5% glycerol, 1 mM TCEP, and 0.01% Tween-20] to a final concentration of 5 μM in a total volume of 50 μl and mixed with 20 μl of Glutathione agarose (Cytiva/GE Healthcare) or amylose beads. After mixing the proteins and the beads, 20 μl were taken as input. The rest of the solution was incubated at 4 °C for 1 h on a thermomixer (Eppendorf) set to 1200 rpm. To separate the proteins bound to the Glutathione or amylose beads from the unbound proteins, the samples were centrifuged at 800 × g for 3 min at 4 °C. The supernatant was removed, and the beads were washed four times with 500 μl of washing buffer (same composition as binding buffer, but NaCl was lowered from 300 mM to 150 mM). After the last washing step, 20 μl of 2× SDS-PAGE sample loading buffer was added to the dry beads. The samples were boiled for 5 min at 95 °C and analyzed by SDS-PAGE and CBB (Coomassie Brilliant Blue) staining. GFP pull-down assays were performed as described above, using GFP-Trap magnetic agarose beads. To separate the proteins bound to the beads from the unbound proteins, the samples were placed on a magnetic rack, the unbound fraction was removed, and the beads were washed three times with 500 μl of washing buffer.

## EMSA assays

DNA or NCPs alone or DNA/M18BP1 or NCP/M18BP1 complexes pre-incubated in SEC buffer supplemented with 0.01% Tween-20 for 10 min on ice at indicated concentrations were run on a 5% acrylamide gel. EMSA loading buffer (40% sucrose, 0.025% bromophenol blue) was added 1:5 to the samples before transferring them to the wells. The native PAGE was run in the cold room at 100 V constant voltage for 60–90 min using 1× TB as running buffer. After the run, the DNA was stained using SYBR Gold (Thermo Fisher Scientific) according to the manufacturer's instructions, and subsequently the gel was scanned on a Chemidoc MP system (Bio-Rad).

## DSBU cross-linking mass spectrometry

Approximately 100 μg of GST-M18BP1$^{312-490}$/CENP-HIKM or GST-M18BP1$^{312-490}$/MBP-CENP-C$^{601-943}$ complexes in cross-linking buffer [50 mM HEPES (pH 7.5), 300 mM NaCl, and 1 mM TCEP] were incubated with 3 mM DSBU (Disuccinimidyl Dibutyric Urea) (diluted from 200 mM stock solution in dimethyl sulfoxide; Alinda Chemical Limited) at 25 °C for 1 h. The reaction was quenched by the addition of 100 mM Tris-HCl (pH 8) and incubated for 30 min. Diagnostic SDS-PAGE gels of samples before and after the cross-linking reaction were run to ensure the completion of the reaction. Samples were precipitated overnight with four volumes of cold acetone, spun down, and following the removal of the supernatant, the pellet was briefly air-dried before resuspension in 8 M urea, 1 mM DTT, and 5.5 mM chloroacetamide. LysC and trypsin were used to digest the samples at 25 °C for ~16 h, and the digestion was stopped by adding trifluoroacetic acid (TFA) to a final concentration of 0.2%. Samples were subjected to size-exclusion chromatography on a Superdex™ 30 Increase 3.2/300 (Cytiva). Peak fractions were pooled and evaporated in a SpeedVac.

## UV-induced crosslinking using Bpa

The Bpa-incorporated GST-M18BP1$^{312-490\_412Bpa}$ was diluted in 200 μL SEC buffer at 5 μM and was incubated with an equimolar amount of MBP-CENP-C-601-943. LED UV light with a wavelength of 365 nm (Nichia, NCSU276A) was used to irradiate the samples for 15 min on ice to activate the cross-linking reaction. The mixture was incubated with 50 μL GSH resin at 4 °C for 30 min to enrich the GST-tagged M18BP1 protein and the crosslinked adducts. The beads were washed four times with high salt washing buffer (containing 20 mM HEPES at pH 7.5, 500 mM NaCl and 1 mM TCEP). In total, 25 μL dry beads were then used for on-bead digest. Crosslinked protein samples on GSH resin were incubated with 8 M urea containing 1 mM DTT at 25 °C for 30 min. Chloroacetamide was added to the solution to a final concentration of 5.5 mM for alkylation. LysC and trypsin were used to digest the samples at 25 °C for ~16 h, and the digestion was stopped by adding trifluoroacetic acid (TFA) to a final concentration of 0.2%. Peptides were purified using Sep-Pak tC18 cartridges (50 mg, Waters), eluted in water containing 60% acetonitrile and 0.1% TFA, and dried completely in tubes.

## LC–MS/MS analysis

LC–MS/MS analysis of DSBU crosslinked peptides was performed as previously reported (Pan et al, 2018) with only minor modifications: peptides were dissolved in 10 μl water containing 0.1% TFA instead of 25 μl, of which 4–5 μl (instead of 5 μl) were injected for LC–MS/MS analysis, and data processing was performed with MeroX version 2.0.1.4 rather than version 1.6.6.6. All other steps followed the previously published protocol. LC–MS/MS analysis of Bpa crosslinked peptides was performed as previously reported (Pan et al, 2019) with the following modifications: Bpa crosslinked peptides were dissolved in 20 μl water containing 0.1% TFA, of which 0.5 μl were injected. The analytical column, trap column, emitter, eluents, and flow rate were identical to the original protocol. The LC gradient, however, was modified as follows: after an initial desalting step at 5% solvent B for 5 min, the percentage of solvent B was increased to 20% over 32 min, then to 32% over 7 min, followed by a ramp to 95% B within 1 min. This final concentration was held for 5 min before re-equilibrating the column. Data were acquired using the Q-Exactive HF mass spectrometer. MS parameters were the same as those used for the DSBU crosslinked samples, with the exception of the resolution settings: due to the different mass spectrometer, MS spectra were acquired at a resolution of 120,000 and MS/MS spectra at 15,000.

## Generation of stable HeLa cell lines

Stable Flp-In T-REx HeLa cell lines constitutively expressing various GFP-tagged M18BP1 constructs were generated by Flp/FRT recombination. Deletion mutants and point mutants of M18BP1 were generated by PCR site-directed mutagenesis, and the sequences of all constructs were verified by Sanger sequencing (Microsynth Seqlab). M18BP1 constructs were cloned into a pcDNA5/FRT plasmid and co-transfected with pOG44 (Invitrogen), a plasmid expressing the Flp recombinase, into cells using X-tremeGENE (Roche) according to the manufacturer's instructions. Following 2 weeks of selection in hygromycin B (250 μg/ml; Thermo Fisher Scientific) and blasticidin (4 μg/ml; Thermo Fisher Scientific), single-cell colonies were collected and subsequently expanded. Expression of the transgenes was checked by immuno-fluorescence microscopy and western blot analysis.

## RNAi transfection and CENP-A deposition experiment

Gene expression of endogenous M18BP1 was inhibited using a single small interfering RNA (siRNA) (sequence: 5'-GAAGUCUG-GUGUUAGGAAAdTdT-3', obtained from Thermo Fisher Scientific), which targets the coding region of M18BP1 mRNA. The expression of codon-optimized M18BP1 rescue constructs was not affected by the siRNA treatment. 30 nanomolars of M18BP1 siRNA was transfected using Lipofectamine RNAiMAX (Thermo Fisher Scientific) according to the manufacturer's instructions. To induce the expression of GFP-tagged M18BP1 rescue constructs, doxycycline (50 ng/ml) (Sigma-Aldrich) was added to the cells at the time of siRNA transfection. Thymidine was added at 1 mM final concentration to arrest cells at G1/S transition. When cells were released from thymidine block, existing CENP-A-SNAP proteins were blocked using 10 μM SNAP-Cell Block (NEB) according to the manufacturer's protocol. 10 μM STLC was used to arrest cells in prometaphase. The cells arrested in prometaphase were separated from other cells by mitotic-shake-off, released from STLC by three consecutive washing steps with 1 mL DMEM, and placed in wells of 24-well plates containing poly-lysine-coated coverslips. Three hours later, the cells in early G1 phase attached to the coverslips and were treated with 3 μM SNAP-Cell 647-SiR (NEB) to label newly synthesized SNAP-CENP-A according to the manufacturer's protocol.

## Protein electroporation

To electroporate recombinant TMR-labeled GST-M18BP1$^{312-490}$ variants into either DLD-1–CENP-C–AID-EYFP cells or into HeLa cells, the Neon Transfection System Kit (Thermo Fisher Scientific) was used. Cells ($3 \times 10^6$) were trypsinized, washed with PBS, and resuspended in electroporation buffer R (Thermo Fisher Scientific) to a final volume of 90 μl. The recombinant protein was diluted 1:2 in buffer R to 50 μM, and 30 μl of the mixture was added to the 90-μl cell suspension. After mixing the sample, 100 μl of the mixture was loaded into a Neon pipette tip (Thermo Fisher Scientific) and electroporated by applying one 20 ms pulse with an amplitude of 1400 V. The sample was subsequently added to 15 ml of prewarmed PBS, centrifuged at 500 g for 3 min, and trypsinized for 7 min to remove noninternalized extracellular protein. After one additional PBS washing step and centrifugation, the cell pellet was resuspended in DMEM and transferred to a 12-well plate containing poly-l-Lysine–coated coverslips. After the electroporation procedure, DLD-1–CENP-C$^{AID-EYFP}$ cells were additionally treated with 500 μM IAA (Sigma-Aldrich) to induce rapid depletion of endogenous CENP-C.

## Immunofluorescence microscopy

Cells grown on poly-L-Lysine (0.0001%, Sigma-Aldrich) coated coverslips were fixed for 10 min at room temperature using a 4% paraformaldehyde (PFA) solution diluted in phosphate-buffer saline (PBS, prepared in-house) followed by three washing steps with PPS. The PFA–fixed cells were permeabilized with PBS-T [PBS buffer containing 0.1% Triton X-100] for 10 min and incubated with PBS-T containing 4% bovine serum albumin (BSA) for 40 min. Cells were incubated for 90 min at room temperature with CREST/anticentromere sera (Antibodies Inc.; dilution 1:200 in 2%

BSA in PBS-T), washed three times with PBS-T, and subsequently treated for 30 min with anti-human Alexa Fluor 647-conjugated secondary antibody (Jackson ImmunoResearch; dilution 1:200 in 2% BSA in PBS-T). In some experiments, a-Tubulin (Sigma #T9026; dilution 1:4000 in 2% BSA in PBS-T) or M18BP1 (generated in-house (Conti et al, 2024); dilution 1:500 in 2% BSA in PBS-T) was additionally immuno-stained. To visualize DNA, 4′,6-diamidino-2-phenylindole (DAPI) (0.5 μg/ml; Serva) was added during the last washing step for 3 min. After drying, the coverslips were mounted with Mowiol mounting media (EMD Millipore) on glass slides.

## Imaging and quantification of centromere fluorescence intensities

Fixed-cell samples were imaged at room temperature using a ×60 oil immersion objective lens on a DeltaVision deconvolution microscope. The DeltaVision Elite System (GE Healthcare, UK) is equipped with an IX71 inverted microscope (Olympus, Japan), a PLAPON ×60/1.42 numerical aperture objective (Olympus) and a pco.edge sCMOS camera (PCO-TECH Inc., USA). Images were acquired as 16 z-sections of 0.2 μM. The z-sections were converted into maximal intensity projections or average intensity projections, converted into 16-bit TIFF files, and exported for further analysis. Quantification of centromere signals was performed using the software Fiji with a script for semiautomated processing as previously described (Pan et al, 2019; Walstein et al, 2021). Briefly, centromere spots were chosen based on the parameters of shape, size, and intensity using the images of the reference channel obtained with CREST staining, and their positions were recorded. In the images of the data channels, the mean intensity value of adjacent pixels of a centromere spot was subtracted as background intensity from the mean intensity value of the centromere spot. Intensity values were exported to Excel (Microsoft), the top and lowest 10% of data points were removed, and the resulting values were plotted using GraphPad Prism software 9.0 (GraphPad Software). For quantifying chromatin GFP signals, DAPI-stained DNA was used as a reference defining the chromatin area, and GFP intensity values in these regions were recorded.

## Immunoblotting

HeLa cells were harvested by trypsinization, and the cell pellet was washed once with PBS. Cells were incubated in lysis buffer [75 mM HEPES (pH 7.5), 150 mM NaCl, 10 mM MgCl$_2$, 10% glycerol, 0.1% NP-40, Benzonase (90 U/ml; Sigma-Aldrich), and protease inhibitor mix HP Plus (Serva)] for 30 min on ice. The lysed cells were centrifuged at $16,000 \times g$ for 30 min at 4 °C, and SDS sample buffer was added to the supernatant. After Tricine–SDS-PAGE, the proteins were blotted on a nitrocellulose membrane using the Trans-Blot Turbo Transfer System (Bio-Rad) and were subsequently detected by Western blot analysis. The following primary antibodies were used: anti-α-Tubulin (1:8000; Sigma-Aldrich, T9026), anti-GFP (1:1000; generated in-house), anti-M18BP1 (1:500; generated in-house), and anti-Vinculin (1:10,000; Sigma-Aldrich V9131). As secondary antibodies, we used anti-mouse (1:10,000; Amersham NXA931), anti-rabbit (1:10,000 or 1:2000 (for anti-M18BP1 primary antibody); Cytiva(GE) NA934V) or anti-rat (1:10,000; Amersham NXA935) conjugated to horseradish

peroxidase. After incubation with ECL Western blotting reagent (GE Healthcare), images were acquired with the ChemiDoc MP System (Bio-Rad) using Image Lab 5.1 software.

## Crystallization of the SANTA domain

The human M18BP1 SANTA domain (residues 372–490) was crystallized as the fusion protein construct EGFP-**AS**-SANTA-**LEGT**-anti-GFP Nanobody-GGHHHHHH (linker residues are shown in bold). The fusion protein was concentrated to 32 mg/ml in buffer containing 50 mM HEPES pH 7.5, 300 mM NaCl and 1 mM DTE. Crystals were grown by sitting-drop vapor-diffusion by mixing 1 μl protein solution with 1 μl reservoir solution containing 0.1 M MES pH 6.0, 50% PEG400, and 5%w/v PEG3000 (optimized from JCSG core III screen (Qiagen, Hilden, Germany), condition F1) and incubation at 20 °C for 5–10 days. Crystals were fished directly from the sitting drops and flash-frozen without the addition of cryoprotectant.

## Structure determination

Data were collected at 100 K using a Pilatus 6MF detector at the P11 beamline at the PETRA synchrotron in Hamburg, Germany, and integrated and scaled using XDS and XSCALE (Kabsch, 2010). The structure was solved in space group C2 with 2 molecules per asymmetric unit via molecular replacement with PHASER (Collaborative Computational Project, 1994) using the crystal structure of GFP:GFPnanobody complex (PDB code 3OGO) as a template. Refinement with PHENIX (Adams et al, 2010) resulted in a model (Emsley et al, 2010) with good Ramachandran geometry and $R_{work}/R_{free}$ values (Appendix Table S1). The C-terminal residues GGHis$_6$ were missing in the electron density for both molecules, as well as residues MV at the N-terminus of the second molecule. In addition, the "linker" between EGFP and the SANTA domain—encompassing the last 9 residues of EGFP and the first 7 (monomer B) or 8 (monomer A) residues of the SANTA domain) was disordered in both molecules. In the last rounds of refinement, the NCS constraints were removed, but apart from a relative rotation of the SANTA domains in monomers A and B by ~15° (relative to the EGFP/nanobody domains), no major conformational differences were apparent (overall main chain r.m.s.d. difference 1.68 Å for 445 $C_\alpha$ atoms, and 0.78 Å for 111 residues of the SANTA domains). The SANTA domain of monomer B was slightly less well defined compared to monomer A as indicated by increased B factors. The coordinates have been submitted to the protein data bank (PDB) with identifier 9R6H.

## AlphaFold predictions

AF predictions were performed using AlphaFold2 (Jumper et al, 2021) (version 2.3.1) or AlphaFold3 (Abramson et al, 2024) (version 3.0.0, checked out from GitHub (https://github.com/google-deepmind/alphafold3 on January 12, 2025). For AlphaFold2, 10 models were predicted for the target protein complex comprising the CENP-C cupin domain (residues 820–943, comprising the most rigid part of the Cupin domain) and M18BP1 (residues 325–265). The predictions were generated using the multimer pipeline to account for protein-protein interactions. For AlphaFold3, a total of 25 models were predicted using five

different random seeds, with each seed generating five models, to enhance conformational sampling. This approach ensured diverse structural outputs for the same target complex. The models were evaluated based on two criteria: the overall Template Modeling Score (TM-score) and the local Distance Difference Test (LDDT) scores for the interface residues of the M18BP1 or HJURP peptide interacting with the CENP-C Cupin domain (residues 820–943). LDDT scores were computed for specific interface residues to evaluate the accuracy of local interactions critical to the protein-ligand interface. For each method, the model with the highest overall TM-score was selected as the primary model. In cases where multiple models had comparable TM-scores, the model with the highest LDDT values for the interface residues was prioritized to ensure optimal representation of the interaction interface.

## Quantification and statistical analysis

Quantification of fluorescence intensities was performed as indicated above. Statistical analysis was performed with a rank-sum, nonparametric test comparing two unpaired groups (Mann–Whitney test) in GraphPad Prism. Symbols indicate: n.s. $P > 0.05$, $*P \leq 0.05$, $**P \leq 0.01$, $***P \leq 0.001$, $****P \leq 0.0001$.

# Data availability

Atomic coordinates of the M18BP1 SANTA domain have been deposited to the Protein Data Bank PDB (https://www.rcsb.org/) with identifier 9R6H. The mass spectrometry datasets have been deposited to the MassIVE database (https://massive.ucsd.edu/) with the following identifiers: ID635 (CENPHIKM datasets (Fig. EV2A,B)): MassIVE MSV000099772. PXD070401 https://massive.ucsd.edu/ProteoSAFe/dataset.jsp?task=7cd1fe12a5184afe85324bd6cab830c1. ID999 (CENP-C DSBU dataset (Appendix Fig. S5B)): MassIVE MSV000099773. PXD070404 https://massive.ucsd.edu/ProteoSAFe/dataset.jsp?task=f9964ad0e44e4bffa4e041ce9ce88276. ID2156 (CENP-C BPA dataset (Appendix Fig. S5C)): MassIVE MSV000099775. PXD070405 https://massive.ucsd.edu/ProteoSAFe/dataset.jsp?task=4023ad4c5a924c299e1355afe29744ba.

The source data of this paper are collected in the following database record: biostudies:S-SCDT-10_1038-S44318-026-00698-z.

# Peer review information

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

## Acknowledgements

We thank the beamline staff of the beamline P11 at the PETRA synchrotron, Hamburg, Germany, for support, and our colleague R Gasper-Schönenbrücher for help with data collection. We are grateful to D Fachinetti DC Cleveland, and D Foltz for sharing reagents; to T-C Li and D Summerer for help setting up cell sorting experiments; to I Hoffmann, C Koerner, B Voss, S Wohlgemuth, T Crocilla, M Nasimi, M Richter, and M Terbeck for general technical assistance and for preparation of CCAN subunits and protein kinases; to Duccio Conti for help with cell line preparation; and to F Müller and M Metz for help with mass spectrometry experiments and data analysis. AM acknowledges funding by the Max Planck Society, the European Research Council (ERC) Synergy Grant 951430 (BIOMECANET), the DGF's Collaborative Research Centre 1430 "Molecular Mechanisms of Cell State Transitions". This work was supported by the European Molecular Biology Organization (EMBO Postdoctoral Fellowship ALFT106-2024 to LH).

## Author contributions

**Kai Walstein**: Conceptualization; Investigation; Visualization; Writing—original draft; Project administration; Writing—review and editing. **Louisa Hill**: Conceptualization; Investigation; Visualization; Writing—original draft; Project administration; Writing—review and editing. **Doro Vogt**: Investigation. **Lina Oberste-Lehn**: Investigation; Writing—review and editing. **Petra Janning**: Data curation; Investigation. **Ingrid R Vetter**: Conceptualization; Investigation; Visualization. **Dongqing Pan**: Conceptualization; Investigation; Visualization. **Andrea Musacchio**: Conceptualization; Supervision; Funding acquisition; Visualization; Writing—original draft; Project administration; Writing—review and editing.

Source data underlying figure panels in this paper may have individual authorship assigned. Where available, figure panel/source data authorship is listed in the following database record: biostudies:S-SCDT-10_1038-S44318-026-00698-z.

## Funding

## Disclosure and competing interests statement

The authors declare no competing interests.

# Expanded View Figures

**Figure EV1.  GST-dimerized M18BP1$^{312-490}$ robustly localizes to the kinetochore.**                                    ▶

(**A**) SDS-PAGE of $^{MBP}$M18BP1$^{312-490}$ or $^{GST}$M18BP1$^{312-490}$ with their predicted and observed molecular weights. (**B**, **C**) MALDI mass spectrometry analyses of the proteins shown in (**A**) with the measured molecular masses. (**D**, **E**) Mass photometry analysis of the same samples, with molecular masses indicated above peaks. (**F**) Comparison of chromatin and kinetochore levels of $^{GFP}$M18BP1$^{1-490}$ or $^{GST}$M18BP1$^{1-490-GFP}$ after inducible expression from stable cell lines. Centromeres were visualized by CREST sera, α-tubulin was immuno-stained and DNA was stained by DAPI. White scale bars indicate 10 μm. (**G**) GFP fluorescence intensity levels were classified as "low" or "high" based on chromatin localization. Red bars indicate mean and standard deviation of all quantified centromere fluorescence intensities from three independent experiments. (**H**) Corresponding kinetochore GFP fluorescence intensity. Red bars indicate mean and standard deviation of all quantified centromere fluorescence intensities from three independent experiments. (**I**) Band intensity of CENP-C$^{601-943}$ bound to $^{GST}$M18BP1$^{312-490}$ after CCC phosphorylation, normalized to bound CENP-C$^{601-943}$ in absence of CCC. Mean and standard deviation from three independent experiments are reported. (**J**) SDS-PAGE of pulldown experiments where $^{MBP}$CENP-C$^{601-943-GFP}$ used as bait (with GFP as negative control) was incubated with $^{MBP}$M18BP1$^{312-490}$ or $^{GST}$M18BP1$^{312-490}$ (treated with CCC). Source data are available online for this figure.

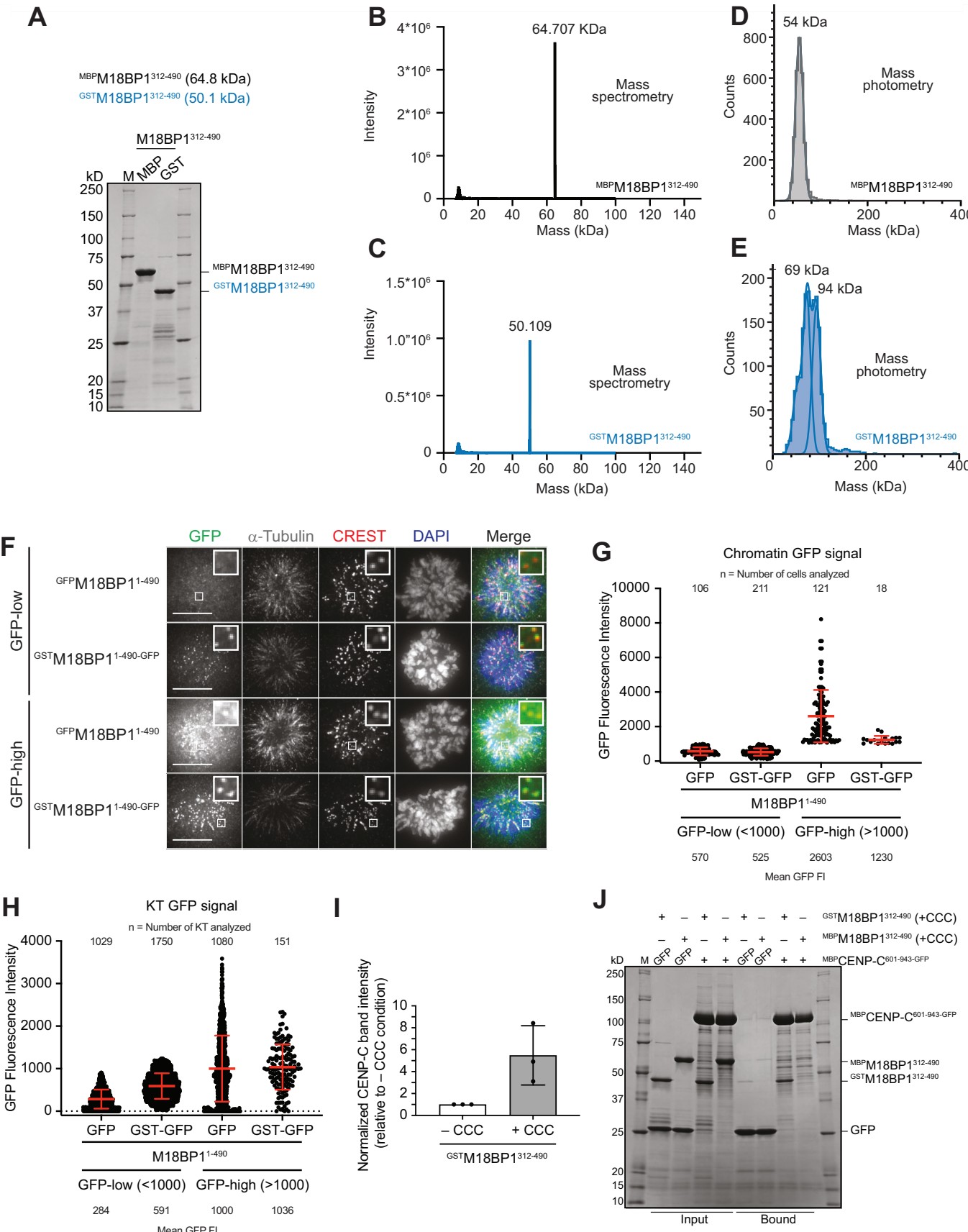

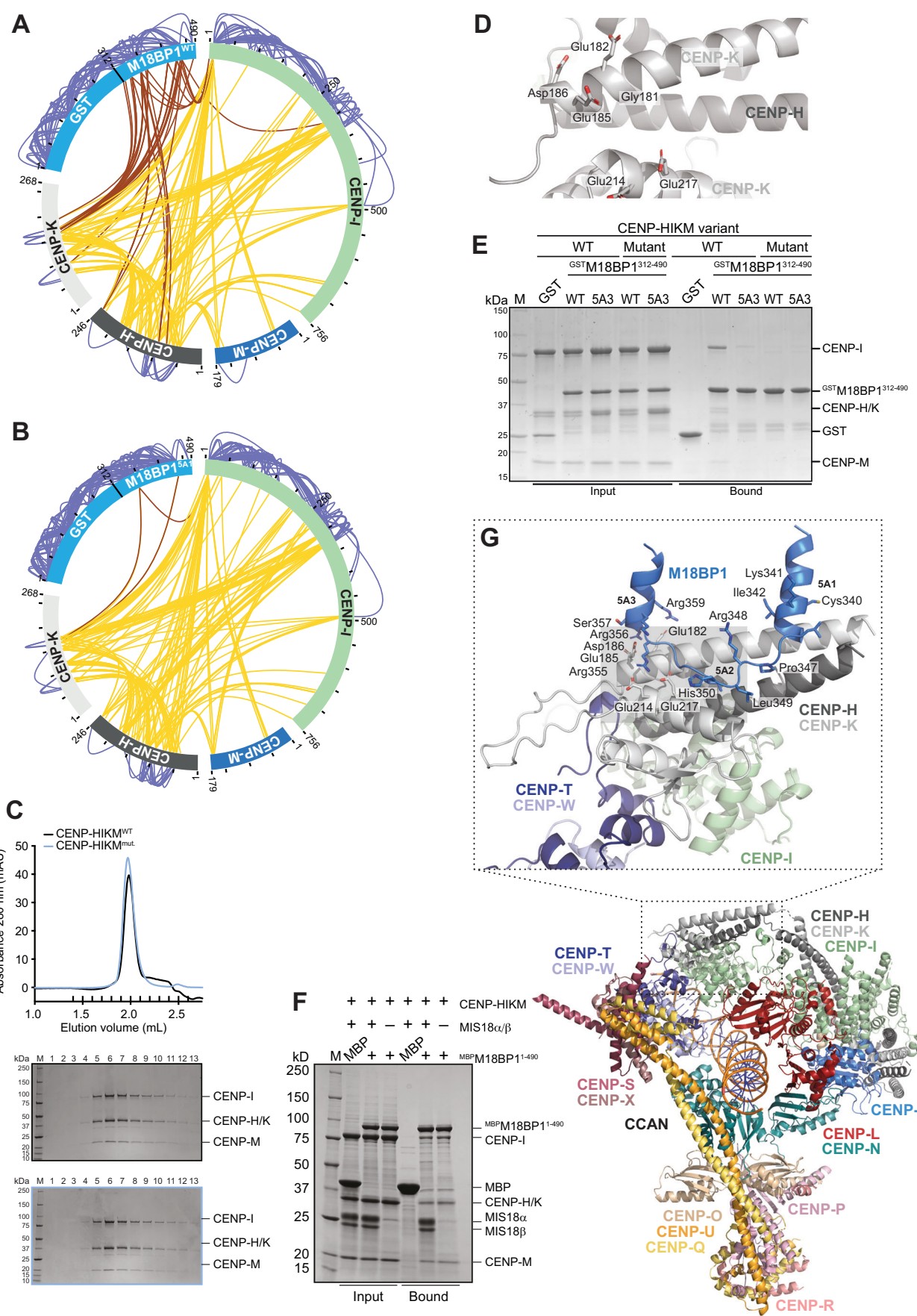

◀ **Figure EV2.　The interaction between M18BP1 and CENP-HIKM.**

(A, B) Schematic showing the results of DSBU crosslinking experiments followed by mass spectrometry analysis. In (A) $^{GST}$M18BP1$^{312–490\ WT}$ was incubated with CENP-HIKM complex. In (B) $^{GST}$M18BP1$^{312–490\ 5A3}$ was incubated with CENP-HIKM complex. Intermolecular crosslinks are highlighted in red, intramolecular crosslinks are highlighted in yellow. (C) Elution profiles from a size-exclusion chromatography column demonstrates wild-type and mutant CENP-HIKM elute identically and in a single sharp peak, indicative of comparable stability. (D) A CENP-HIKM mutant was generated on the basis of crosslinking experiments to potentially disrupt the interaction with M18BP1 (CENP-I$^{Y488A}$, CENP-H$^{WT}$, CENP-K$^{S113R/G181R/E182K/E185K/D186R/E214K/E217K}$, CENP-M$^{WT}$). The position of mutations on CENP-K later shown to be at the M18BP1 interface (see **G**) are highlighted. (E) GSH-resin pull-down assay with GST (negative control) and $^{GST}$M18BP1$^{312–490}$ (WT and 5A3 mutant) as baits testing the binding to WT and mutant CENP-HIKM complexes. The M18BP1 variants used as baits and the CENP-HIKM variants used as preys are indicated above each lane. (F) Amylose bead pull-down assay showing binding of CENP-HIKM to immobilized $^{MBP}$M18BP1$^{1–490}$ in presence or absence of Mis18 complex to evaluate its potential interference. (G) AlphaFold3 prediction of the interaction of M18BP1 residues 300-400 with the CENP-HIKM complex is shown in the context of the full CCAN. Prediction quality indicators are shown in Appendix Fig. S3A. The area with a gray background is the one highlighted in (D) and where mutated CENP-K residues lie. Source data are available online for this figure.

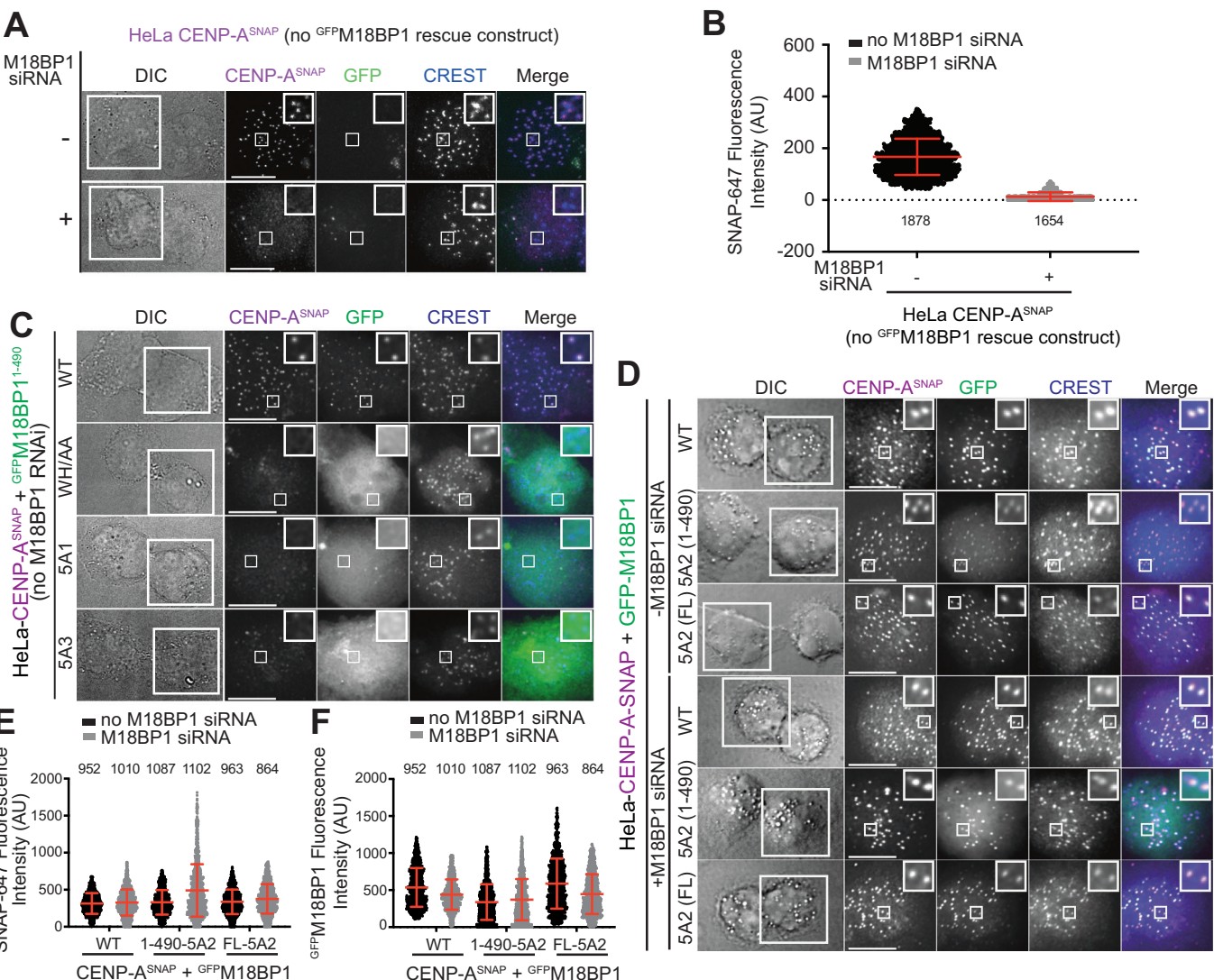

**Figure EV3. M18BP1^(1-490) does not tolerate 5A1, 5A3 and WH/AA mutations.**

(A) Representative images of fixed HeLa cells showing CENP-A^SNAP fluorescence labeled with SNAP-Cell 647-SiR and GFP autofluorescence in the absence of a ^GFP M18BP1 rescue construct in early G1 in the absence or presence of M18BP1 siRNA treatment. Centromeres were visualized by CREST sera. The G1 couple is shown in the differential interference contrast (DIC) channel. White scale bars indicate 10 μm. (B) Quantification of the centromeric CENP-A-SNAP-fluorescence of a HeLa cell line that does not express any ^GFP M18BP1 construct in the absence or presence of M18BP1 siRNA treatment. Centromeres were detected in the CREST channel using a script for semiautomated quantification. Data shown as mean with SD of all quantified centromeres from three independent experiments. (C) Representative images of fixed HeLa cells showing CENP-A^SNAP fluorescence labeled with SNAP-Cell 647-SiR and GFP fluorescence of indicated stably expressed M18BP1 variants in early G1. Centromeres were visualized by CREST sera. The G1 couple is shown in the differential interference contrast (DIC) channel. White scale bars indicate 10 μm. (D) Representative images of fixed HeLa cells showing CENP-A^SNAP fluorescence labeled with SNAP-Cell 647-SiR and GFP fluorescence of indicated stably expressed M18BP1 variants in early G1 in the absence or presence of M18BP1 siRNA treatment. Centromeres were visualized by CREST sera. The G1 couple is shown in the differential interference contrast (DIC) channel. White scale bars indicate 10 μm. The same control micrographs are shown in Figs. EV3D and EV4B,F, since all data were obtained from the same experiment. (E, F) Quantification of the centromeric CENP-A-SNAP-fluorescence and centromeric GFP-fluorescence intensities, respectively, of HeLa cell lines stably expressing the indicated M18BP1 variants in the absence or presence of M18BP1 siRNA treatment. Centromeres were detected in the CREST channel using a script for semiautomated quantification. Data shown as mean with SD of all quantified centromeres from three independent experiments. Source data are available online for this figure.

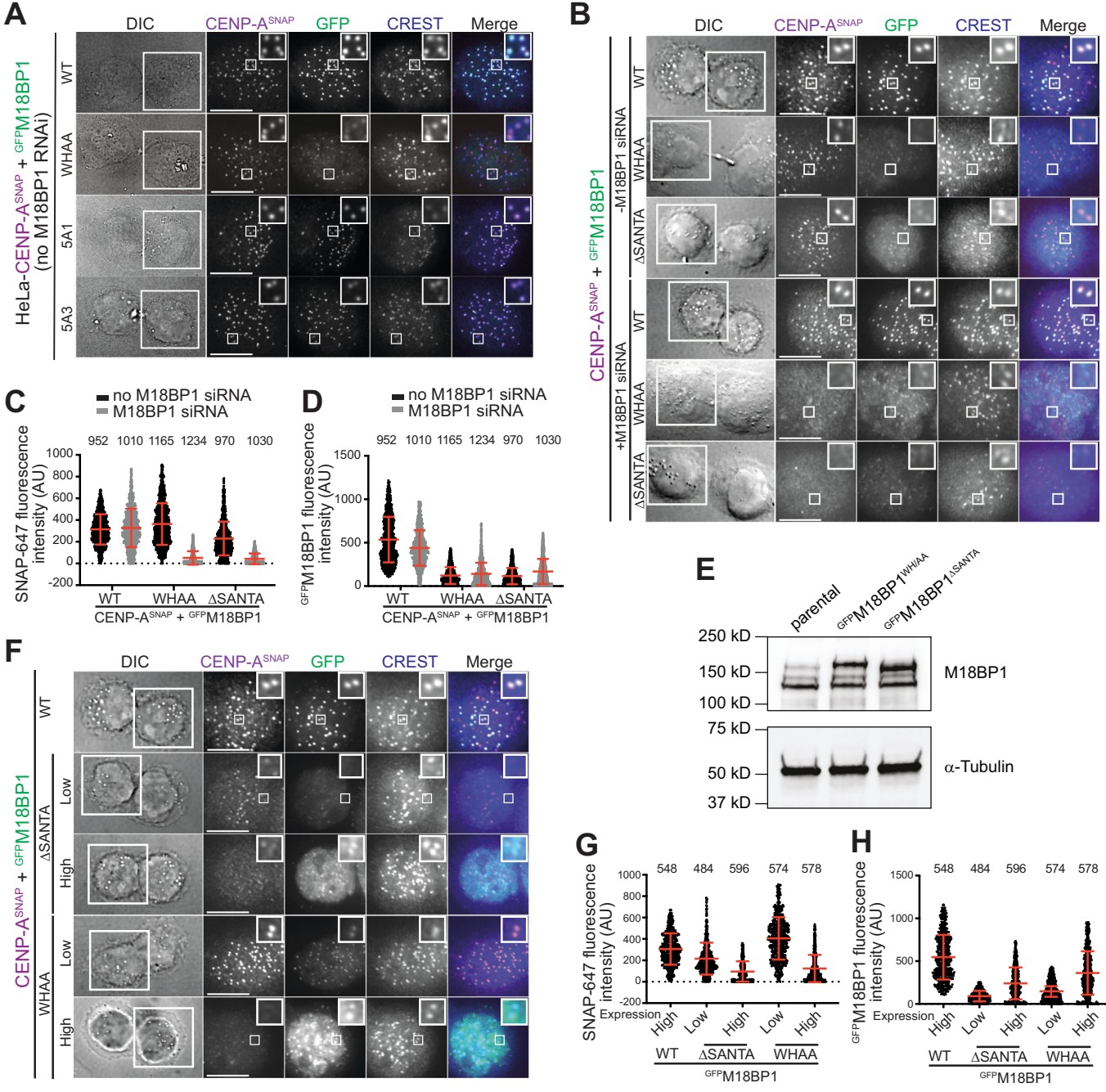

◀ **Figure EV4.   Additional CENP-A Loading experiments.**

(A) Representative images of fixed HeLa cells showing CENP-A^SNAP fluorescence labeled with SNAP-Cell 647-SiR and GFP fluorescence of indicated stably expressed M18BP1 variants in early G1. Centromeres were visualized by CREST sera. The G1 couple is shown in the differential interference contrast (DIC) channel. White scale bars indicate 10 μm. (B) Representative images of fixed HeLa cells showing CENP-A^SNAP fluorescence labeled with SNAP-Cell 647-SiR and GFP fluorescence of indicated stably expressed M18BP1 variants in early G1 in the absence and presence of M18BP1 siRNA treatment. Centromeres were visualized by CREST sera. The G1 couple is shown in the differential interference contrast (DIC) channel. White scale bars indicate 10 μm. The same control micrographs are shown in Figs. EV3D and EV4B,F, since all data were obtained from the same experiment. (C, D) Quantification of the centromeric CENP-A-SNAP and GFP fluorescence intensities, respectively, of HeLa cell lines stably expressing the indicated M18BP1 variants in the absence and presence of M18BP1 siRNA treatment. Centromeres were detected in the CREST channel using a script for semiautomated quantification. Data shown as mean with SD of all quantified centromeres from three independent experiments. (E) Western blot of endogenous M18BP1 and of the indicated M18BP1 exogenous transgenes after induction of expression. The anti-M18BP1 antibody recognizes an epitope in the fragment 825–875. (F) Representative images of fixed HeLa cells showing CENP-A^SNAP fluorescence labeled with SNAP-Cell 647-SiR and GFP fluorescence of indicated stably expressed M18BP1 variants in early G1. Centromeres were visualized by CREST sera. The G1 couple is shown in the differential interference contrast (DIC) channel. White scale bars indicate 10 μm. The same control micrographs are shown in Figs. EV3D and EV4B,F, since all data were obtained from the same experiment. (G, H) Quantification of the centromeric CENP-A-SNAP and GFP fluorescence intensities of HeLa cell lines expressing the indicated exogenous M18BP1 variants. Centromeres were detected in the CREST channel using a script for semiautomated quantification. Data shown as mean with SD of all quantified centromeres from three independent experiments. Source data are available online for this figure.

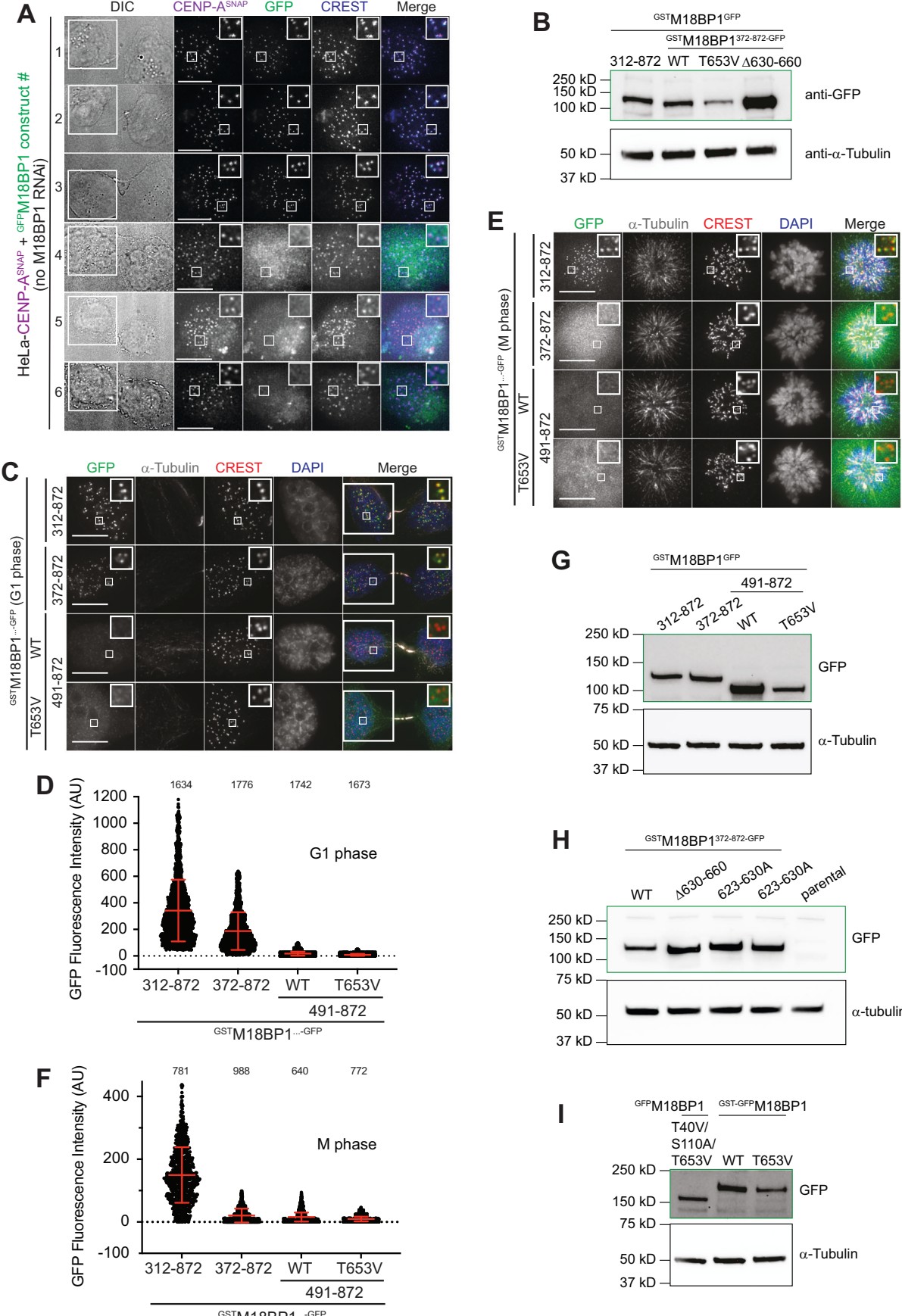

◀ **Figure EV5. The kinetochore-localization determinant of the 491-872 region depends on the presence of the SANTA domain.**

(A) Representative images of fixed HeLa cells showing CENP-A$^{SNAP}$ fluorescence labeled with SNAP-Cell 647-SiR and GFP fluorescence of indicated stably expressed M18BP1 variants in early G1. Centromeres were visualized by CREST sera. The G1 couple is shown in the differential interference contrast (DIC) channel. White scale bars indicate 10 µm. (B) Anti-GFP and anti-Tubulin immunoblots of HeLa cell lysates expressing the indicated exogenous $^{GST}$M18BP1$^{GFP}$ variants. (C) Representative images of fixed HeLa cells showing GFP fluorescence of indicated stably expressed $^{GST}$M18BP1$^{GFP}$ variants and immuno-stained α-tubulin in G1 phase. Centromeres were visualized by CREST sera, and DNA was stained by DAPI. White scale bars indicate 10 µm. (D) Quantification of the centromeric GFP-fluorescence intensities of the indicated expressed M18BP1 variants in G1 phase. Centromeres were detected in the CREST channel using a script for semiautomated quantification. Data shown as mean with SD of all quantified centromeres from three independent experiments. (E) Representative images of fixed HeLa cells showing GFP fluorescence of indicated stably expressed $^{GST}$M18BP1$^{GFP}$ variants and immuno-stained α-tubulin in mitosis. Centromeres were visualized by CREST sera, and DNA was stained by DAPI. White scale bars indicate 10 µm. (F) Quantification of the centromeric GFP-fluorescence intensities of the indicated expressed M18BP1 variants in mitosis. Centromeres were detected in the CREST channel using a script for semiautomated quantification. Data shown as mean with SD of all quantified centromeres from three independent experiments. (G–I) Anti-GFP and anti-Tubulin immunoblots of HeLa cell lysates expressing the indicated exogenous M18BP1 variants. Source data are available online for this figure.

