## [Peer Review File · The EMBO Journal]

M18BP1 valency and a distributed interaction footprint determine epigenetic centromere specification in humans

Kai Walstein, Louisa Hill, Doro Vogt, Lina Oberste-Lehn, Petra Janning, Ingrid Vetter, Dongqing Pan, and Andrea Musacchio

Corresponding author(s): Andrea Musacchio (andrea.musacchio@mpi-dortmund.mpg.de)

Review Timeline:

Submission Date:	26th Jun 25
Editorial Decision:	1st Aug 25
Revision Received:	21st Nov 25
Editorial Decision:	22nd Dec 25
Revision Received:	1st Jan 26
Accepted:	15th Jan 26

Editor: Hartmut Vodermaier

Transaction Report:

Dr. Andrea Musacchio
Max Planck Institute of Molecular Physiology
Mechanistic Cell Biology
Otto Hahn Strasse 11
Dortmund 44227
Germany

1st Aug 2025

Re: EMBOJ-2025-121730
M18BP1 valency and a distributed interaction footprint determine epigenetic centromere specification in humans

Dear Andrea,

Thank you for submitting your study on M18BP1 valency roles in human CENP-A incorporation pathways for our editorial consideration. We have now received input from three expert referees, copied below for your information. Since all referees acknowledge the interest and importance of the work as well as its overall quality, we would be happy to consider a revised version further for EMBO Journal publication. Nevertheless, the referees do raise a number of well-taken technical as well as presentational issues that I would ask you to address prior to resubmission. Should you want to discuss any of these points, or how to best address them, please do not hesitate to get back to me anytime during your revision; this may be helpful in light of our single-major-revision round policy. We would also be open to extending the revision deadline if that should be helpful.

Detailed information on preparing, formatting and uploading a revised manuscript can be found below and in our Guide to Authors. Thank you again for the opportunity to consider this work for The EMBO Journal, and I look forward to your revision in due time.

With kind regards,

Hartmut

5) Point-by-point response letters should include the original referee comments in full together with your detailed responses to

them (and to specific editor requests if applicable), and also be uploaded as editable (e.g., .docx) text files.

9) To facilitate reproducibility and cross-laboratory adoption of methodologies, please structure the Materials & Methods section as outlined in our guide to authors, including a completed Reagents and Tools Table that can be downloaded from our author guidelines as well (<https://www.embopress.org/page/journal/14602075/authorguide#structuredmethods>).

10) Digital image enhancement is acceptable practice, as long as it accurately represents the original data and conforms to community standards. If a figure has been subjected to significant electronic manipulation, this must be clearly noted in the figure legend and/or the 'Materials and Methods' section. The editors reserve the right to request original versions of figures and the original images that were used to assemble the figure. Finally, we generally encourage uploading of numerical as well as gel/blot image source data; for details see: embopress.org/page/journal/14602075/authorguide#sourcedata

In the interest of ensuring the conceptual advance provided by the work, we recommend submitting a revision within 3 months (30th Oct 2025). Please discuss the revision progress ahead of this time with the editor if you require more time to complete the revisions. Use the link below to submit your revision:

Link Not Available

Referee #1:

Human centromeres are epigenetically defined through a protein-based, self-templating feedback loop, in which chromatin-bound CENP-A-containing nucleosomes recruit the assembly machinery for the deposition of new CENP-A nucleosomes. Understanding how the assembly machinery is targeted to the centromere is of specific importance. This study by Walstein, Musacchio and colleagues aims to define how one key component of the assembly machinery, M18BP1, interacts with the centromere complex. They discover that the interaction of the assembly machinery with the centromere is largely mediated by M18BP1, and they define the key interaction surfaces responsible for this binding, how it is regulated by CDK phosphorylation, and the role of dimerisation of M18BP1 via Mis18 α and β in creating sufficient affinity for M18BP1 binding to the centromere.

Overall, this is a really excellent study, of very high technical quality and rigour, with numerous experiments systematically dissecting the contributions of specific regions within M18BP1 in centromere targeting. The paper is well-written with a balanced introduction and discussion, citing appropriate literature and outlining important differences in how the Mis18 complex is controlled in other species. The data overall are solid with the confirmation of key observations using different orthogonal approaches, including solid-phase binding assays, gel filtration, and in-cell complementation experiments. While key questions remain, particularly regarding the role of the conserved SANTA domain, for which the authors solved the structure, they nevertheless offer several important and novel insights into how the CENP-A assembly machinery may target centromeres.

While I'm in strong support of publication in principle, there are some adjustments I'd like to see in the manuscript and answers to a few questions.

Key points:

A central theme in the paper is the finding that the requirements for MIS18 α and β for M18BP1 recruitment to the centromere appear to be largely via their role in dimerising M18BP1. Consistent with this, the authors can bypass the need for MIS18 α/β by artificial dimerisation. GST should form a dimer efficiently, but I notice that the authors didn't actually show any evidence that M18BP1 is dimerised by GST. Including this would be helpful.

The observation that M18BP1 phosphorylation by CDKs promotes binding to CENP-C is somewhat surprising, considering that CDK activity has largely been described as inhibitory to the assembly process. In their discussion, the authors note the possibility that CDK phosphorylation regulates the role of BP1 in condensin recruitment in mitosis. However, their experiments show that phosphorylation of the N-terminal part of BP1, which is involved in CENP-A assembly, also promotes CENP-C binding. The authors should add a bit more discussion on this. Perhaps CDK phosphorylation of M18BP1 could have a role in recruitment of BP1, already in mitosis, that can then accelerate or facilitate CENP-A assembly, in G1 after loss of CDK activity.

In Figure S1, blots are shown, indicating expression levels of the various GFP-tagged constructs. However, there is no measure of how much these constructs are overexpressed relative to endogenous BP1 levels. Having some measure of this would be important, particularly in the case of mutants that showed dominant-negative effects on wild-type BP1 localisation and CENP-A assembly. Perhaps these extracts could be probed with an M18BP1 antibody that would detect both unmodified BP1 as well as the tagged constructs for comparison.

In Figure 1H and the middle of page 6, the authors claim a strong enhancement of the M18BP1 interaction with CENP-C due to CDK phosphorylation. A quantification is needed to support this statement. I note that under phosphorylated conditions, there is also more BP1 in the pull-downs. Perhaps the ratio between the amount of CENP-C in the pulldown versus input can be quantified and presented.

In Figure 6, the authors explore the curious role of T653. They hypothesise that its phosphorylation may regulate binding events within the 491-872 fragment. However, I suppose it's also possible that it controls interactions with the SANTA domain. The authors show in Supplemental Figure 7 that a complete loss of the SANTA domain cannot be rescued by a T653 mutation, but have the authors tested suppression of more subtle mutations within SANTA? I'm thinking of the WH point mutations that selectively interfere with SANTA binding.

Minor points:

The structure in Figure 1F is based on a prior publication listed as PDB TR5S. I may be incorrect, but I couldn't find this annotation in the database. Please double-check the PDB reference.

The structure in Figure 1F lacks CENP-C, which is then listed separately in Figure 1G as a linear map. It would help the reader to have a brief explanation of why CENP-C is not included in the structure. Perhaps add a sentence to state that CENP-C, due to its large unstructured domains, has not yet been solved within the CCAN complex. This may not be obvious to non-centromere experts.

In Figure 2F, the fully conserved residues are indicated with asterisks. I notice that there are a few fully conserved residues that are not highlighted. I'm wondering whether there is a specific reason for this, particularly the leucine residue at the end of the β 1 strand.

In Figure S1K, a western blot is shown of various M18BP1 deletion constructs. The number 3 construct appears to be larger than the number 1 construct, even though the number 1 protein should be bigger according to the map in Figure 5I. Perhaps there is a mix-up here?

In Figures 3B & 3C, it is shown that motif 3 is required in G1 phase but not in mitosis. The authors should briefly discuss this result at the top of page 8 to present their explanation for this effect.

In the middle of page 8, the authors cite a structurally stable CENP-HIKM mutant complex. What is the evidence that this complex is stable?

In the last paragraph of page 8, the authors discuss their observation that mutation of motif 2 renders the binding to the CCAN insensitive to phosphorylation. I suspect this is due to mutation of a CDK site at the boundary of motif 2. Only later, on page 10, do the authors bring up this possibility, i.e. the loss of a CDK site in the motif 2 mutant. Mentioning this already on page 8, something along the lines of "(see below)", would help guide the reader to the later explanation for the observation described on page 8.

Referee #2:

Maintaining correct levels of CENP-A at centromeres is crucial for establishing functional kinetochores capable of achieving high-fidelity genome transmission. The CENP-A loading machinery ensures correct CENP-A levels at centromeres by

replenishing CENP-A to compensate for the replication-mediated dilution of centromeric CENP-A through active CENP-A deposition. The Mis18 complex, consisting of Mis18BP1, Mis18a and Mis18b, is the core protein assembly of the CENP-A loading machinery that recruits CENP-A-specific chaperone HJURP loaded with CENP-A/H4 heterodimer to the centromere for subsequent incorporation into the centromeric chromatin. However, the molecular mechanism by which the CENP-A loading machinery is recruited to the centromere remains less well understood. Here, by combining a very careful and thorough structural and biochemical characterisation with cellular functional evaluations, Walstein et al., provide crucial insights into the molecular determinants contributing to the centromere recruitment of the CENP-A loading machinery. Overall, this study is a solid contribution to addressing an important knowledge gap in the field, and hence I have no hesitation in enthusiastically supporting the publication of this work. However, addressing the following points prior to publication, in my opinion, is important to strengthen some of the key conclusions of the work.

GFP-Mis18BP11-490, although it cannot dimerise on its own, shows kinetochore localisation on mitotic chromosomes during metaphase, albeit along with a strong diffused signal (Figure S1 C and E). It would be good to show whether GFP-Mis18BP1 1-490 retains or loses its ability to localise at mitotic kinetochores when expressed at a level comparable to that of GST-Mis18BP1 1-490-GFP. This is important to address (as to some extent, also discussed in the discussion section of the manuscript), particularly considering that Borsellini et al, Mol Cell, 2025 recently showed that Mis18BP1 activates condensin II at mitotic onset, and Figure 5B in that study shows 'punctae' like localisation of EGFP Mis18BP1 on mitotic chromosomes (resembling kinetochore localisation) during metaphase.

According to the AlphaFold predicted model shown in Figure S3E, 5A2 motif to make the most contacts with CENP-H/K appears, at least based on the orientation shown in the figure. It would be good to include the calculated buried accessible surface area for the three motifs (5A1, 5A2 and 5A3) in this model. If the 5A2 motif is the one that (through its interactions with CENP-H/K) orients 5A1 and 5A3 to make additional favourable interactions, why does the 5A2 mutant still manage to associate with kinetochores (Figure 3B and 3C)? As also noted below, please include ipTM and pLDDT scores corresponding to the models in the figure.

Figure 8, 2nd paragraph, "Thus, the crosslinking data provide a good explanation of the effects of mutations in Motifs 2 and 3." Motif 2 mutations (5A2) perturb the interaction with CENP-HIKM *in vitro*, but in cellular experiments, 5A2 localises to kinetochores as efficiently as the WT both in M phase and G1 phase. This needs to be discussed explicitly.

Only the interaction of M18BP1 with CENP-C is evaluated with and without CDK1 phosphorylation. It is not clear why CDK1 phosphorylation condition is not included in the evaluation of M18BP1 - CENP-HIKM interaction.

Figure 3F and S4C, show that mutating motif 1 (5A1) perturbs M18BP1 interaction with CENP-C and CENP-A nucleosome bound CENP-C. However, considering the spatial proximity of motifs 1 - 3 and that part of motif 1 is also involved in CENP-HIKM binding according to the AlphaFold model, can Motifs 1 -3 interact with both CENP-C and CENP-HIKM at the same time? It is important to address this question using recombinant proteins/protein complexes.

Page 12, last paragraph: "A related sequence motif is also present in HJURP (Figure S5E). AF3 predicts this new motif, which we refer to as Motif 4, to bind the same pocket of the Cupin domain bound by Motif 1." This is not clear. Including the AF3 models and their respective ipTM and pLDDT scores will be helpful.

Page 12, last paragraph: "In vitro, however, we only observed modest binding to the CENP-C Cupin domain of recombinant M18BP1 constructs encompassing Motif 4 (M18BP1581-630 or M18BP1591-640; LH and AM, unpublished observations)" - without any information/data on the kind of 'in vitro' experiments performed, 'we only observed modest binding' does not mean much.

One of the major conclusions of the study is that Mis18a/b contributes to the kinetochore recruitment of M18BP1 mainly by dimerising M18BP1. The possible kinetochore targets were identified by characterising the interaction of GST-M18BP1 and CENP-HIKM/CENP-C using recombinant proteins/protein complexes (Figure 3E for CENP-HIKM; Figure 3F and G and Figure 4D & E for CENP-C interaction). However, to strengthen these conclusions, it is important to show that M18BP1 dimerised via Mis18a/b (like in a physiological situation) can also interact with CENP-HIKM and CENP-C to the same extent as observed with the GST-M18BP1 with CENP-HIKM and CENP-C.

Minor points:

Page 6, last paragraph- "These results implicate the disordered segment M18BP1312-371 (herewith also referred to as pre-SANTA region) in CENP-C601-943 and CENP-HIKM binding" - This sentence needs to be rephrased. I guess part of the sentence was unintentionally deleted during editing.

Explicitly state domain boundaries of motifs 1 - 3 (5A1, 5A2 and 5A3)

Figure S3 A and B, for easy comparison, keep the order of the proteins the same in both figures. CENP-M is in a different

position.

Figure S3E, CENP-I Y488A is not highlighted in the figure. It would be helpful to highlight the CENP-K residues mutated based on XL-MS data in the figure.

Figure 4C, please label the boundaries of M18BP1 motif 1 in the cartoon diagram. Here and in other figures showing AlphaFold model include ipTM and pLDDT scores, indicating the confidence of complex interface prediction.

Referee #3:

The manuscript investigates the kinetochore targeting of M18BP1, a key player for CENP-A deposition at the centromere and thus centromere maintenance. The current model for CENP-A loading is that M18BP1 forms a complex with Mis18a/b in G1 to recruit the CENP-A/H4 chaperone HJURP for CENP-A deposition at the centromere. The manuscript challenges part of this model and demonstrates that it is not the complex formation with Mis18a/b per se, but the dimerization of M18BP1 that is required for kinetochore targeting. As this dimerization on Mis18a,b is normally prevented outside G1 by CDK-mediated phosphorylation, centromere targeting is only observed in G1. However, dimerization can be artificially induced by fusion with a GST tag, resulting in kinetochore targeting in absence of Mis18a/b binding.

The authors take advantage of this artificial dimerization and mainly use a combination of immunofluorescence and in vitro binding assays to probe different GST- and GFP-tagged M18BP constructs. They show that M18BP1 312-490 shows robust kinetochore localization in both G1 and M when dimerized via GST tag. The authors go on to use this to determine the centromere targeting sequences within M18BP1. This region of M18BP1 encompasses the pre-SANTA region (aa 312-371) and the SANTA domain (aa 380-490). It binds to the CENP-C C-terminal region, and the pre-SANTA region binds to CENP-HIKM. The authors solve the crystal structure of SANTA and identify conserved residues that are candidates for the interaction with the kinetochore, among which also three motifs in the pre-SANTA region. They show that Motif1 and WH residues within the SANTA domain are important for kinetochore localization, while Motif 3 is only important during interphase but not M. They further show that Motifs 2 and 3 are required to bind to CENP-HIKM, with CENP-K mainly providing the interaction interface. The interaction with CENP-C is mainly mediated through the SANTA domain and Motif 1 that targets the CUPIN domain of CENP-C. Acute depletion of CENP-C prevents M18BP1-GST construct localization to kinetochores. The authors show that the constructs that are targeted to the kinetochore are also proficient in CENP-A loading, and that there is an additional region C-terminal of the SANTA domain (M18BP1 491-872) that also contributes to kinetochore localization and thus CENP-A loading. Within this region, they identify an important CDK phosphorylation site that prevents kinetochore localization in mitosis, and an adjacent motif (Motif 4) that is important for kinetochore localization in G1.

The manuscript assembles an impressive amount of data that reveal intriguing details about how M18BP1 is recruited to the centromere for the maintenance of CENP-A and centromeres. The experiments are well designed and documented, and significantly advance our understanding of how M18BP1 interacts with the kinetochore. The reconcile previous, partly contradictory observations, and will be the basis for further studies of CENP-A maintenance. Having that said, the text sometimes does not entirely reflect the data, and there are a few puzzling observations that are not explained. It would be helpful if the authors could address these in a revised version of the manuscript.

1. The authors claim that M18BP1 1-490 in G1 phase localized to kinetochores regardless of the presence of GST or not (Figure S1B,D), while in M phase it only localized in the presence of GST (Figure S1 C,E). However, in the data, it is very hard to make out a difference in localization between G1 and S phases, and both constructs behave similarly in both cases. I understand the overexpression argument, but given that this is the premise of the whole paper, the justification should be better explained or documented.

2. Figure 1H. The N-terminal part of CENP-C is binding weakly to M18BP1. This is mentioned, but not considered for the later discussion of the interaction between the two proteins, nor in the model.

3. Figure 3D. The 5A3 mutant is present at kinetochores at M phase, but not at G1 phase (so the opposite of full length untagged M18BP1). This is not sufficiently explained. What prevents the 5A3 mutant to localize to the kinetochore during G1?

4. Figure 3E-F. The 5A2 mutant does not bind CENP-HIKM anymore, yet this mutant localizes like wildtype in G1 and M phase. This suggests that binding to CENP-HIKM is not required for kinetochore targeting. The same mutant also binds more weakly to CENP-C. It remains unexplained why this mutant localizes to kinetochores in Figure 2B-D. Unfortunately, this mutant was not tested in the size-exclusion chromatography experiments in Figure S4C.

5. Figure 3G. There seems to be a slight increase in binding upon phosphorylation of CENP-C, but the authors state that phosphorylation of CENP-C has no effect.

6. Are CENP-HIKM still recruited to the kinetochore upon CENP-C depletion? The authors argue that it is not, based on previous

findings, but it will be important to show this in the context of the current study, as it is critically important for the interpretation of the results.

7. Figure 5G. The authors claim that the WH/AA mutant has a mild dominant negative effect on CENP-A loading in the presence endogenous M18BP1. This is not visible in the data. In fact, CENP-A signal is higher in the WH/AA mutant compared to the WT.

8. Figure 5K-L. How can M18BP1 constructs lacking the region 491-872 be dominant negative for M18BP1 localization, yet still mediate CENP-A loading? This is not the case for the dominant negative 1-490 constructs (panel C-D), where a dominant negative behavior of the M18BP1 leads to an absence of CENP-A loading. This is a puzzling observation that requires more explanation.

9. Figure 5I-L. The authors conclude that the constructs lacking the SANT domain (2 and 3) behave like wildtype, which leads them to state that the SANT domain is not involved in kinetochore targeting. However, the data show a clear reduction in kinetochore targeting for these constructs, albeit not as strong as for constructs 4, 5 and 6. The conclusion that the SANT domain has no role in kinetochore targeting therefore seems too strong.

10. Figure 6J. The authors call the kinetochore localization of the WT fragment lacking the pre-SANTA and the T653 mutants during G1 phase "robust". However, they seem to be clearly less targeted to the kinetochore compared the 312-872 fragment.

11. The importance of M18BP1 dimerization for kinetochore targeting is a key observation for this manuscript. However, it remains unexplained why M18BP1 needs to dimerize for kinetochore localization. Given all the interaction interfaces, why are dimers, but not monomers, targeted to kinetochores in absence of M18a/b interaction?

Referee #1:

Human centromeres are epigenetically defined through a protein-based, self-templating feedback loop, in which chromatin-bound CENP-A-containing nucleosomes recruit the assembly machinery for the deposition of new CENP-A nucleosomes. Understanding how the assembly machinery is targeted to the centromere is of specific importance. This study by Walstein, Musacchio and colleagues aims to define how one key component of the assembly machinery, M18BP1, interacts with the centromere complex. They discover that the interaction of the assembly machinery with the centromere is largely mediated by M18BP1, and they define the key interaction surfaces responsible for this binding, how it is regulated by CDK phosphorylation, and the role of dimerisation of M18BP1 via Mis18 α and β in creating sufficient affinity for M18BP1 binding to the centromere.

Overall, this is a really excellent study, of very high technical quality and rigour, with numerous experiments systematically dissecting the contributions of specific regions within M18BP1 in centromere targeting. The paper is well-written with a balanced introduction and discussion, citing appropriate literature and outlining important differences in how the Mis18 complex is controlled in other species. The data overall are solid with the confirmation of key observations using different orthogonal approaches, including solid-phase binding assays, gel filtration, and in-cell complementation experiments. While key questions remain, particularly regarding the role of the conserved SANTA domain, for which the authors solved the structure, they nevertheless offer several important and novel insights into how the CENP-A assembly machinery may target centromeres.

While I'm in strong support of publication in principle, there are some adjustments I'd like to see in the manuscript and answers to a few questions.

We are very grateful to the reviewer for a positive assessment of our work

Key points:

A central theme in the paper is the finding that the requirements for MIS18 α and β for M18BP1 recruitment to the centromere appear to be largely via their role in dimerising M18BP1. Consistent with this, the authors can bypass the need for MIS18 α / β by artificial dimerisation. GST should form a dimer efficiently, but I notice that the authors didn't actually show any evidence that M18BP1 is dimerised by GST. Including this would be helpful.

We agree with the reviewer that this is an important control. We carried out mass spectrometry experiments that verified the expected molecular mass of the (unfolded) monomers of MBP- and GST-M18BP1³¹²⁻⁴⁹⁰ (Figure EV1A-C). We next carried out mass photometry experiments with the same material to measure the mass of the native complexes (Figure EV1D-E). A molecular mass of 54 kDa was measured for MBP-M18BP1³¹²⁻⁴⁹⁰, ~10.7 kDa lower than the monomer mass measured by mass spectrometry. On the other hand, masses of 94 and 69 kDa were measured for GST-M18BP1³¹²⁻⁴⁹⁰. Thus, for both samples, mass photometry provided mass measures that were lower than those expected for the M18BP1³¹²⁻⁴⁹⁰ monomer (MBP-tagged) or dimer (GST-tagged), with comparable offsets (10.7 kDa for the MBP monomer and 6 kDa for the GST dimer). These results provide strong support for GST-driven dimerization. The 69 kDa peak in the GST-M18BP1³¹²⁻⁴⁹⁰ fusion may represent a degradation product or a measurement artefact. In a new experiment in Figure EV1J, we used GFP as bait to immobilize CENP-C, and probed this bait with GST- or MBP-M18BP1³¹²⁻⁴⁹⁰ prey. The result confirmed that GST dimerization promotes stronger binding of M18BP1 to CENP-C.

The observation that M18BP1 phosphorylation by CDKs promotes binding to CENP-C is

somewhat surprising, considering that CDK activity has largely been described as inhibitory to the assembly process. In their discussion, the authors note the possibility that CDK phosphorylation regulates the role of BP1 in condensin recruitment in mitosis. However, their experiments show that phosphorylation of the N-terminal part of BP1, which is involved in CENP-A assembly, also promotes CENP-C binding. The authors should add a bit more discussion on this. Perhaps CDK phosphorylation of M18BP1 could have a role in recruitment of BP1, already in mitosis, that can then accelerate or facilitate CENP-A assembly, in G1 after loss of CDK activity.

As suggested by the reviewer, we have briefly expanded the Discussion to include a comment on this counterintuitive observation. Specifically, we now write: “As cyclin-dependent activity inhibits assembly of the M18BP1:Mis18 complex and CENP-A deposition, the observation that M18BP1 phosphorylation by CDK1 in the pre-SANTA region promotes binding to CENP-C is surprising. The functional relevance of this mitotic phosphorylation of the pre-SANTA localization module is unclear. It may promote a weak interaction with centromeres that facilitates deposition of Condensin II. Alternatively, or in addition, it may facilitate CENP-A deposition in the subsequent G1 phase. These hypotheses will need to be verified in future studies.”

In Figure S1, blots are shown, indicating expression levels of the various GFP-tagged constructs. However, there is no measure of how much these constructs are overexpressed relative to endogenous BP1 levels. Having some measure of this would be important, particularly in the case of mutants that showed dominant-negative effects on wild-type BP1 localisation and CENP-A assembly. Perhaps these extracts could be probed with an M18BP1 antibody that would detect both unmodified BP1 as well as the tagged constructs for comparison.

We agree with the reviewer on this point, and note that the comment also echoes a similar point by Reviewer 2. We now present Western blots obtained with an in-house generated anti-M18BP1 antibody, which recognizes an epitope within residues 825-875 in the C-terminal region of M18BP1 and can therefore only be used with exogenous constructs expressing full length M18BP1. As the focus of these experiments was to assess what level of expression of M18BP1 transgenes results in a dominant-negative effect, we have limited this analysis to full length M18BP1 constructs with dominant-negative effects on endogenous M18BP1 function. We have included this new control as Figure EV4E, to complement the observations in Figure EV4G-H.

In Figure 1H and the middle of page 6, the authors claim a strong enhancement of the M18BP1 interaction with CENP-C due to CDK phosphorylation. A quantification is needed to support this statement. I note that under phosphorylated conditions, there is also more BP1 in the pull-downs. Perhaps the ratio between the amount of CENP-C in the pulldown versus input can be quantified and presented.

In addition to Figure 1H, the role of CDK1 on the interaction of M18BP1 with CENP-C is examined more thoroughly in Figure 3G. We have now repeated these experiments and provide quantifications in Figure EV1I.

In Figure 6, the authors explore the curious role of T653. They hypothesise that its phosphorylation may regulate binding events within the 491-872 fragment. However, I suppose it's also possible that it controls interactions with the SANTA domain. The authors show in Supplemental Figure 7 that a complete loss of the SANTA domain cannot be rescued by a T653 mutation, but have the authors tested suppression of more subtle mutations within SANTA? I'm thinking of the WH point mutations that selectively interfere with SANTA binding.

We carefully considered this experiment, but eventually decided not to perform it as we suspect its results would be inconclusive. The reason is that we only observed a rescue of mitotic localization after mutating T653 for constructs that show localization in G1 (when T653 is not phosphorylated). Since the WH/AA mutant, like the Δ SANTA, does not localize to kinetochores in G1 (when T653 is dephosphorylated, or in any other cell cycle phases), we anticipate that mutating T653 would not rescue localization.

Minor points:

The structure in Figure 1F is based on a prior publication listed as PDB 7R5S. I may be incorrect, but I couldn't find this annotation in the database. Please double-check the PDB reference.

We apologize for the confusion, the correct PDB code is 7R5S, and not TR5S as written in the original manuscript. We have corrected this.

The structure in Figure 1F lacks CENP-C, which is then listed separately in Figure 1G as a linear map. It would help the reader to have a brief explanation of why CENP-C is not included in the structure. Perhaps add a sentence to state that CENP-C, due to its large unstructured domains, has not yet been solved within the CCAN complex. This may not be obvious to non-centromere experts.

The structure of CCAN in Figure 1F has very short segments of CENP-C, but indeed the majority of CENP-C is intrinsically disordered. We have added a sentence to the legend to clarify this important point, as suggested by the reviewer: "Due to its large disordered regions, CENP-C was largely invisible in the structure."

In Figure 2F, the fully conserved residues are indicated with asterisks. I notice that there are a few fully conserved residues that are not highlighted. I'm wondering whether there is a specific reason for this, particularly the leucine residue at the end of the β 1 strand.

We thank the reviewer for raising this point. This was not intentional and we have now harmonized the labelling and revised the figure accordingly. We had already reported in the legend that "Conserved residues in the hydrophobic core and on the surface are indicated with black and green asterisk, respectively."

In Figure S1K, a western blot is shown of various M18BP1 deletion constructs. The number 3 construct appears to be larger than the number 1 construct, even though the number 1 protein should be bigger according to the map in Figure 5I. Perhaps there is a mix-up here?

The reviewer is correct. In Figure 5I the schemes and names of construct 1 and 3 had been swapped by mistake. The correct order should have been 1: Δ 873-1132, 2: Δ 875-925, 3: Δ 931-1132, as all data (galleries, quantification, WB) match this order of constructs. We have therefore corrected Figure 5I.

In Figures 3B & 3C, it is shown that motif 3 is required in G1 phase but not in mitosis. The authors should briefly discuss this result at the top of page 8 to present their explanation for this effect.

We believe this to be due to CDK1 activity, which increases the binding affinity for CENP-C in mitosis, reducing dependence on motif 3 (i.e. on HIKM interaction). Therefore, we now write:

“Thus, Motif 3 is necessary for interphase recruitment of ^{GST}M18BP1^{312-490-EGFP}, but dispensable for its mitotic localization, likely because increased binding affinity caused by CDK1 activity (as shown in detail below) reduces dependency on this motif during mitosis.”

In the middle of page 8, the authors cite a structurally stable CENP-HIKM mutant complex. What is the evidence that this complex is stable?

We agree with the reviewer that we had not provided evidence that the mutant is stable. We have now formally compared the behavior of the wild type and mutant complexes by size-exclusion chromatography and show them to be identical. We present these new data in Figure EV2C.

In the last paragraph of page 8, the authors discuss their observation that mutation of motif 2 renders the binding to the CCAN insensitive to phosphorylation. I suspect this is due to mutation of a CDK site at the boundary of motif 2. Only later, on page 10, do the authors bring up this possibility, i.e. the loss of a CDK site in the motif 2 mutant. Mentioning this already on page 8, something along the lines of "(see below)", would help guide the reader to the later explanation for the observation described on page 8.

We have now included “(an observation on which we return below” to inform readers that the issue is discussed more thoroughly in a subsequent section.

Referee #2:

Maintaining correct levels of CENP-A at centromeres is crucial for establishing functional kinetochores capable of achieving high-fidelity genome transmission. The CENP-A loading machinery ensures correct CENP-A levels at centromeres by replenishing CENP-A to compensate for the replication-mediated dilution of centromeric CENP-A through active CENP-A deposition. The Mis18 complex, consisting of Mis18BP1, Mis18a and Mis18b, is the core protein assembly of the CENP-A loading machinery that recruits CENP-A-specific chaperone HJURP loaded with CENP-A/H4 heterodimer to the centromere for subsequent incorporation into the centromeric chromatin. However, the molecular mechanism by which the CENP-A loading machinery is recruited to the centromere remains less well understood. Here, by combining a very careful and thorough structural and biochemical characterisation with cellular functional evaluations, Walstein et al., provide crucial insights into the molecular determinants contributing to the centromere recruitment of the CENP-A loading machinery. Overall, this study is a solid contribution to addressing an important knowledge gap in the field, and hence I have no hesitation in enthusiastically supporting the publication of this work. However, addressing the following points prior to publication, in my opinion, is important to strengthen some of the key conclusions of the work.

We are very grateful to the reviewer for a positive assessment of our work

GFP-Mis18BP11-490, although it cannot dimerise on its own, shows kinetochore localisation on mitotic chromosomes during metaphase, albeit along with a strong diffused signal (Figure S1 C and E). It would be good to show whether GFP-Mis18BP1 1-490 retains or loses its ability to localise at mitotic kinetochores when expressed at a level comparable to that of GST-Mis18BP1 1-490-GFP. This is important to address (as to some extent, also discussed in the discussion section of the manuscript), particularly considering that Borsellini et al, Mol Cell, 2025 recently showed that Mis18BP1 activates condensin II at mitotic onset, and Figure 5B in that study shows 'punctae' like localisation of EGFP Mis18BP1 on mitotic chromosomes (resembling kinetochore localisation) during metaphase.

This is a fair point, but also one that is technically hard to address, due to the different expression levels of our GFP constructs with or without GST, with the latter (without GST) being expressed at higher levels. We have nevertheless tried to answer this question with new panels in Figure EV1F-H. In these panels, we compared the chromatin and kinetochore signals of the GFP and GFP-GST constructs in individual cells. We used the chromatin levels as a proxy for expression levels in each cell. We classified chromatin GFP levels as low- and high. We then asked how the kinetochore levels compared in these two groups. In cells of the “chromatin-low” group, the mean GFP kinetochore signal of the GFP-GST construct was decidedly higher than that of the GFP-only construct. In cells of the “chromatin-high” group, the mean chromatin levels were much higher for the GFP-only construct, due to overall higher expression levels. Nevertheless, the mean of the distribution of kinetochore levels was essentially identical, in line with our contention that GST dimerization facilitates kinetochore localization. In this revision, we have added several additional experiments that support the role of GST in dimerization of M18BP1 and increased binding to kinetochore receptors, including a direct demonstration with purified proteins in Figure EV1J.

According to the AlphaFold predicted model shown in Figure S3E, 5A2 motif to make the most contacts with CENP-H/K appears, at least based on the orientation shown in the figure. It would be good to include the calculated buried accessible surface area for the three motifs (5A1, 5A2 and 5A3) in this model. If the 5A2 motif is the one that (through its interactions with CENP-H/K) orients 5A1 and 5A3 to make additional favourable interactions, why does the 5A2 mutant still manage to associate with kinetochores (Figure 3B and 3C)? As also noted below, please include ipTM and pLDDT scores corresponding to the models in the figure.

We thank the reviewer for raising these points. We have added the ipTM and pLDDT scores of the model in a dedicated Appendix Figure S3. We have calculated the accessible surface areas with program "NACCESS" (Hubbard, S.J. and Thornton, J.M. (1993). Mis18BP1 buries in total 1893.8 Å² of chain K and 671.7 Å² of chain H, for a total of 2565.5 Å², with no additional contacts with other chains of CENP-HIKM-TWSX. These are distributed as follows: 5A1 buries 199.74 Å² of chain K and 149.91 Å² of chain H, for a total of 349.65 Å²; 5A2 buries 233.04 Å² of chain K and 192.60 of chain H, for a total of 425.64 Å², and 5A3 buries 260.41 Å² of chain K, and nothing on chain H, for a total of 260.41 Å². Thus, the results confirm the idea that the more extensive interaction of HIKM is with the 5A2 motif. However, we would like to clarify that our observations only imply that motif 2 is not strictly necessary for kinetochore recruitment of M18BP1, which does not imply that it is not implicated in an interaction with HIKM. We interpret our data to indicate that this interaction can be compensated by other interactions of neighboring motifs.

Figure 8, 2nd paragraph, "Thus, the crosslinking data provide a good explanation of the effects of mutations in Motifs 2 and 3." Motif 2 mutations (5A2) perturb the interaction with CENP-HIKM in vitro, but in cellular experiments, 5A2 localises to kinetochores as efficiently as the WT both in M phase and G1 phase. This needs to be discussed explicitly.

We agree with the reviewer and we now explicitly discuss this point as follows: “In contrast, even if interfering with the interaction with CENP-HIKM in vitro, the 5A2 mutant showed robust CENP-A loading in G1...in agreement with our observation that it lacks penetrance even when introduced in the minimal localization module ^{GST}M18BP1^{312-490-GFP} (Figure 3B-D).”

Only the interaction of M18BP1 with CENP-C is evaluated with and without CDK1 phosphorylation. It is not clear why CDK1 phosphorylation condition is not included in the evaluation of M18BP1 - CENP-HIKM interaction.

The reason for this is Figure 1H, where we show there is no noticeable difference in a GST-M18BP1/CENP-HIKM pull-down in the presence and absence of CDK1. Thus, we did not include an analysis of the effects of CDK1 phosphorylation in subsequent experiments with these constructs.

Figure 3F and S4C, show that mutating motif 1 (5A1) perturbs M18BP1 interaction with CENP-C and CENP-A nucleosome bound CENP-C. However, considering the spatial proximity of motifs 1 - 3 and that part of motif 1 is also involved in CENP-HIKM binding according to the AlphaFold model, can Motifs 1-3 interact with both CENP-C and CENP-HIKM at the same time? It is important to address this question using recombinant proteins/protein complexes.

This is an important question, and we had already tried moving steps to address it but we eventually opted not to include new results in the manuscript because we feel that they are premature. Alphafold predictions suggest that CENP-HIKM and CENP-C can interact concomitantly with motifs 1-3. Size-exclusion chromatography experiments, on the other hand, have suggested a weak competition. We are not yet confident regarding the significance of these experiments, in particular because we cannot exclude a role of the SANTA domain in stabilization of an intra-molecular conformation of M18BP1 that would accommodate all binders at the same time. We have therefore opted not to include new predictions and experiments that we do not consider robust enough to be truly informative at this stage.

Page 12, last paragraph: "A related sequence motif is also present in HJURP (Figure S5E). AF3 predicts this new motif, which we refer to as Motif 4, to bind the same pocket of the Cupin domain bound by Motif 1." This is not clear.

predicts this new motif, which we refer to as Motif 4, to bind the same pocket of the Cupin domain bound by Motif 1." This is not clear. Including the AF3 models and their respective ipTM and pLDDT scores will be helpful.

We have now included the AF3 model and the respective ipTM and pLDDT scores into a new Appendix Figure S3.

Page 12, last paragraph: "In vitro, however, we only observed modest binding to the CENP-C Cupin domain of recombinant M18BP1 constructs encompassing Motif 4 (M18BP1581-630 or M18BP1591-640; LH and AM, unpublished observations)" - without any information/data on the kind of 'in vitro' experiments performed, 'we only observed modest binding' does not mean much.

We include the mentioned pulldowns (performed with M18BP1 fragments containing

motif 4) here for the reviewer's examination. They are ordered according to GST-M18BP1 vs. MBP-CENP-C used as bait. The upper experiments were done using the 50-residue fragments, while in the lower ones a 372-630 fragment was used as well. None of these experiments gave a truly definitive answer, as already indicated, and that is the reason why we decided not to include them in the manuscript.

One of the major conclusions of the study is that Mis18a/b contributes to the kinetochore recruitment of M18BP1 mainly by dimerising M18BP1. The possible kinetochore targets were identified by characterising the interaction of GST-M18BP1 and CENP-HIKM/CENP-C using recombinant proteins/protein complexes (Figure 3E for CENP-HIKM; Figure 3F and G and Figure 4D & E for CENP-C interaction). However, to strengthen these conclusions, it is important to show that M18BP1 dimerised via Mis18a/b (like in a physiological situation) can also interact with CENP-HIKM and CENP-C to the same extent as observed with the GST-M18BP1 with CENP-HIKM and CENP-C.

We thank the reviewer for raising this suggestion. We note that the multimerization effect we have postulated is only effective if a multivalent prey (a GST-dimerized prey in this case) is exposed to a properly distributed bait. By this we imply a bait distributed in such a way that concomitant binding of the prey's two moieties (generated by the dimerization mechanism) is possible. For various reasons, it is not a foregone expectation that our *in vitro* reconstitutions ought to reproduce this. For instance, the distribution of the bait may not mimic the active distribution *in vivo*. Furthermore, not all elements of the kinetochore receptor contributing to M18BP1 recruitment are currently known. Therefore, we don't know which of them has the most impact on the dimerization mechanism we have postulated. While we started experimenting along the lines proposed by the reviewer, the respectively permanent and temporary departures from the laboratory of Drs. Kai Walstein and Louisa Hill, the manuscript's first authors, prevented us from providing a complete answer to the question. Hampering complete progress were the necessity of introducing new tags on the CENP-HIKM complex to turn it into a suitable prey, and making new preparations of the Mis18 α/β complex as bait. We were therefore unable to complete these experiments and apologize for it. Nonetheless, we present a new experiment in Figure EV2F demonstrating that addition of the Mis18 α/β complex does not interfere with the binding of immobilized M18BP1 to CENP-HIKM. In another new experiment, we demonstrate that GST enhances binding of M18BP1 to immobilized GFP-CENP-C (Figure EV1J).

Minor points:

Page 6, last paragraph- "These results implicate the disordered segment M18BP1312-371 (herewith also referred to as pre-SANTA region) in CENP-C601-943 and CENP-HIKM binding" - This sentence needs to be rephrased. I guess part of the sentence was unintentionally deleted during editing.

After rereading the sentence, we felt it is correct as written

Explicitly state domain boundaries of motifs 1 - 3 (5A1, 5A2 and 5A3)

We now explicitly state domain boundaries and sequences of the motifs.

Figure S3 A and B, for easy comparison, keep the order of the proteins the same in both figures. CENP-M is in a different position.

Thank you, we have corrected this and we now show plots with the same order.

Figure S3E, CENP-I Y488A is not highlighted in the figure. It would be helpful to highlight the CENP-K residues mutated based on XL-MS data in the figure.

Thank you for the suggestion. We have generated a new panel Figure EV2D that shows the position of mutated residues on CENP-K. We highlight the same area also in panel G of the same figure, where we display the AF prediction.

Figure 4C, please label the boundaries of M18BP1 motif 1 in the cartoon diagram. Here and in other figures showing AlphaFold model include ipTM and pLDDT scores, indicating the confidence of complex interface prediction.

We now labelled the boundaries in the cartoon of Figure 4C. ipTM and pLDDT scores of all AF predictions are now collected in Appendix Figure S3.

Referee #3:

The manuscript investigates the kinetochore targeting of M18BP1, a key player for CENP-A deposition at the centromere and thus centromere maintenance. The current model for CENP-A loading is that M18BP1 forms a complex with Mis18a/b in G1 to recruit the CENP-A/H4 chaperone HJURP for CENP-A deposition at the centromere. The manuscript challenges part of this model and demonstrates that it is not the complex formation with Mis18a/b per se, but the dimerization of M18BP1 that is required for kinetochore targeting. As this dimerization on Mis18a,b is normally prevented outside G1 by CDK-mediated phosphorylation, centromere targeting is only observed in G1. However, dimerization can be artificially induced by fusion with a GST tag, resulting in kinetochore targeting in absence of Mis18a/b binding.

The authors take advantage of this artificial dimerization and mainly use a combination of immunofluorescence and in vitro binding assays to probe different GST- and GFP-tagged M18BP constructs. They show that M18BP1 312-490 shows robust kinetochore localization in both G1 and M when dimerized via GST tag. The authors go on to use this to determine the centromere targeting sequences within M18BP1. This region of M18BP1 encompasses the pre-SANTA region (aa 312-371) and the SANTA domain (aa 380-490). It binds to the CENP-C C-terminal region, and the pre-SANTA region binds to CENP-HIKM. The authors solve the crystal structure of SANTA and identify conserved residues that are candidates for the interaction with the kinetochore, among which also three motifs in the pre-SANTA region. They show that Motif1 and WH residues within the SANTA domain are important for kinetochore localization, while Motif 3 is only important during interphase but not M. They further show that Motifs 2 and 3 are required to bind to CENP-HIKM, with CENP-K mainly providing the interaction interface. The interaction with CENP-C is mainly mediated through the SANTA domain and Motif 1 that targets the CUPIN domain of CENP-C. Acute depletion of CENP-C prevents M18BP1-GST construct localization to kinetochores. The authors show that the constructs that are targeted to the kinetochore are also proficient in CENP-A loading, and that there is an additional region C-terminal of the SANTA domain (M18BP1 491-872) that also contributes to kinetochore localization and thus CENP-A loading. Within this region, they identify an important CDK phosphorylation site that prevents kinetochore localization in mitosis, and an adjacent motif (Motif 4) that is important for kinetochore localization in G1.

The manuscript assembles an impressive amount of data that reveal intriguing details about how M18BP1 is recruited to the centromere for the maintenance of CENP-A and centromeres. The experiments are well designed and documented, and significantly advance our understanding of how M18BP1 interacts with the kinetochore. The reconcile previous, partly contradictory observations, and will be the basis for further studies of CENP-A maintenance. Having that said,

the text sometimes does not entirely reflect the data, and there are a few puzzling observations that are not explained. It would be helpful if the authors could address these in a revised version of the manuscript.

We are very grateful to the reviewer for a positive assessment of our work

1. The authors claim that M18BP1 1-490 in G1 phase localized to kinetochores regardless of the presence of GST or not (Figure S1B,D), while in M phase it only localized in the presence of GST (Figure S1 C,E). However, in the data, it is very hard to make out a difference in localization between G1 and S phases, and both constructs behave similarly in both cases. I understand the overexpression argument, but given that this is the premise of the whole paper, the justification should be better explained or documented.

This concern echoes the first point raised by Referee #2. We have now carried out new experiments to address this point, and kindly ask the reviewer to refer to our answer of the first point of Referee #2.

2. Figure 1H. The N-terminal part of CENP-C is binding weakly to M18BP1. This is mentioned, but not considered for the later discussion of the interaction between the two proteins, nor in the model.

Given that this interaction appears to be very weak and involves an unfolded region on CENP-C, which might increase the risk of an unspecific effect, we assumed it might be spurious and did not give it further attention in this study. We opted not to include this interaction in the model as we feel this is not fully supported yet. We have included a short comment on this in the context of the discussion of our summary model in Figure 7G. “The N-terminal region of CENP-C binds weakly to M18BP1 (Figure 1H) and may also contribute to this interaction scheme, but here we did not characterize its role further.”

3. Figure 3D. The 5A3 mutant is present at kinetochores at M phase, but not at G1 phase (so the opposite of full length untagged M18BP1). This is not sufficiently explained. What prevents the 5A3 mutant to localize to the kinetochore during G1?

This concern also echoes a minor point by reviewer 1. We have now explicitly clarified our line of thinking on this observation and write: “Thus, Motif 3 is necessary for interphase recruitment of ^{GST}M18BP1^{312-490-EGFP}, but dispensable for its mitotic localization, likely because increased binding affinity caused by CDK1 activity (as shown in detail below) reduces dependency on this motif during mitosis.”

4. Figure 3E-F. The 5A2 mutant does not bind CENP-HIKM anymore, yet this mutant localizes like wildtype in G1 and M phase. This suggests that binding to CENP-HIKM is not required for kinetochore targeting. The same mutant also binds more weakly to CENP-C. It remains unexplained why this mutant localizes to kinetochores in Figure 2B-D. Unfortunately, this mutant was not tested in the size-exclusion chromatography experiments in Figure S4C.

We think that the reviewer meant Figure 3B-D, not Figure 2B-D. The reviewer is correct. We did not run SEC assays with this mutant as we did not expect major differences, given that the mutant construct seems to localize normally.

5. Figure 3G. There seems to be a slight increase in binding upon phosphorylation of CENP-C, but the authors state that phosphorylation of CENP-C has no effect.

We now write: "...while pre-phosphorylation of ^{MBP}CENP-C⁶⁰¹⁻⁹⁴³ only caused a very minor difference"

6. Are CENP-HIKM still recruited to the kinetochore upon CENP-C depletion? The authors argue that it is not, based on previous findings, but it will be important to show this in the context of the current study, as it is critically important for the interpretation of the results.

We have already performed this experiment in the two cited instances, and are writing up a third manuscript in which we confirm the slow loss of other CCAN subunits (which requires hours) after the extremely rapid CENP-C depletion caused by IAA (which occurs in minutes). That CENP-C is the dominant cause of this phenotype is corroborated by the localization of the 5A2 mutant in the context of the 312-490 fragment.

7. Figure 5G. The authors claim that the WH/AA mutant has a mild dominant negative effect on CENP-A loading in the presence endogenous M18BP1. This is not visible in the data. In fact, CENP-A signal is higher in the WH/AA mutant compared to the WT.

We write in the manuscript that "The SANTA deletion mutant, and to a minor extent the WH/AA mutant, had a dominant negative effect on CENP-A loading in presence of endogenous M18BP1, an effect that was more prominent in cells with high expression levels of these M18BP1 mutants (Figure S8E-G)." The data supporting our claim are those in former Figure S8E-G, now Figure EV4E-H, which we have extended to include a blot demonstrating expression levels of exogenous constructs relative to endogenous M18BP1 protein. The other data in this set demonstrate that high expression levels of the transgene suppress CENP-A loading.

8. Figure 5K-L. How can M18BP1 constructs lacking the region 491-872 be dominant negative for M18BP1 localization, yet still mediate CENP-A loading? This is not the case for the dominant negative 1-490 constructs (panel C-D), where a dominant negative behavior of the M18BP1 leads to an absence of CENP-A loading. This is a puzzling observation that requires more explanation.

We have carefully considered the reviewer's comment but did not fully understand it. We do not display the localization of endogenous M18BP1 in these experiments. As there is no localization of the constructs mentioned by the reviewer (Δ 491-872 and containing the 5A3 mutation), CENP-A loading in the absence of RNAi is carried out by endogenous M18BP1, and it does not occur at all when endogenous M18BP1 is depleted. We see no contradiction.

9. Figure 5I-L. The authors conclude that the constructs lacking the SANT domain (2 and 3) behave like wildtype, which leads them to state that the SANT domain is not involved in kinetochore targeting. However, the data show a clear reduction in kinetochore targeting for these constructs, albeit not as strong as for constructs 4, 5 and 6. The conclusion that the SANT domain has no role in kinetochore targeting therefore seems too strong.

We have considered this point and we would like to confirm that we don't see a clear reduction in localization upon depletion of the SANT domain. To be sure, there was a mix-up of constructs 1 and 2 in panel I (the scheme) that we have now corrected. The correction does not affect our impression that these constructs are largely equivalent. We consider localization under RNAi the most relevant indicator. We also note that the SANT domain, contrary to the 492-872 region, does not compensate for the negative effects on localization of the 5A3 mutations. This confirms our impression that the SANT domain is not a major centromere localization

determinant of M18BP1. We cannot rule out, however, that it may contribute to chromosome recruitment of M18BP1, but this will require further analyses.

10. Figure 6J. The authors call the kinetochore localization of the WT fragment lacking the pre-SANTA and the T653 mutants during G1 phase "robust". However, they seem to be clearly less targeted to the kinetochore compared the 312-872 fragment.

We have replaced "robust" with "clearly localizing".

11. The importance of M18BP1 dimerization for kinetochore targeting is a key observation for this manuscript. However, it remains unexplained why M18BP1 needs to dimerize for kinetochore localization. Given all the interaction interfaces, why are dimers, but not monomers, targeted to kinetochores in absence of M18a/b interaction?

In Figure 7H, we convey the point that the target of the loading machinery is a CCAN dimer, not a monomer. If the binding target is a multimer, dimerization of the binder will increase binding affinity. This is a natural consequence of multivalency, the same effect displayed by the binding of immunoglobulins, which are dimers or pentamers, to their targets. These proteins often recognize multiple copies of a target, e.g. a surface protein on a virus, and their valency increases binding affinity. In this case, we think that the dimerization of M18BP1 through interactions with the Mis18 complex, promotes increased affinity to an otherwise multimeric target.

Dr. Andrea Musacchio
Max Planck Institute of Molecular Physiology
Mechanistic Cell Biology
Otto Hahn Strasse 11
Dortmund 44227
Germany

22nd Dec 2025

Re: EMBOJ-2025-121730R
M18BP1 valency and a distributed interaction footprint determine epigenetic centromere specification in humans

Dear Andrea,

Thank you for submitting your revised manuscript on human M18BP1 roles in centromere specification to The EMBO Journal. Two of the original referees have now assessed it once more, and I am happy to say they both were fully satisfied with the revisions. After incorporation of the following remaining editorial issues, we should therefore be able to proceed with formal acceptance of the study:

- Please adjust the order and the headers of the different manuscript sections: Title page with complete author information, Abstract, Keywords, Introduction, Results, Discussion, Methods, Data Availability, Acknowledgements, Disclosure and Competing Interests Statement, References, Main Figure Legends, Tables.
- Please reduce the number of keywords to max. five (ideally choosing broad general terms), and make sure to place them below the abstract.
- Please carefully go through the reference list and make sure that all references have their complete citation information - including year, journal name & volume, and page/locator numbers - such information is currently missing for several of them, and seems to be repeatedly missed by popular citation manager apps esp. for certain journals. Also, please adjust the format for citation of preprints as specified in our author guidelines:
The citation in the text should be: "(preprint: [author name1] et al, [year])"
The citation in the reference list: "[author name1], [author name2], ... , [author name10] (et al), [year] [article title]. bioRxiv doi: [doi xxx] (preprint)"
- Thank you for providing the required Source Data, but please note that the individual figure panels provided in Excel files should also be labeled.
- Please update the format of the "Data Availability" section, which should only mention data deposited in external repositories. Please include a general access link to each of the utilized databases (e.g. PDB, Proteosafe), plus the respective accession codes for the various datasets. Please also remove referee access information now, and ensure that all datasets become publicly accessible at this point.
- Please remove the Reagents and Tools table from the main article file, and upload it as a separate text file. Also, please make sure to adhere to the template table downloadable from our author guidelines:
<https://www.embopress.org/page/journal/14693178/authorguide#structuredmethods>
- Please make sure to include information related to "n" in the legends of Figures S1 D, E
- Our routine pre-acceptance image checks showed that certain (control) micrographs appear to have been repeatedly used in Figures EV3D, EV4B and EV4F, without any corresponding indication in the respective figure legends. Please double-check and clarify this reason (and possible justification) for this reuse. Even if the reuse should be warranted, it would need to be explicitly mentioned in each of the respective figure legends, to avoid potential misunderstandings.
- On the title page of the Appendix PDF, please include a proper Table of Contents specifying also the page numbers for each of the listed items. Furthermore, the legends for the Appendix figures should always be placed directly under each respective figure.
- Finally, please provide suggestions for a short 'blurb' text prefacing and summing up the conceptual aspect of the study in two sentences (max. 250 characters), followed by 3-5 one-sentence 'bullet points' with brief factual statements of key results of the paper; they will form the basis of an editor-written 'Synopsis' accompanying the online version of the article. Please also upload a synopsis image, which can be used as a "visual title" for the synopsis section of your paper. The image (maybe based on

Figure 7G or 7H?) should be ideally in JPG format, and please make sure that it remains in the modest dimensions of (exactly) 550 pixels wide and between 300-500 pixels high.

I am returning the manuscript to you for a final round of minor revision, to allow you to make these modifications and upload the revised files using the link below. Once we will have received them, I hope we should be ready to proceed with formal acceptance and production of the manuscript. Please do not hesitate to contact me with any questions you should have in this regard.

With kind regards,

Hartmut

*** PLEASE NOTE: All revised manuscript are subject to initial checks for completeness and adherence to our formatting guidelines. Revisions may be returned to the authors and delayed in their editorial re-evaluation if they fail to comply to the following requirements. As a first step please read our guidelines for revised submissions:
<https://link.springer.com/journal/44318/submission-guidelines#cms-Revised-submissions>

1) Every manuscript requires a Data Availability section (even if only stating that no deposited datasets are included). Primary datasets or computer code produced in the current study have to be deposited in appropriate public repositories prior to resubmission, and reviewer access details provided in case that public access is not yet allowed.

4) Each main and each Expanded View (EV) figure should be uploaded as individual production-quality files (preferably in .eps, .tif, .jpg formats). For suggestions on figure preparation/layout, please refer to our Figure Preparation Guidelines:
<https://media.springernature.com/original/springer-cms/rest/v1/content/27825798/data/v1>

6) Please complete our Author Checklist, and make sure that information entered into the checklist is also reflected in the manuscript; the checklist will be available to readers as part of the Review Process File.

8) Please note that supplementary information at EMBO Press has been superseded by the 'Expanded View' for inclusion of additional figures, tables, movies or datasets; with up to five EV Figures being typeset and directly accessible in the HTML version of the article.

9) To facilitate reproducibility and cross-laboratory adoption of methodologies, please structure the Materials & Methods section as outlined in our guide to authors, including a completed Reagents and Tools Table.

10) Digital image enhancement is acceptable practice, as long as it accurately represents the original data and conforms to community standards. If a figure has been subjected to significant electronic manipulation, this must be clearly noted in the figure legend and/or the 'Materials and Methods' section. The editors reserve the right to request original versions of figures and the original images that were used to assemble the figure. Finally, we generally encourage uploading of numerical as well as gel/blot image source data.

In the interest of ensuring the conceptual advance provided by the work, we recommend submitting a revision within 3 months (22nd Mar 2026). Please discuss the revision progress ahead of this time with the editor if you require more time to complete the revisions. Use the link below to submit your revision:

Link Not Available

Referee #1:

I thank the authors for carefully considering my comments and those of the other reviewers. I have now assessed the updated manuscript, and in my view, the authors have appropriately addressed all my major and minor points and have added sufficient experimental data and further textual elaboration. I am happy to support publication.

Referee #2:

Although the revised manuscript does not include all suggested experiments, I find their reasoning/justifications for not including those experiments reasonable. Overall, I am happy with the revision and extend my support to accept this solid manuscript for publication.

All minor editorial requests have been addressed by the authors.

Dr. Andrea Musacchio
Max Planck Institute of Molecular Physiology
Mechanistic Cell Biology
Otto Hahn Strasse 11
Dortmund 44227
Germany

15th Jan 2026

Re: EMBOJ-2025-121730R1
M18BP1 valency and a distributed interaction footprint determine epigenetic centromere specification in humans

Dear Andrea,

Thank you for submitting your final revised manuscript for our consideration. I am pleased to inform you that we have now accepted it for publication in The EMBO Journal.

You may qualify for financial assistance for your publication charges - either via a Springer Nature fully open access agreement or an EMBO initiative. Check your eligibility: <https://link.springer.com/journal/44318/how-to-publish-with-us>

With kind regards,

Hartmut

Please note that it is The EMBO Journal policy for the transcript of the editorial process (containing referee reports and your response letters) to be published as an online supplement to each paper. If you should prefer removal of any referee-only figures included in the point-by-point response(s), e.g. because they may still be used for future publication or because they have been reproduced from published work by others, please do let us know immediately via response email.

More information is available here: <https://link.springer.com/partners/embo-press/editorial-policies#Peer%20review>